



# Bromine from short–lived source gases in the Northern Hemisphere UTLS

Timo Keber[1], Harald Bönisch[1,2], Carl Hartick[1,3], Marius Hauck[1], Fides Lefrancois[1], Florian Obersteiner[1,2], Akima Ringsdorf[1,4], Nils Schohl[1], Tanja Schuck[1], Ryan Hossaini[5], Phoebe Graf[6], Patrick Jöckel[6] and Andreas Engel[1].

[1]University of Frankfurt, Institute for Atmospheric and Environmental Sciences, Altenhöferallee 1, 60438 Frankfurt, Germany
[2]now at Karlsruhe Institute of Technology, Institute of Meteorology and Climate Research, Hermann-von-Helmholtz-Platz 1, 76344 Eggenstein-Leopoldshafen, Germany.
[3]now at Research Centre Jülich, Institute for Agrosphere (IBG-3), Wilhelm-Johnen-Straße, 52428 Jülich, Germany
[4] now at Max Planck Institute for Chemistry, Hahn-Meitner-Weg 1, 55128 Mainz, Germany
[5]Lancaster Environment Centre, Lancaster University, Lancaster, UK.
[6] Institut für Physik der Atmosphäre, Deutsches Zentrum für Luft- und Raumfahrt (DLR), Oberpfaffenhofen, Germany

*Correspondence to*: Timo Keber and Andreas Engel (keber and an.engel@iau.uni-frankfurt.de)

**Abstract.**

We present novel measurements of five short-lived brominated source gases ($CH_2Br_2$, $CHBr_3$, $CH_2ClBr$, $CHCl_2Br$ and $CHClBr_2$) obtained using a gas chromatograph-mass spectrometer system on board the High Altitude and Long Range Research Aircraft (HALO). The instrument is extremely sensitive due to the use of chemical ionisation, allowing detection limits in the lower parts per quadrillion ($10^{-15}$) range. Data from three campaigns using the HALO aircraft are presented, where the Upper Troposphere/Lower Stratosphere (UTLS) of the Northern Hemisphere mid to high latitudes were sampled during winter and during late summer to early fall. We show that an observed decrease with altitude in the stratosphere is consistent with the relative lifetimes of the different compounds. Distributions of the five source gases and total organic bromine just below the tropopause shows an increase in mixing ratio with latitude, in particular during polar winter. This increase in mixing ratio is explained by increasing lifetimes at higher latitudes during winter. As the mixing ratio at the extratropical tropopause are generally higher than those derived for the tropical tropopause, extratropical troposphere-to-stratosphere transport will result in elevated levels of organic bromine in comparison to air transported over the tropical tropopause. The observations are compared to model estimates using different emission scenarios. A scenario which has emissions most strongly concentrated to low latitudes cannot reproduce the observed latitudinal distributions and will tend to overestimate bromine input through the tropical tropopause from $CH_2Br_2$ and $CHBr_3$. Consequently, the scenario also overestimates the amount of brominated organic gases in the stratosphere. The two scenarios with the highest overall emissions of $CH_2Br_2$ tend to overestimate mixing ratios at the tropical tropopause but are in much better agreement with extratropical tropopause values, showing that not only total emissions but also latitudinal distributions in the emissions are of importance. While an increase in tropopause values with latitude is reproduced with all emission scenarios during winter, the simulated extratropical tropopause values are on average lower than the observations during late summer to fall. We show that a good knowledge of the latitudinal distribution of tropopause mixing ratios and of the fractional contributions of tropical and extratropical air is needed to derive stratospheric inorganic bromine in the lowermost stratosphere from observations. Depending on the underlying emission scenario, differences of a factor 2 in reactive bromine derived from observations and model outputs are found for the lowermost stratosphere, based on source gas injection. We conclude that a good representation of the contributions of different source regions is required in models for a robust assessment of the role of short-lived halogen source gases on ozone depletion in the UTLS.



## 1. Introduction

Following the detection of the ozone hole during springtime over Antarctica (Farman et al., 1985) and the attribution of the decline in both polar and global ozone to the emissions of man-made halogenated compounds (see e.g. Molina and Rowland, 1974;Solomon, 1999;Engel and Rigby, 2018), production and use of long-lived halogenated species, in particular

chlorofluorocarbons (CFCs), have been regulated by the Montreal Protocol (WMO, 2018). This has led to decreasing levels of chlorine in the atmosphere (Engel and Rigby, 2018), despite recent concerns over ongoing emissions of CFC-11, which have been attributed to unreported and thus illegal production (Montzka et al., 2018;Engel and Rigby, 2018;Rigby et al., 2019). Bromine reaching the stratosphere has been identified as an even stronger catalyst for the depletion of stratospheric ozone than chlorine (Wofsy et al., 1975;Sinnhuber et al., 2009). Its relative efficiency on a per molecule basis is currently estimated to be

60-65 times larger than that of chlorine (see discussion in Daniel and Velders, 2006). Long-lived bromine gases include $CH_3Br$ with partly natural and partly anthropogenic sources and halons, which are of purely anthropogenic origin. Next to long-lived gases, some chlorine and bromine from so-called "very short-lived substances" (VSLS), i.e. substances with atmospheric lifetimes less than 6 months, can reach the stratosphere. It has been estimated that for the year 2016, about 25% of the bromine entering the stratosphere is from VSLS (Engel and Rigby, 2018). Due to the decline in chlorine and bromine from long-lived

species, the relative contribution of short-lived species to stratospheric halogen loading is expected to increase, also driven by increasing anthropogenic emissions of some short-lived chlorinated halocarbons (Hossaini et al., 2017;Oram et al., 2017;Leedham Elvidge et al., 2015;Engel and Rigby, 2018;Hossaini et al., 2019).

A number of factors control the abundance of ozone at mid-latitudes, including known important influences from dynamics,

chemical destruction, aerosol loading and the solar cycle (e.g. Feng et al., 2007;Harris et al., 2008;Dhomse et al., 2015). In the lowermost stratosphere, the breakdown of VSLS provides a significant bromine source, in a region where (a) ozone loss cycles involving bromine chemistry are known to be important (e.g. Salawitch et al., 2005) and (b) on a per-molecule basis, ozone perturbations have a relatively large radiative effect (Hossaini et al., 2015). At present, VSLS are estimated to supply a total of ~5 (3-7) parts per trillion (ppt, $10^{-12}$) Br to the stratosphere, with source gas injection estimated to provide 2.2 (0.8-4.2) ppt

Br and product gas injection 2.7 (1.7-4.2) ppt Br (Engel and Rigby, 2018). Attribution of lower stratospheric ozone trends is complex and trends in this region are highly uncertain (Steinbrecht et al., 2017;Ball et al., 2018;Chipperfield et al., 2018). It has been suggested that continuing negative ozone trends observed in the lower stratosphere (defined as about 13 to 24 km in the mid latitudes) may partly be related to increasing anthropogenic and natural VSLS (Ball et al., 2018). While Chipperfield et al. (2018) suggested that the main driver for variability and trends in lower stratospheric ozone is dynamics rather than

chemistry, the bromine budget of the upper troposphere and lower stratosphere (UTLS) needs to be well understood.

In the past, the main focus of upper tropospheric bromine studies for VSLS has been on the tropics, as this is the main entry region for air masses to reach above 380 K potential temperatures (see e.g. discussion in Engel and Rigby, 2018) and thus for the main part of the stratosphere. However, as many authors have shown, the lowermost stratosphere, i.e. the part of the

stratosphere situated below 380 K but above the extratropical stratosphere, is influenced by transport from the tropics and from the extratropics (e.g. Holton et al., 1995;Gettelman et al., 2011;Fischer et al., 2000;Hoor et al., 2005). Some authors have quantified the fraction of air in the lowermost stratosphere, which did not pass the tropical tropopause from tracer measurements (Hoor et al., 2005;Boenisch et al., 2009;Ray et al., 1999;Werner et al., 2010) and others have used trajectory analyses to study mass fluxes and stratosphere-troposphere exchange (e.g. Stohl et al., 2003;Wernli and Bourqui, 2002;Škerlak

et al., 2014;Appenzeller et al., 1996). Based on tracer measurements of mainly CO, Hoor et al. (2005) estimated that the fraction of air with extratropical origin in the mid latitude lowermost stratosphere of the Northern Hemisphere ranged between about 35% during winter and spring to about 55% during summer and fall. Using a different approach based on $CO_2$ and $SF_6$ observations, Boenisch et al. (2009) found a similar seasonality but higher extratropical fractions, which were consistently



higher than 70% during summer and fall and values above 90% in the entire lowermost stratosphere during October. Similarly, Boenisch et al. (2009) also derived much lower fractions of air with recent extratropical origin during winter and spring, which was sometimes as low as 20% during April. It has also been argued that the relative role of different source regions for the UTLS could alter with a changing circulation (Boothe and Homeyer, 2017).

Both extratropical and tropical source regions are important for the lowermost stratosphere. A recent compilation of entry mixing ratios of brominated VSLS to the stratosphere (Engel and Rigby, 2018) has focused on values representative of the tropical tropopause. Two pathways for input of halogens from short-lived gases are discussed: Source Gas Injection (SGI), where the halogen is transported to the stratosphere in the form of the source gases; and Product Gas Injection (PGI), where

photochemical breakdown products of source gases are transported into the stratosphere, usually in inorganic form (i.e. $Br_y$). While halogens transported into the stratosphere due to PGI are usually directly in a form available for catalytic ozone depletion reactions, halogens from source gases must first be released in the stratosphere photochemically. Due to the short lifetimes of VSLS, this release is expected to occur in the lowest part of the stratosphere. Therefore, brominated VSLS are particularly effective with respect to ozone chemistry in the lower and lowermost stratosphere, below about 20 km, with the associated

ozone decreases exerting a significant radiative effect (Hossaini et al., 2015). It has been shown that observed and modelled ozone show a better agreement if bromine from short-lived species is included in models (Sinnhuber and Meul, 2015;Fernandez et al., 2017;Oman et al., 2016). In particular for the Antarctic ozone hole, an enhancement in size by 40% and an enhancement in mass deficit by 75% was simulated due to VSLS (Fernandez et al., 2017) in comparison with a model run without VSLS. A delay in polar ozone recovery by about a decade has also been reported due to the inclusion of brominated VSLS (Oman et

al., 2016). In order to have solid projections on the effect of VSLS on ozone and climate, a good knowledge of their atmospheric distribution is thus needed for models.

The main source of brominated VSLS is believed to be from oceans and in particular from coastal regions. Four global emission scenarios of short-lived brominated gases have been proposed (Warwick et al., 2006;Ordoñez et al., 2012;Ziska et al.,

2013;Liang et al., 2010) , with variations in VSLS source strengths of more than a factor of two between them (Engel and Rigby, 2018). In the past, these scenarios have been compared to each other and to observations; large differences have been identified in modelled tropospheric mixing ratios of $CHBr_3$ and $CH_2Br_2$, along with estimates of stratospheric bromine input (Hossaini et al., 2013;Hossaini et al., 2016;Sinnhuber and Meul, 2015). It has also been proposed that VSLS emissions may have increased by 6-8 % between 1979-2013 (Ziska et al., 2017), although no observational evidence for this has been found

(Engel and Rigby, 2018). A further future increase has been projected (Ziska et al., 2017;Falk et al., 2017), although this projection is very uncertain and the processes associated with the oceanic production of brominated VSLS are still poorly understood. It has also been proposed that certain source regions could be more effective with respect to transport to the stratosphere, in particular the Indian Ocean, the Maritime Continent and the tropical Western Pacific (Liang et al., 2014;Fernandez et al., 2014;Tegtmeier et al., 2012). The Asian monsoon has also been named as a significant pathway for

transport of bromine from VSLS to the stratosphere (Liang et al., 2014;Fiehn et al., 2017;Hossaini et al., 2016).

While most investigations of natural VSLS focused on tropical injection of bromine to the stratosphere, in this study we focus on the extratropical bromine VSLS budget. In order to investigate the regional variability of bromine input into the lowermost stratosphere and the inorganic bromine loading of the extratropical lowermost stratosphere, we have performed a range of

airborne measurement campaigns using an in-situ gas chromatograph (GC) coupled to a mass spectrometer (MS) on board the High Altitude and Long Range Research Aircraft (HALO). These observations are compared with results from state-of-the-art atmospheric models run with the different emission scenarios mentioned above. The implications for stratospheric reactive bromine from observations and from models are discussed. In Section 2 we give a brief introduction into the instrument, the





available observations and the models used for this study. Typical distributions of brominated VSLS derived from these observations are then presented in Section 3 and compared to model output from two different atmospheric models in Section 4. Finally, in Section 5 the implications of the observations for inorganic bromine in the stratosphere are discussed.

## 2. Observations and models.

5 **Observations**

The data presented here have been measured with the in-situ **G**as **Ch**romatograph for **O**bservational **S**tudies using **T**racers (GhOST-MS) deployed on board the HALO aircraft. GhOST-MS is a two channel GC instrument. An isothermal channel uses an Electron Capture Detector (ECD), in a similar set-up as used during the SPURT campaign (Boenisch et al., 2009;Boenisch et al., 2008;Engel et al., 2006) to measure $SF_6$ and CFC-12 with a time resolution of one minute. The second channel is 10 temperature-programmed and uses a cryogenic pre-concentration system (Obersteiner et al., 2016;Sala et al., 2014) and a mass spectrometer (MS) for detection. It is similar to the set-up described by Sala et al. (2014) and measures halocarbons in the chemical ionization mode (e.g. Worton et al., 2008) with a time resolution of 4 minutes. The instrument is tested for non-linearities, memory and blank signals, which are corrected where necessary (see the description in Sala (2014) and Sala et al. (2014) for details). Here we focus on the brominated hydrocarbons measured with GhOST-MS (see Table 1). Table 1 also 15 includes typical local lifetimes of the different VSLS species and the global lifetimes of the long-lived species. The instrument was deployed during several campaigns of the German research aircraft HALO, providing observations in the UTLS over a wide range of latitudes and different seasons mainly in the Northern Hemisphere. Some observations from the Southern Hemisphere are also available, but due to their sparsity will not be part of this work.

20 GhOST-MS measurements from three HALO Missions will be presented and discussed here. The first atmospheric science mission of HALO was TACTS (Transport and Composition in the Upper Troposphere/Lowermost Stratosphere), conducted between August and September 2011, with a focus on the Atlantic sector of the mid latitudes of the Northern Hemisphere. The second campaign was PGS, a mission consisting of three sub-missions: **P**OLSTRACC (The Polar Stratosphere in a Changing Climate), **G**W-LCYCLE (Investigation of the Life cycle of gravity waves) and **S**ALSA 25 (Seasonality of Air mass transport and origin in the Lowermost Stratosphere). PGS took place mainly in the Arctic between December 2015 and March 2016. Finally, the GhOST-MS was deployed during the WISE (Wave driven isentropic exchange) mission between September and October 2017. The dates of the missions and some parameters on the available observations are summarized in Table 2 and the flight tracks are shown in Figures 1 and 2. As the WISE and TACTS campaigns covered a similar time period and latitude range, we have chosen to present the results from both campaign in a merged format, i.e. the 30 data from the two campaigns have been combined to single data set which we will refer to as "WISE_TACTS". Figure 3 shows an example time series of Halon-1301 ($CF_3Br$), $CH_2Br_2$ and $CHBr_3$, ozone and mean age calculated from the $SF_6$ measurements obtained during a typical flight in the Arctic in January 2016. It is clearly visible that the halocarbons are correlated amongst each other, whereas they are anticorrelated with ozone and mean age. It is further evident from Figure 3 that the shortest-lived halocarbon measured by GhOST-MS, i.e. $CHBr_3$, decreases much faster with increasing ozone than the longer-lived $CH_2Br_2$ 35 or the long- lived source gas Halon 1301. Note that the local lifetimes of the halocarbons may differ significantly from their typical mid latitude lifetimes shown in Table 1. Lifetimes generally increase with a) decreasing temperature for species with a sink through the reaction with the OH radical and b) with decreasing solar irradiation for species with direct photolytic sink. Therefore, in particular during winter, lifetimes are estimated to increase considerably with increasing latitude due to the decreased solar illumination and low temperatures.

40





### Models and Meteorological Data

Data from two different models were used in this study: ESCiMo (Earth System Chemistry ntegrated Modelling) data from the EMAC (ECHAM/MESSy Atmospheric Chemistry) chemistry climate model (CCM) and the TOMCAT (Toulouse Off-line Model of Chemistry And Transport) chemistry transport model (CTM).

For EMAC data, we used results from the simulations in the so-called specified dynamics mode, for which the model was nudged towards ERA-Interim meteorological reanalysis data from European Centre for Medium-Range Weather Forecasts (ECMWF; (Dee et al., 2011). T42 spectral model resolution was used, corresponding to a quadratic Gaussian grid of approximately 2.8° by 2.8° horizontal resolution, and the vertical resolution comprised 90 sigma-hybrid pressure levels up to 0.01 hPa. The model output has been subsequently interpolated to pressure levels between 1000 and 0.01 hPa. The emissions

of VSLS were taken from Warwick et al. (2006). The model setup for the ESCiMo simulation is described in detail by Jöckel et al. (2016).

The TOMCAT (Toulouse Off-line Model of Chemistry And Transport) model (Chipperfield, 2006;Monks et al., 2017) is driven by analyzed wind and temperature fields taken from the ECMWF ERA-Interim product. Here, the model was run with T42 horizontal resolution (2.8° 2.8°) and with 60 vertical levels, extending from the surface to ~60 km. This configuration of

the model has been used in a number of VSLS-related studies and is described by Hossaini et al. (2019). In this study, three different VSLS emission scenarios are used with TOMCAT (Liang et al., 2010;Ordoñez et al., 2012;Ziska et al., 2013). Emitted VSLS ($CHBr_3$, $CH_2Br_2$, $CH_2BrCl$, $CHBr_2Cl$ and $CHBrCl_2$) are destroyed by reaction with OH and photolysis in the model, calculated using the relevant kinetic data from Burkholder et al. (2015).

Local tropopause information for the flights with HALO have been derived from ERA-interim data (J.U.Grooß, FZ Jülich,

private communication). The climatological tropopause has been calculated based on potential vorticity (PV) according to the method described in Škerlak et al. (2015) and Sprenger et al. (2017) based on the ERA-Interim reanalysis (M. Sprenger, ETH Zürich, private communication). As the PV tropopause is not physically meaningful in the tropics, the level with a potential temperature of 380 K has been adapted for the tropopause where the 2-PVU (Potential Vorticity Unit) level is located above the 380-K level.

**3.  Observed distribution and atmospheric gradients of different brominated VSLS**

Spatial distributions are shown in tropopause-relative coordinates and as functions of equivalent latitude. As equivalent latitude is mainly a useful horizontal coordinate for the stratosphere, we chose to use standard latitude for all measurements below the tropopause and equivalent latitude for all measurements above the tropopause. We refer to this coordinate as equivalent latitude*. As the observations typically cover a range of latitudes, vertical profiles are shown for 20° bins. In the vertical

direction, three different coordinates are used in this paper. These are potential temperature $\theta$, potential temperature above the tropopause $\Delta\theta$, and finally a coordinate we refer to as $\theta^*$, which is calculated by adding the potential temperature of the mean tropopause to $\Delta\theta$. We used the dynamical tropopause, defined by a potential vorticity value of 2 PVU or by a potential temperature value of 380 K in the tropics (see Section 2), as a reference surface.

### Mean vertical profiles.

All measurements from the individual campaigns have been binned into 10 K potential temperature bins between -40 and 100 K of $\Delta\theta$. In addition, we have also binned the data in potential temperature in 10 K potential temperature intervals ranging from 40 K below the mean tropopause to 100 K above the mean tropopause. In this way, the centers of the $\Delta\theta$ and $\theta$ bins are the same relative to the mean tropopause observed during the campaign. The results are presented for the two main bromine species $CH_2Br_2$ and $CHBr_3$, averaged over equivalent latitude* of 40-60°N in Figure 4 for the PGS winter campaign and the

WISE_TACTS combined data set representing late summer to fall conditions. Only bins which contain at least five data points



have been included in the analysis. The results are also summarized in Tables 3 and 4 for the same latitude intervals for all species and for total bromine derived from the five brominated VSLS. The tropopause mole fractions shown in Tables 3 and 4 have been derived as the average of all values in that latitude interval and within 10 K below the tropopause. The potential temperature of the average tropopause has been used for $\theta$ averaging, while the potential temperature difference to the local

tropopause has been used as reference when averaging in $\Delta\theta$ coordinates. In the WISE_TACTS data set, total organic bromine at the dynamical tropopause between 40 and 60 °N was 3.4 and 3.6 ppt, using $\Delta\theta$ and $\theta$ as vertical coordinates, respectively. Higher values of total bromine were found during the winter campaign PGS, when average tropopause values were 5.2 and 4.9 ppt both using $\Delta\theta$ and $\theta$ as vertical coordinates. These values are considerably higher than the tropical tropopause values of organic bromine derived in the vicinity of the tropical tropopause (Engel and Rigby, 2018) as will be discussed in detail

below. When using the WMO definition of the tropopause, the total bromine at mid-latitudes was about 0.3 to 0.5 ppt lower than using the PV tropopause, reflecting the fact that the WMO tropopause is usually slightly higher than the dynamical tropopause using the 2 PVU definition (e.g. Gettelman et al., 2011).

Of all species discussed here, $CH_2BrCl$ showed the smallest vertical gradients and $CHBr_3$ the largest. This is well in line with their atmospheric lifetimes (see Table 1), which is shortest for $CHBr_3$ and longest for $CH_2BrCl$. $CHBr_2Cl$ showed the second

strongest vertical gradients, while $CH_2Br_2$ and $CHBrCl_2$ usually showed comparable relative decreases with altitude, again in line with the atmospheric lifetime, which will generally decrease with an increase in bromine atoms in the molecule. The strongest vertical gradients with respect to both $\theta$ and $\Delta\theta$ were observed during the winter campaign PGS, with the exception of $CHBr_3$, which was nearly completely depleted for all campaigns at 40 K above the tropopause and thus shows very similar averaged gradients over this potential temperature region. When evaluated only for the first 20 K above the tropopause, the

gradient of $CHBr_3$ was also highest during PGS.

We further determined the variability of the different species in 10 K intervals of $\theta$ and $\Delta\theta$. For all campaigns, the variability averaged over the four lowest stratospheric bins was always lower when using $\Delta\theta$ as a coordinate. In the troposphere, the variability is very similar for $\theta$ and $\Delta\theta$ coordinates, indicating that the variability in the free troposphere is not strongly influenced by the potential temperature of the tropopause. The observed variabilities were found to be very similar for the

WMO and PV tropopause definitions (not shown). As the dynamical PV tropopause is generally expected to be better suited for tracer studies, we decided to reference all data to the dynamical tropopause.

### Latitude altitude cross sections

As in previous work (e.g. Boenisch et al., 2011;Engel et al., 2006), we present the latitudinal distribution as a zonal mean and using equivalent latitude and potential temperature as horizontal and vertical coordinates. However, we propose a somewhat

different approach here, in which equivalent latitude* is used as a horizontal coordinate, i.e. latitude for all tropospheric observations and equivalent latitude for observations at or above the tropopause. As a vertical coordinate we have chosen to use a modified potential temperature coordinate, which we refer to as $\theta^*$ and which is calculated by adding the potential temperature of the climatological tropopause to $\Delta\theta$. In this way, all measurements are presented relative to a climatological tropopause, which has been derived from ERA-Interim reanalysis as zonal mean for the latitude of interest and the specific

months of the campaign (see Section 2 for campaign details). This is expected to reduce variability by applying the information from $\Delta\theta$, yet the absolute vertical information is also maintained. In order to ensure that this tropopause value is representative also for the period of our observations, we compare the potential temperature of the campaign-based tropopause, averaged for all the location and times when we have observations, with this climatological tropopause. For the latitude band between 40 and 60°N, the climatological PV tropopause for the TACTS_WISE time period was derived to be at 329 K, in excellent

agreement with the campaign-based tropopause, which was also at 329 K. For the PGS campaign, both the climatological tropopause and the campaign-based tropopause were found to be at 312 K. In contrast to the campaign-based tropopause, the



climatological tropopause is also available for latitude bands and longitudes not covered by our observations and will be more representative for typical conditions during the respective season and latitude.

Figure 5 shows the distributions of the two main bromine gases $CH_2Br_2$ and $CHBr_3$ in the coordinate system discussed above

for the two campaign seasons (PGS: winter; WISE_TACTS: late summer to early fall). The data have been binned in 5° latitude and 5 K intervals of potential temperature. As expected, the distributions closely follow the tropopause (indicated by the dashed line), with values decreasing with distance to the tropopause and also with increasing equivalent latitude. The distributions observed during the WISE and the TACTS campaigns show significant amounts of $CH_2Br_2$, which has a rather long lifetime in the cold upper troposphere and lower stratosphere (Hossaini et al., 2010) even quite deep into the stratosphere. The shorter-

lived $CHBr_3$ is strongly depleted already about 20 K above the tropopause. In the case of the winter campaign PGS, values close to zero at the highest flight levels are also observed for the longer-lived $CH_2Br_2$, indicating that in the most stratospheric air masses observed during PGS nearly all bromine from VSLS has been converted to inorganic bromine. This is in agreement with the observation of air masses with very high mean age of air derived from $SF_6$ observations of GhOST-MS (see e.g. Figure 3), reaching up to 5 years for the oldest air (not shown). This is air which has descended inside the polar vortex and has

not been in contact with tropospheric sources for a long time, allowing even the longer-lived $CH_2Br_2$ to be nearly completely depleted.

**Upper tropospheric latitudinal gradients**

If air is transported into the lowermost stratosphere via exchange with the extratropical upper troposphere, the levels of organic bromine compounds are likely to be different than for air being transported into the stratosphere via the tropical tropopause.

In order to investigate the variability and the gradient in the upper tropospheric input region, we binned our data according to latitude and to potential temperature difference to the tropopause. In order to characterize the input region, we have chosen to average all data in a range of 10 K below the local dynamical tropopause. Again, for the tropospheric data, standard latitude has been chosen, while equivalent latitude was used for all data with $\Delta\theta$ above zero. The latitudinal gradients are shown in Figure 6 for $CH_2Br_2$, $CHBr_3$ and total bromine derived from the sum of all VSLS (including the mixed bromochlorocarbons

$CH_2BrCl$, $CHBrCl_2$ and $CHBr_2Cl$), each weighted by the amount of bromine atoms. For the tropical tropopause, input values from different measurement campaigns have recently been reviewed by Engel and Rigby (2018). They found that total bromine from these five compounds averaged between 375 and 385 K, i.e. around the tropical tropopause was 2.2 (0.8-4.2) ppt, and in the upper TTL (365-375K potential temperature) was around 2.8 (1.2-4.6) ppt. The average values derived here for the 10 K interval below the extratropical tropopause are significantly larger. For the late summer to early fall data from TACTS and

WISE, they increase from 2.6 ppt around 30°N (20-40° N equivalent latitude*) to 3.8 ppt around 50°N (40-60°N equivalent latitude*), while no further increase is found for higher latitudes with a value of 3.4 ppt. For the winter measurements during PGS a clear increase with latitude is observed from 3.3 ppt around 30°N (20-40°N equivalent latitude*) to 3.8 ppt around 50°N (40-60°N equivalent latitude*) to 5.5 ppt in the high latitudes (60-80°N equivalent latitude*). There is considerable variability in these values derived around the tropopause, due to the short lifetime of these compounds and the high variability in emissions

depending on the source region. Nevertheless, there is a clear tendency for an increase in tropopause values with latitude, particularly during Northern Hemisphere winter. This is most probably related to the increase in lifetime with latitude, as especially during the wintertime PGS campaign the photolytical breakdown in higher latitudes is significantly slower than in lower latitudes. Additional effects due to the sources and their latitudinal, seasonal and regional variability cannot be excluded. However, we note that emissions are most likely to be largest during summer, as shown e.g. in Hossaini et al. (2013), which

would not explain the large values of brominated VSLS in the upper troposphere in high latitudes during winter.

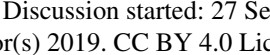



### 4. Comparison with model derived distributions.

As bromocarbons are an important source of stratospheric bromine, it is worthwhile to investigate if current models can reproduce the observed distributions shown in Section 3, and are thus able to realistically simulate the input of bromine from VSLS to the stratosphere, but also the further chemical breakdown and the transport processes related to the propagation of these gases in the stratosphere. As explained in Section 2, we used two different models, with different emissions scenarios

for the brominated very short-lived source gases. The ESCiMo simulation results from the chemistry climate model EMAC (Jöckel et al., 2016) are based on the emission scenario by Warwick et al. (2006), while the TOMCAT model (Hossaini et al., 2013) was run with three different emission scenarios (Ordoñez et al., 2012;Ziska et al., 2013;Liang et al., 2010). Both models have been used in the past to investigate the effect of brominated VSLS on the stratosphere (e.g. Sinnhuber and Meul,

2015;Hossaini et al., 2012;Wales et al., 2018;Hossaini et al., 2015;Graf, 2017). For the EMAC model, we have chosen to use results from a so called "specified dynamics" simulation, which has been extended from the ESCiMo simulations to cover our campaign time period (see Section 2). The model data have been extracted for the time period and latitude ranges of the observations and have been zonally averaged. Here we compare vertical profiles, geographical distributions and latitudinal gradients between our observations and the model results, in a similar way as the observations have been presented in Section

3. We also compare results for total bromine. Only the scenarios of Warwick et al. (2006) and Ordoñez et al. (2012) contain emissions of the mixed bromochlorocarbons $CH_2BrCl$, $CHBrCl_2$ and $CHBr_2Cl$. For the calculation of total VSLS bromine based on the emission scenarios by Liang et al. (2010) and Ziska et al. (2013) we have therefore adopted the results from the TOMCAT model using the emissions by Ordoñez et al. (2012). The contribution from these mixed bromochlorocarbons to total VSLS bromine are typically on the order of 20%, while about 80% of total VSLS bromine in the upper troposphere and

lower stratosphere is due to $CH_2Br_2$ and $CHBr_3$.

### Mean vertical profiles.

Observed vertical profiles are available up to the maximum flight altitude of the HALO aircraft, which is about 15 km, corresponding to about 400 K in potential temperature. Due to the variability of the tropopause potential temperature, this translates into maximum values of $\Delta\theta$ of about 100 K. The emphasis of this Section is on the mid latitudes of the northern

hemisphere, i.e. values averaged between 40 and 60° equivalent latitude*. All comparison are shown as function of $\Delta\theta$. As no direct tropopause information was available for the TOMCAT output, we have chosen to derive $\Delta\theta$ for this comparison from the difference between model potential temperature and the potential temperature of the climatological zonal mean tropopause, which has been derived as explained in Section 2. We have also compared this climatological tropopause with the tropopause derived from the EMAC model results for the time of our campaigns. The potential temperature of the EMAC tropopause and

the climatological tropopause differed by less than 3 K for all campaigns at mid latitudes.

Figure 7 presents the model-measurement comparisons for two bromine species for the winter PGS campaign and for the combined dataset from WISE and TACTS. Overall the Liang et al. (2010) and the Ordoñez et al. (2012) emission scenarios give the best agreement with our observations of $CH_2Br_2$, with an averaged deviation of 0.1 ppt or less, averaged over all campaigns and all stratospheric measurements in the 40-60°N equivalent latitude band, corresponding to a mean absolute

percentage difference (MAPD) on the order of 10-25%. Using the Ziska et al. (2013) emissions, $CH_2Br_2$ is overestimated in the mid latitude lowest stratosphere during both campaigns by about 0.2 ppt, corresponding to about 40-60% overestaimtion. Using the Warwick et al. (2006) emissions in the EMAC model, the overestimation is even larger with 0.25-0.3 ppt, corresponding to 50-70%. As $CHBr_3$ is nearly completely depleted in the upper part of the profiles, differences will become negligible there. Therefore, we only compared values in the lowest 50 K potential temperature above the tropopause. In this

region, the best agreement is again found with the Liang et al. (2010) and Ordoñez et al. (2012) emission scenarios, with mean differences always below 0.1 ppt, corresponding to about a MAPD of 20-30%. Using the Ziska et al. (2013) emission scenario





we find an underestimation on the order of 0.05-0.1 ppt (40-70%), while CHBr$_3$ is overestimated by about 0.15 ppt (120-180%) in the EMAC model based on the Warwick et al. (2006) emission scenario.

Using the Ziska et al. (2013) emission scenario, the overestimation of CH$_2$Br$_2$ and the underestimation of CHBr$_3$ tend to cancel out, resulting in a reasonable agreement in total VSLS bromine. Because of the different chemical lifetimes of the two species,

this results in a wrong vertical distribution of Br$_y$ with too high mixing ratios above 20 K above the tropopause in winter and a much steeper vertical gradient in late summer. The EMAC model with the Warwick et al. (2006) emissions significantly overestimates both CH$_2$Br$_2$ and CHBr$_3$ in the lowermost stratosphere of the mid latitudes. The vertical profiles of CH$_2$Br$_2$ and CHBr$_3$ from the EMAC model with the Warwick et al. (2006) emission scenario is therefore completely different from the observations, showing a maximum around the tropopause or even above.

In order to investigate if this large deviation when using the EMAC model with the Warwick et al. (2006) emission scenario is due to the model or due to the specific emission scenario, we additionally compare model data from EMAC simulations using all four emission scenarios (Graf, 2017). Note that these simulations are only available for the time period up to 2011 and not in the specified dynamics mode. This comparison for the January-March period (representative for the PGS campaign) is shown in Figure 8 for CH$_2$Br$_2$ and CHBr$_3$. Fig. 8 looks qualitatively very similar to the comparisons in Figure 7, i.e. both

CH$_2$Br$_2$ and CHBr$_3$ using the Warwick et al. (2006) emission scenario show highest values in the lower stratosphere and CHBr$_3$ shows the least pronounced vertical gradients. Also, the pattern for the Ziska et al. (2013) emission scenario are the same, with second highest CH$_2$Br$_2$ values and lowest CHBr$_3$ values. It is therefore clear that the observed differences are not primarily caused by the model but rather by the emission scenarios.

**Latitude altitude cross sections**

As has been shown in the comparison of the vertical profiles, significant differences between model results and observations are found, especially in the case of the Ziska et al. (2013) emissions in the TOMCAT model and in case of the Warwick et al. (2006) emissions in the EMAC model for the Northern Hemisphere mid latitudes (40-60°N). To visualize these differences, we present latitude-altitude cross sections of the model data sets and the differences to our observations in Fig. 9 and 10. Again, we use equivalent latitude* as the latitudinal coordinate for the observations and $\theta^*$ as vertical coordinate. For the model

results, the zonal mean data are displayed as function of latitude and potential temperature $\theta$. The comparison is shown here for the winter data set from PGS, for which the observational set covers a wide range of latitudes and also reaches very low tracer mole fractions. The comparison for the late summer to fall campaigns (TACTS and WISE) gives a rather similar picture (not shown). The overall best agreement in the vertical profiles has been found for the TOMCAT model using the emissions scenarios by Liang et al. (2010) and Ordonez et al. (2012). The latitude-altitude cross section for these two datasets are therefore

shown in Figure 9 and 10. Using these two emissions scenarios, the TOMCAT model tends to overestimate high latitude tropospheric mole fractions of CHBr$_3$ during this winter campaign. However, the stratospheric distribution is rather well reproduced with absolute deviations to the model mostly being below 0.1 ppt. In the case of CH$_2$Br$_2$, overall stratospheric mole fractions are slightly larger in the model results compared to the observations. The deviations between the TOMCAT model using the Ziska et al. (2010) emissions and the EMAC model using the Warwick et al. (2006) emissions are significantly larger.

These are shown in Figures 11 and 12 again for the PGS campaign. As noted above, the TOMCAT model with the Ziska et al. (2013) emissions overestimates stratospheric CH$_2$Br$_2$, while stratospheric CHBr$_3$ is reasonably well captured. The largest discrepancies between model and observations are observed in the case of the EMAC model with the Warwick et al. (2006) emissions. In this case, both CH$_2$Br$_2$ and CHBr$_3$ are overestimated significantly in the lower stratosphere.

The direct comparison of the distributions between the different model data sets is also interesting. In the case of CHBr$_3$, the

two emission scenarios which have a more even distribution of emissions with latitude, i.e, the emission scenarios by Liang et al. (2010) and Ordoñez et al. (2012) show the best agreement with the observations. The emission scenario by Warwick et al. (2006) yields much higher mole fractions in the tropics and has the poorest agreement with measurement data. The emission





scenario by Ziska et al. (2013) yields overall much lower CHBr$_3$ in large parts of the atmosphere and seems to be the only setup in which mid latitude tropopause mole fractions of CHBr$_3$ are underestimated in comparison to our observations. For CH$_2$Br$_2$, again the Ordoñez et al. (2012) and Liang et al. (2010) emission scenarios in the TOMCAT model result in rather similar distributions and rather good agreement with our observations. In the case of the TOMCAT model with the Ziska

emissions, very high mole fractions of CH$_2$Br$_2$ are simulated throughout the tropics. Our low latitude observations from HALO and the values compiled in the WMO 2018 report for the tropics (Engel and Rigby, 2018) are much lower than the values of CH$_2$Br$_2$ in the tropics using the Warwick et al. (2006) and Ziska et al. (2013) emissions. The latitudinal distribution in the upper troposphere in models and observations is therefore investigated in more detail in the next section.

### Upper tropospheric latitudinal gradients

The input of organic bromine into the stratosphere is crucial in understanding the stratospheric bromine budget and, therefore, also in determining the amount of inorganic bromine available for catalytic reactions involved in ozone depletion. For air masses in the stratosphere above about 400 K, it is generally assumed that the input is nearly exclusively through the tropical tropopause. For the lowermost stratosphere, however, input via the extratropical tropopause is also expected to play an important role (e.g. Holton et al., 1995; Gettelman et al., 2011). Therefore, we compare the observed mole fractions of the

brominated VSLS in the upper troposphere with those determined from the different model setups, in order to investigate if the models are able to represent the latitudinal gradient in upper tropospheric mole fractions. For this purpose, the model data have been averaged in an interval of 10 K below the climatological (TOMCAT), or respectively modelled (EMAC), tropopause. The results are shown for the two main bromine VSLS, CH$_2$Br$_2$ and CHBr$_3$, as well as for total VSLS bromine in Figure 13 for the two campaign periods in comparison to observations. Note that for the scenarios by Liang et al. (2010) and

Ziska et al. (2013), no estimates of emissions for the mixed bromochlorocarbons are available; instead, we have used the model results based on the Ordonez et al. (2012) emissions for the calculation of total VSLS bromine.

During the two campaigns in late summer to fall (TACTS and WISE), all model setups show a decrease of CH$_2$Br$_2$ mixing ratios with latitude. Although, the latitudinal gradients are much steeper when the scenarios by Warwick et al. (2006) and Ziska et al. (2013) are used, which is due to overestimated values at low latitudes. This is in good agreement with findings by

Hossaini et al. (2013), who showed that TOMCAT using the Warwick et al. (2006) emission scenario significantly overestimated HIAPER Pole-to-Pole Observations (HIPPO) in Northern Hemisphere mid latitudes. An increase in observed mixing ratios with latitude was found, especially during the winter PGS campaign, which is presumably related to the increase in atmospheric lifetime of compounds in the cold and dark high latitude tropopause region during winter. This feature is qualitatively reproduced by the TOMCAT simulations with Liang et al. (2010) and Ordonez et al. (2012) scenarios, but not

for the Ziska et al. (2013) and Warwick et al. (2006) scenario based results, which show a moderate decrease and no latitudinal gradient. This feature is consistent with emissions in these two scenarios being more strongly biased towards the tropics.

For CHBr$_3$, the observations show an increase with latitude, especially during the PGS campaign. The late summer to fall data from TACTS and WISE show a less clear picture, with an increase between the subtropics and mid latitudes but a decrease towards high latitudes. This general tendency during the wintertime is reproduced by the TOMCAT model using all emission

scenarios. Nontheless, the gradient in the EMAC model results with the Warwick et al. (2006) emissions is reversed, which is mainly caused by the extremely high tropical mixing ratios, also evident from the latitude altitude cross sections shown before. We also note that the sub-tropical values based on the Ziska et al. (2013) emissions are lower than the observations.

The results for total bromine, including the three mixed bromochlorocarbons, can be mainly understood as a combination of the behavior of CH$_2$Br$_2$ and CHBr$_3$. In the case of the TOMCAT model with Ziska et al. (2013) emissions, a certain

compensation is observed, i.e. total bromine is better reproduced than each compound by itself. This is due to an overestimation of CH$_2$Br$_2$, especially at low latitudes and an underestimation of CHBr$_3$. Total bromine from VSLS in the EMAC model using the Warwick et al. (2006) emissions is very different from the observations. It shows nearly constant values with latitude



during northern hemispheric winter (PGS) and a strong decrease during the late summer to fall period of the TACTS and WISE campaigns. Most importantly, the overall levels, especially in the low latitudes, are much higher than our observations and also much higher than the tropical observations compiled in the WMO report (Engel and Rigby, 2018). This will result in too much VSLS bromine being simulated in the stratosphere, and therefore also in a misrepresentation of the input to the lowermost
stratosphere via the different pathways.

### 5. Implications for stratospheric inorganic bromine

As shown in the previous Section, significant discrepancies exist between the various combinations of models and emission scenarios with respect to our observations, both around the tropopause and in the lower stratosphere. In this Section we will discuss the possible implications for inorganic bromine in the lower and lowermost stratosphere. Note that this discussion only
focuses on the input of bromine in the form of organic source gases (so called source gas injection, SGI, (see e.g. Engel and Rigby, 2018)) from VSLS. The input of bromine into the stratosphere in the inorganic form (product gas injection, PGI) is expected to add more bromine: however, this cannot be investigated with the source gas measurements presented here. Here, we focus on assessing what the different mixing ratios of bromine source gases at the tropical and extratropical tropopause in both observations and in model results imply for the total bromine and inorganic bromine content of the lower and lowermost
stratosphere. Inorganic bromine is of key importance, as this is the form of bromine which can influence ozone through e.g. catalytic ozone depletion cycles.

We have shown in Sections 3 and 4 that the organic bromine around the tropopause shows significant variability and also latitudinal gradients and very significant differences between the different model setups and observations are found. As mentioned in the introduction, the air in the extratropical lower and lowermost stratosphere is influenced by both transport
through the tropical and extratropical tropopause. Several authors have attempted to quantify the relative fractions of air masses from the different source regions based on tracer measurements (e.g. Hoor et al., 2005;Boenisch et al., 2009;Ray et al., 1999;Werner et al., 2010). No studies on mass fractions are available for the campaigns discussed here, so we will rely on previous studies for these fractions. The differences in $Br_y$ discussed here should thus be taken as a sensitivity study and the values derived below can only be considered to be estimates showing to which order of magnitude the inorganic bromine may
differ between different model setups and observations. In general, air masses close to the extratropical tropopause will be mainly of extratropical origin, while air masses near 400 K will almost be entirely of tropical origin. As a simplified approach, we have therefore chosen to assume that at the extratropical tropopause ($\Delta\theta = 0$), the extratropical fraction is 100% and that this fraction decreases linearly to 0% at 100 K above the tropopause. The organic bromine species transported into the stratosphere are chemically or photochemically depleted and the bromine is transferred to the inorganic form. The total bromine
content from VSLS in an air parcel in the lowermost stratosphere at $\Delta\theta$ above the tropopause, $Br_{tot}(\Delta\theta)$, is thus the sum of organic, $Br_{org}(\Delta\theta)$, and inorganic, $Br_{inorg}(\Delta\theta)$, bromine. Inorganic bromine is usually referred to as $Br_y$.

$$Br_{tot}(\Delta\theta) = Br_{inorg}(\Delta\theta) + Br_{org}(\Delta\theta) = Br_y(\Delta\theta) + Br_{org}(\Delta\theta) \qquad (1)$$

The total bromine can also be described by summing up the organic bromine transported to the stratosphere via input through the tropical and extratropical tropopause.

$$Br_{tot}(\Delta\theta) = f^{ex-trop}(\Delta\theta) * Br_{org}^{ex-trop}(0) + f^{trop}(\Delta\theta) * Br_{org}^{trop}(0) \qquad (2)$$

where $f^{ex-trop}$ and $f^{trop}$ are the fractions of air of extratropical and of tropical origin, respectively, and $Br_{org}^{ex-trop}(0)$ and $Br_{org}^{trop}(0)$ are the total organic VSLS bromine contents in air at the tropical, respectively extratropical (40-60°N) tropopause,





i.e. at $\Delta\theta = 0$. For observations only, the extratropical $Br_{org}^{ex-trop}(0)$ is available from our HALO aircraft campaigns. $Br_{org}^{trop}(0)$ for the observations is therefore taken from observations at the tropical tropopause compiled in the 2018 WMO Ozone assessment (Engel and Rigby, 2018). For the different model set-ups $Br_{org}^{ex-trop}(0)$ and $Br_{org}^{trop}(0)$ are derived from the global model fields. For the tropical values, the model output has been averaged between 10°S and 10°N in a potential

temperature range from 365 to 375 K, in a similar way as used for the observations (Engel and Rigby, 2018). Extratropical values have been derived by averaging the models, respective observations, in a range of 10 K below the tropopause. In order to be consistent between models and observations, extra-tropical reference values are taken as the values during the time of the campaign, while the tropical tropopause values are taken as seasonal mean.

Due to mass conservation, the sum of $f^{ex-trop}$ and $f^{trop}$ must be unity, so we can rewrite equation (2) to yield

$$Br_{tot}(\Delta\theta) = f^{ex-trop}(\Delta\theta) * Br_{org}^{ex-trop}(0) + \left(1 - f^{ex-trop}(\Delta\theta)\right) * Br_{org}^{trop}(0) \qquad (3)$$

If we assume that $f^{ex-trop}$ increases linearly from 1 at $\Delta\theta = 0$ K to 0 at $\Delta\theta = 100$ K, the total bromine from VSLS SGI can
be derived and the inorganic bromine $Br_y(\Delta\theta)$ is then calculated by combining (1) and (3)

$$Br_y(\Delta\theta) = \left(f^{ex-trop}(\Delta\theta) * Br_{org}^{ex-trop}(0) + \left(1 - f^{ex-trop}(\Delta\theta)\right) * Br_{org}^{trop}(0)\right) - Br_{org}(\Delta\theta) \qquad (4)$$

where $Br_{org}(\Delta\theta)$ is the organic bromine measured, respectively simulated at $\Delta\theta$ above the tropopause.

Figure 14 compares the vertical profiles of total and inorganic bromine derived in this way from the observations and the different model set-ups for the PGS campaign and the combined WISE-TACTS dataset. The values of $Br_{org}^{ex-trop}(0)$ and $Br_{org}^{trop}(0)$ used for the models, respectively the observations, are shown in Table 5.

Due to the nature of the setup for the calculation of the SGI contribution to $Br_y$ described above, both model- and observation-derived $Br_y$ is close to zero at the extratropical tropopause. The assumed fractional contribution of tropical air increases with
altitude and thus the amount of organic bromine assumed at the tropical tropopause becomes more important in the calculation of total bromine and thus also in $Br_y$. Overall, all model setups capture $Br_y$ from $CH_2Br_2$ rather well. For all campaigns, the $Br_y$ estimate from the observations is smaller than the model calculations above about 60 K above the tropopause and larger below this level. The larger $Br_y$ derived in the model calculations above 60 K is caused by the higher total bromine values from $CH_2Br_2$, which are caused by the higher $CH_2Br_2$ levels at the tropical tropopause in comparison to the observations. For
the late summer/early fall campaigns this difference is largest for the TOMCAT model with the Ziska et al. (2013) emissions and the EMAC model with the Warwick et al. (2006) emissions, consistent with these two model setups having the largest $CH_2Br_2$ values at the tropical tropopause (1.13 and 1.28 ppt, see Table 5). In the lower part the discrepancy is more due to higher simulated $CH_2Br_2$ in the lowermost stratosphere than found in the observations. Using the emission scenarios by Liang et al. (2010) and Ordonez et al. (2012), the differences are usually below 0.3 ppt of $Br_y$, corresponding to a MAPD of less than
35   40%.

Much larger variations are found in the amount of $Br_y$ derived from $CHBr_3$. As can be seen from Figure 7, the remaining organic bromine in the form of $CHBr_3$ is very small for all three setups using the TOMCAT model and the observations already at about 30 to 40 K above the tropopause. The $Br_y$ from $CHBr_3$ (solid lines in Figures 14) is thus close to the total bromine in form of $CHBr_3$ (dotted lines in Figure 14). In contrast, EMAC results using the Warwick et al. (2006) emissions still show
significant amounts of $CHBr_3$ in the organic form even at 50 K above the tropopause and above. For the EMAC setup, the $Br_y$ derived from $CHBr_3$ is thus influenced by both the assumed input and the remaining organic $CHBr_3$ in the stratosphere. However, the tropical input of $CHBr_3$ in the EMAC model using the Warwick et al. (2006) emissions is very large (0.84 ppt,





corresponding to about 2.5 ppt of bromine). Therefore, despite the fact that EMAC still shows significant remaining CHBr$_3$ rather deep into the lowermost stratosphere, this model setup significantly overestimates the amount of Br$_y$ due to CHBr$_3$ in comparison to the observations, with differences of about 1.5 ppt of Br$_y$ at about 100 K above the tropopause, which is about factot of 3 higher than the value derived from the observations. Br$_y$ from CHBr$_3$ in the different emission scenarios used in

TOMCAT is mainly determined by the amount of CHBr$_3$ reaching the stratosphere, and especially for regions with $\Delta\theta$ above 50 K by the tropical input. As the TOMCAT model with the Ziska et al. (2013) emissions underestimates these tropical tropopause values, it shows too little Br$_y$ from CHBr$_3$ throughout the stratosphere. In contrast, the tropical tropopause values of CHBr$_3$ from the Ordonez et al. (2012) and Liang et al. (2010) scenarios are in better agreement with the observations presented here and thus Br$_y$ estimates at 100 K above the tropopause are in good agreement with the observation-based

estimates.

The total Br$_y$ from VSLS SGI can be understood mainly as an addition of the contributions of CH$_2$Br$_2$ and CHBr$_3$, as these are responsible for about 80% of total VSLS bromine. As the differences are largest for CHBr$_3$, the differences in total Br$_y$ from VSLS SGI is dominated by the differences in CHBr$_3$. Interestingly, while the Ziska et al. (2013) emissions in TOMCAT showed some significant differences, in particular of CHBr$_3$ at the tropopause, the differences in total Br$_y$ are not as large. The

underestimation of Br$_y$ from CHBr$_3$ is partly compensated by an overestimation of Br$_y$ from CH$_2$Br$_2$. The EMAC model with the Warwick et al. (2006) emissions overestimates Br$_y$ from both CH$_2$Br$_2$ and CHBr$_3$, so that in total a difference in Br$_y$ of more than 2 ppt is derived, corresponding to an overestimation by a factor of more than 2 with respect to observation derived values. This difference is expected to have e a significant effect on ozone chemistry in the lower stratosphere.

The Br$_y$ values derived in the approach described above depend on the assumed input values but also on the assumed fractional

contribution of air from the tropics and the extratropics. In order to test the sensitivity of the results on the assumed fractions, we have varied the fractional input. Figure 15 shows the Br$_y$ derived from CH$_2$Br$_2$ and CHBr$_3$ for the PGS campaign at 40K above the tropopause, as a function of the assumed fractional contribution from the extratropical source region ($f^{ex-trop}$); the tropical fraction $f^{trop}$ is then always 1- $f^{ex-trop}$). While the differences are not very large for CH$_2$Br$_2$, which shows a much less pronounced latitudinal gradient, differences for CHBr$_3$ can be very large. In particular for the EMAC model with the

Warwick et al. (2006) scenario, where the dependency of Br$_y$ on the fractional input behaves in an opposite way to the CHBr$_3$ observations and the other model-emission scenario combinations. This shows that for the calculation of Br$_y$ in the lowermost stratosphere from observations, it is necessary to have a good knowledge on the relative contributions and that for models it is necessary to have a realistic representation not only of chemistry but also of transport in the lowermost stratosphere.

## 6. Summary and outlook

We present a large dataset of in-situ observations of five brominated VSLS with the GhOST-MS instrument in the UTLS region using the HALO aircraft. We have used data from the three HALO missions: TACTS, WISE and PGS. Data are presented in tropopause relative co-ordinates, i.e. the difference in potential temperature relative to the dynamical tropopause, defined by the value of 2 PVU. Stratospheric data are sorted by equivalent latitude, while we have used normal latitude for tropospheric data. We have shown systematic variabilities with latitude, altitude and season. The shortest-lived VSLS mixing

ratios decrease fastest with altitude. During polar winter, vertical gradients are larger than during late summer to early fall, which is line with the well-known diabatic descent of stratospheric air during polar winter. An important aspect of the observed distributions is that CHBr$_3$ mixing ratios at the extratropical tropopause are systematically higher than at the tropical tropopause. A similar feature is found for CH$_2$Br$_2$, although the latitudinal gradient is less pronounced than in the case of CHBr$_3$. The increase of VSLS mole fractions is especially clear during Northern Hemisphere winter, when lifetimes become

very long in high latitudes.





We have further compared the observed distributions with a range of modelled distributions from TOMCAT and EMAC, run with different global emission scenarios. The features of the observed distribution are partly reproduced by the model calculations, with large differences produced by the different emissions. Overall, for $CH_2Br_2$, much better agreement between observations and model outputs is found for simulations using the emission scenarios by Liang et al. (2010) and Ordoñez et

al. (2012), which have lower overall emissions than the scenarios by Ziska et al. (2013) and Warwick et al. (2006). This is in agreement with a downward revision of the best estimate of global $CH_2Br_2$ emissions recently proposed (Engel and Rigby, 2018). In the case of $CHBr_3$, the use of the emission scenario by Ziska et al. (2013), which has the lowest global emissions, results in too low mixing ratios at the tropical tropopause and also at the extratropical tropopause. The use of the emission scenario by Warwick et al. (2006) results in strongly elevated mixing ratios of $CHBr_3$ at the tropical tropopause and a reversed

latitudinal gradient at the tropopause in comparison to the observations. These findings are in good agreement with previous comparisons of the different emission scenarios (Hossaini et al., 2013;Hossaini et al., 2016) for $CH_2Br_2$. For $CHBr_3$, Hossaini et al. (2016) found that the lower emissions in the Ziska et al. (2013) scenario generally gave best agreement with ground based observations in the tropics. However, we find that the tropopause values using this scenario are too low, both in the tropics and in the extratropics. In a recent paper, Fiehn et al. (2018) discussed that a modified version of the Ziska et al. (2013)

scenario with seasonally varying emissions, yielded significantly higher tropopause values. The Ordonez et al. (2012) scenario, which has higher emissions than the Ziska et al. (2013) scenario, yielded too high mixing ratios of $CHBr_3$ during the winter period. While it is not the main purpose of this paper to evaluate different emission scenarios, it is clear that no scenario is able to capture tropical and extratropical values from our observations. However, it is clear from the comparison with the scenario by Warwick et al. (2006), which restricts emissions to latitudes below 50°, that the sources of these short-lived brominated

compounds are not only in the tropics, but that significant emissions must also occur in higher latitudes. This is consistent with comparison of tropospheric data (see e.g. Fig. 6 in Hossaini et al., 2013). For future improved emission scenarios, more emphasis on the seasonality of the sources might also lead to an improvement.

Air in the lowermost stratosphere is composed of air masses originating from both the tropical and the extratropical upper troposphere. The latitudinal gradient of VSLS will therefore impact the amount of bromine transported into the stratosphere

and thus also the amount of reactive, inorganic bromine ($Br_y$) in the lowermost stratosphere able to contribute to catalytic ozone depletion. The bromine budget in the lower stratosphere will depend on the relative fraction of air from the tropical and extratropical tropopause. The relative contribution of extratropical air will decrease with latitude and should reach zero at about 400 K potential temperature. Using simplified assumptions about the fractional distributions, we have shown that there will be significant differences in stratospheric $Br_y$ depending on the emission scenario, which can be as high as 2 ppt,

corresponding to a difference of a factor 2 relative to observation-derived values. This is expected to have an impact on modelled ozone depletion in the lower stratosphere. Further, as the efficiency of bromine to destroy ozone depends on the amount of available chlorine, it is also likely that modelled temporal trends of ozone will be influenced, even if there are no long-term changes in VSLS bromine. If relative contributions of the different pathways (tropical vs. extratropical air) change, e.g. due to changes in stratospheric circulation, this could further influence ozone due to the different amounts of bromine in

these air masses. As shown in our sensitivity study (Section 5), the assumptions on the relative contribution of the different source regions has a significant impact especially on the $Br_y$ produced from $CHBr_3$ in the lowermost stratosphere.

While the dataset presented here gives a much better picture of the distribution of brominated VSLS in the UTLS region than previously available, there are still considerable gaps in our knowledge of the distribution of these species. Only late summer to fall and winter data have been presented here for the Northern Hemisphere. Spring and early summer are less well covered,

as is the Southern Hemisphere. Southern hemispheric distributions are expected to differ significantly from northern hemispheric distributions, as the main sources of many brominated VSLS are believed to be from coastal ocean regions. Due to the different distribution of oceans, land and coastal areas between the hemispheres, it is not possible to extrapolate northern





hemispheric observations to the Southern Hemisphere. Further, while no signs of increasing emissions of natural brominated VSLS have been observed so far, such an increase is possible in a changing climate and needs to be monitored.

## 7. Acknowledgements

The work of University Frankfurt has been funded through several projects by the German Science foundation for the development and operation of GHOST-MS and for the measurement campaigns (EN367/5, EN367/8, EN367/11. EN367/13 and EN367/14). A.E. would like to thank CSIRO in Aspendale/Australia for a Frohlich Fellowship during which parts of this analysis was performed. Many thanks also to Kieran Stanley for proof reading and improving the manuscript. We would further like to thank the DLR staff, including pilots and ground staff, for the operation of HALO and the support during the

campaigns. The good collaboration with the other groups involved in the HALO campaigns is also acknowledged. We would like to thank Andreas Zahn from KIT Karlruhe for provision of the ozone data in Figure 3. We further thank Jens-Uwe Grooß from FZ Jülich for the calculation of the tropopause and equivalent latitude for the HALO campaigns and Michael Sprenger from ETH Zürich for the provision of the climatological dynamical tropopause from ERA-Interim data. R.H. is supported by a NERC Independent Research Fellowship (NE/N014375/1). The EMAC simulations have been performed at the German

Climate Computing Centre (DKRZ) through support from the Bundesministerium für Bildung und Forschung (BMBF). DKRZ and its scientific steering committee are gratefully acknowledged for providing the HPC and data archiving resources for this consortial project ESCiMo (Earth System Chemistry integrated Modelling).

## 8. Author Contribution

T.K., F.O., H.B. and A.E. were involved in developing the GhOST instrument, operating it in the field during the missions,

data evaluation and interpretation. F.L., M.H. and T.S. were also involved in the operation, evaluation and interpretation. N.S., A.R. and C.H. were involved in the evaluation and interpretation. R.H., P.G. and P.J. have provided model data and also participated in the discussion of the data and the comparisons. A.E and T.K. have mainly written the manuscript. All co-authors were involved in the discussion and iterations of the manuscript.

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



## 10. Graphics and Tables

*Table 1:* Brominated species measured with Gas Chromatograph for Observational Studies using Tracers-Mass spectrometer (GhOST-MS) during three High Altitude and Long Range Research Aircraft campaigns, described in Table 2. Tropospheric mole fractions (parts per trillion, ppt; $10^{-12}$) of the halons are taken from table 1-1 in (Engel and Rigby, 2018) and from table 1-7 for the bromocarbons (marine boundary layer values). Lifetimes of bromocarbons are local lifetimes for upper tropospheric conditions (10 km altitude, 25-60°N) from table 1-5 in (Carpenter and Reimann, 2014) and global /stratospheric lifetimes are from table A-1 in WMO 2018 (Burkholder, 2018). Reproducibilities and detection limits of GhOST have been determined during the WISE campaigns. n.a. means not applicable.

| Name | Formula | troposph. Mole fraction [ppt] | GhOST-MS characteristics | | typical lifetime | | |
|---|---|---|---|---|---|---|---|
| | | | Reproduca-bility [%] | Detection limit [ppq] | fall [days] | winter [days] | Global/ stratospheric [years/years] |
| Halon 1301 | $CF_3Br$ | 3.36 | 1 | 50 | n.a. | n.a. | 72/73.5 |
| Halon 1211 | $CBrClF_2$ | 3.59 | 0.5 | 6 | n.a. | n.a. | 16/41 |
| Halon 1202 | $CBr_2F_2$ | 0.014 | 7.6 | 6 | n.a. | n.a. | 2.5 / 36 |
| Halon 2402 | $CBrF_2CBrF_2$ | 0.41 | 1.5 | 7 | n.a. | n.a. | 28/41 |
| Dibromomethane | $CH_2Br_2$ | 0.9 | 0.7 | 11 | 405 | 890 | n.a. |
| Tribromomethane | $CHBr_3$ | 1.2 | 2.2 | 85 | 44 | 88 | n.a. |
| Bromochloromethane | $CH_2BrCl$ | 0.1 | 9.2 | 130 | 470 | 1050 | n.a. |
| Dichlorobromomethane | $CHBrCl_2$ | 0.3 | 3.4 | 2 | 124 | 250 | n.a. |
| Dibromochloromethane | $CHBr_2Cl$ | 0.3 | 2.2 | 2 | 85 | 182 | n.a. |

*Table 2:* Brief description of measurement campaigns with the High Altitude and Long Range Research aircraft used for this study.

| Name | Time period | Campaign base | brief description |
|---|---|---|---|
| **TACTS**, Transport and Composition in the Upper Troposphere/Lowermost Stratosphere | late August 2012-September 2012 | Oberpfaffenhofen/ Germany and Sal/Cape Verde | Cover changes in UTLS chemical composition during the transition from summer to fall |
| **WISE**, Wave driven isentropic exchange | September - October 2017 | Shannon/Ireland | Study Troposphere-Stratosphere Exchange in mid latitudes |
| **PGS**, POLSTRACC, GW-Lcycle, SALSA* | December 2015 - March 2016 | Kiruna/Sweden | Study the polar UTLS during winter, including the effect of chemical ozone depletion. |

\* **PGS** is a synthesis of three measurement campaigns: **POLSTRACC** (The Polar Stratosphere in a Changing Climate), **GW-LCYCLE** (Investigation of the Life cycle of gravity waves) and **SALSA** (Seasonality of Air mass transport and origin in the Lowermost Stratosphere).





*Table 3:* Averaged mole fractions (parts per trillion, ppt; $10^{-12}$) and vertical gradients of brominated very short lived substances from the combined Wave driven isentropic exchange (WISE) and Transport and Composition in the Upper Troposphere/Lowermost Stratosphere (TACTS) data set, representative for 40-60°N during late summer to early fall (data from late August to October). Data have been averaged using potential temperature and potential temperature difference to the tropopause as vertical profiles coordinates. Tropopause values are from the 10 K bin below the dynamical tropopause (see text for details). The average potential temperature of the tropopause during the WISE and TACTS campaigns has been calculated from the European Centre for Medium Weather Forecast data at the locations of our measurements.

| WISE and TACTS | Potential Temperature | | | | $\Delta\theta$ | | | |
|---|---|---|---|---|---|---|---|---|
| | Mole fraction [ppt] | | Gradient | 10 K bin stdev. (TP − TP + 40K) | Mole fraction [ppt] | | Gradient | 10 K bin stdev. (TP − TP + 40K) |
| | TP | TP+(30-40 K) | [%/K] | [ppt] | TP | TP+(30-40 K) | [%/K] | [ppt] |
| $CH_2Br_2$ | 0.80 | 0.58 | 0.70 | 0.12 | 0.77 | 0.55 | 0.72 | 0.09 |
| $CHBr_3$ | 0.47 | 0.14 | 1.77 | 0.20 | 0.40 | 0.09 | 1.92 | 0.10 |
| $CH_2BrCl$ | 0.23 | 0.13 | 1.14 | 0.08 | 0.19 | 0.14 | 0.73 | 0.07 |
| $CHBrCl_2$ | 0.16 | 0.12 | 0.64 | 0.03 | 0.15 | 0.11 | 0.71 | 0.02 |
| $CHBr_2Cl$ | 0.12 | 0.07 | 1.13 | 0.04 | 0.12 | 0.06 | 1.24 | 0.02 |
| total Br | 3.64 | 1.89 | 1.20 | 0.83 | 3.40 | 1.76 | 1.20 | 0.51 |

*Table 4:* Averaged mole fractions (parts per trillion, ppt; $10^{-12}$) and vertical gradients of brominated VSLS during the PGS campaign. Data have been averaged using potential temperature and potential temperature difference to the tropopause as vertical profiles coordinates. Tropopause values are from the 10 K bin below the dynamical tropopause (see text for details). The average potential temperature of the tropopause during the PGS campaign has been calculated from ECMWF data at the locations of our measurements.

| PGS | Potential Temperature | | | | Delta Theta | | | |
|---|---|---|---|---|---|---|---|---|
| | Mole fraction [ppt] | | Gradient | 10 K bin stdev | Mole fraction [ppt] | | Gradient | 10 K bin stdev |
| | TP | TP + 40 K | [%/K] | [ppt] | TP | TP + 40 K | [%/K] | [ppt] |
| $CH_2Br_2$ | 1.08 | 0.50 | 1.34 | 0.18 | 1.09 | 0.53 | 1.28 | 0.11 |
| $CHBr_3$ | 0.66 | 0.07 | 2.22 | 0.26 | 0.75 | 0.07 | 2.26 | 0.13 |
| $CH_2BrCl$ | 0.25 | 0.13 | 1.16 | 0.05 | 0.26 | 0.14 | 1.14 | 0.03 |
| $CHBrCl_2$ | 0.20 | 0.09 | 1.35 | 0.03 | 0.20 | 0.10 | 1.29 | 0.02 |
| $CHBr_2Cl$ | 0.16 | 0.04 | 1.89 | 0.04 | 0.16 | 0.04 | 1.86 | 0.03 |
| total Br | 4.91 | 1.53 | 1.72 | 1.28 | 5.20 | 1.60 | 1.73 | 0.70 |





***Table 5:*** Values of organic VSLS bromine in air at the tropical, respectively extratropical (40-60°N) tropopause ($Br_{org}^{ex-trop}$ and $Br_{org}^{trop}$) used in the calculation of inorganic bromide ($Br_y$) for the observation (OBS), respectively the models using the emission scenarios of Liang et al. (2010), Ordonez et al. (2012), Ziska et al. (2013) and Warwick et al. (2006). For the Warwick et al. (2006) scenario, the data have been derived from the EMAC model, while for the other scenarios the TOMCAT model has been used. For the Tropics, annual average for the years 2012 to 2016 have been calculated between 10°N and 10°S in a potential temperature range from 365 to 375 K. The tropical values for the observations are from the observations compiled in the 2018 WMO report (Engel and Rigby, 2018) in the tropics between 365 and 375 K potential temperature. All data presented are shown in parts per trillion ($10^{-12}$)

| | Tropics | | | ML WISE/TATS | | | ML PGS | | |
|---|---|---|---|---|---|---|---|---|---|
| | $CH_2Br_2$ | $CHBr_3$ | TOT | $CH_2Br_2$ | $CHBr_3$ | TOT | $CH_2Br_2$ | $CHBr_3$ | TOT |
| OBS | 0.73 | 0.28 | 2.80 | 0.83 | 0.56 | 3.99 | 1.09 | 0.75 | 5.20 |
| LIANG | 0.82 | 0.26 | 3.06 | 0.70 | 0.32 | 2.84 | 0.99 | 1.00 | 5.73 |
| ORDONEZ | 0.91 | 0.28 | 3.30 | 0.79 | 0.44 | 3.27 | 1.10 | 1.21 | 6.58 |
| ZISKA | 1.13 | 0.10 | 3.18 | 0.87 | 0.18 | 2.77 | 1.13 | 0.69 | 5.10 |
| WARWICK | 1.28 | 0.84 | 5.48 | 0.83 | 0.37 | 3.07 | 1.16 | 0.62 | 4.59 |

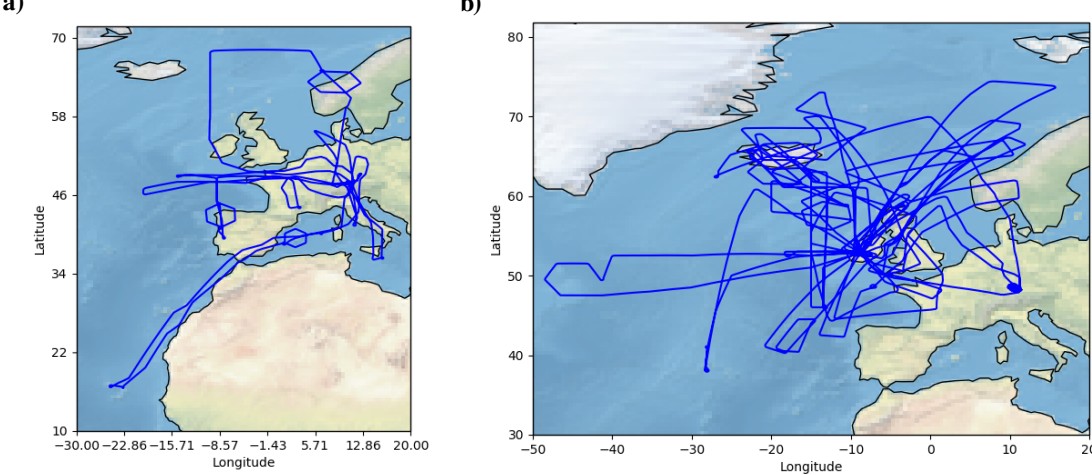

**Figure 1:** Flight tracks of High Altitude and Long Range Research Aircraft during the a) Transport and Composition in the Upper Troposphere/Lowermost Stratosphere (TACTS) campaign (late August and September 2012) and the Wave driven isentropic exchange (WISE) campaign (September/October 2017). The basis of the TACTS campaign was mainly Oberpfaffenhofen (near Munich) in Germany, while the basis of the WISE campaign was Shannon (Ireland).





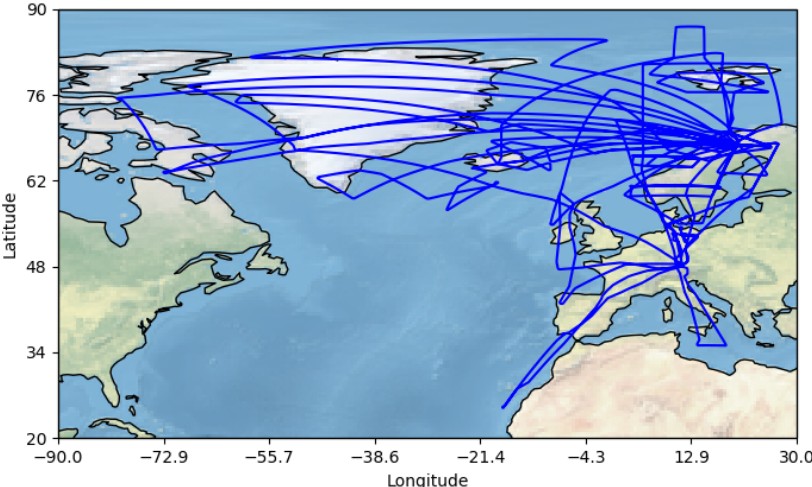

**Figure 2:** Flight tracks of High Altitude and Long Range Research Aircraft during the PGS (Polar Stratosphere in a Changing Climate, Investigation of the Life cycle of gravity waves and Seasonality of Air mass transport and origin in the Lowermost Stratosphere) campaign (December 2015 to April 2016). The basis of the campaign was mainly Kiruna in Northern Sweden.

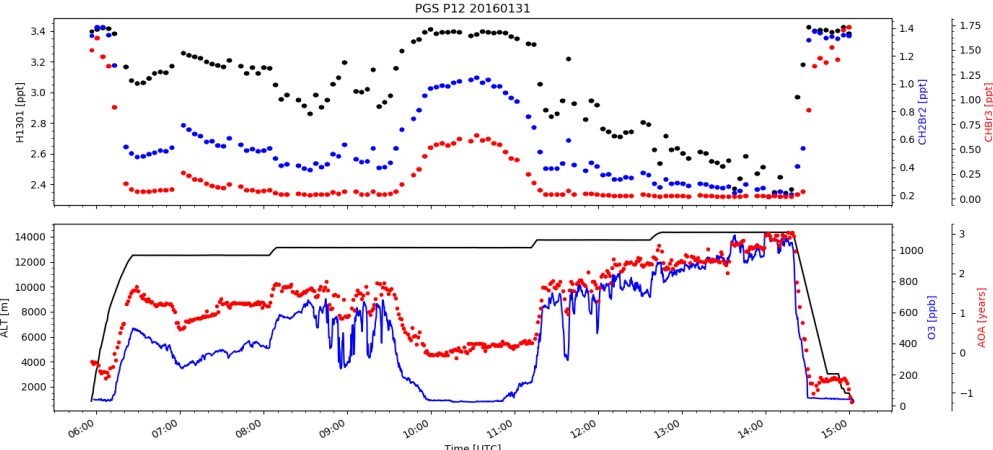

**Figure 3:** Example of data gathered during a single Flight of the High Altitude and Long Range Research Aircraft during the PGS (Polar Stratosphere in a Changing Climate, Investigation of the Life cycle of gravity waves and Seasonality of Air mass transport and origin in the Lowermost Stratosphere) campaign. The flight PGS 12 started on 31 January 2016 from Kiruna in Northern Sweden. The upper panel shows measurements (parts per trillion, ppt; $10^{-12}$) of the long-lived brominated source gas Halon 1301 ($CF_3Br$) and the short-lived source gases CH$_2$Br$_2$ and CHBr3, all measured with GhOST MS. The lower panel shows flight altitude, as well as ozone (parts per billion, ppb; $10^{-9}$; measured by the FAIRO instrument (Zahn et al., 2012) and of mean age of air derived from SF$_6$ measurements from the ECD channel of GhOST-MS (1 minute time resolution, see e.g. (Boenisch et al., 2009) for a description of the measurement technique). An air mass with low ozone and also low mean age of air was observed during the middle of the flight between about 10 and 11 UTC. High mixing ratios of all three source gases are found in this region, as well as during take-off and landing of the aircraft. CHBr$_3$ values are close to detection limit when flying in aged stratospheric air masses, indicating a complete conversion of the bromine to its inorganic form.

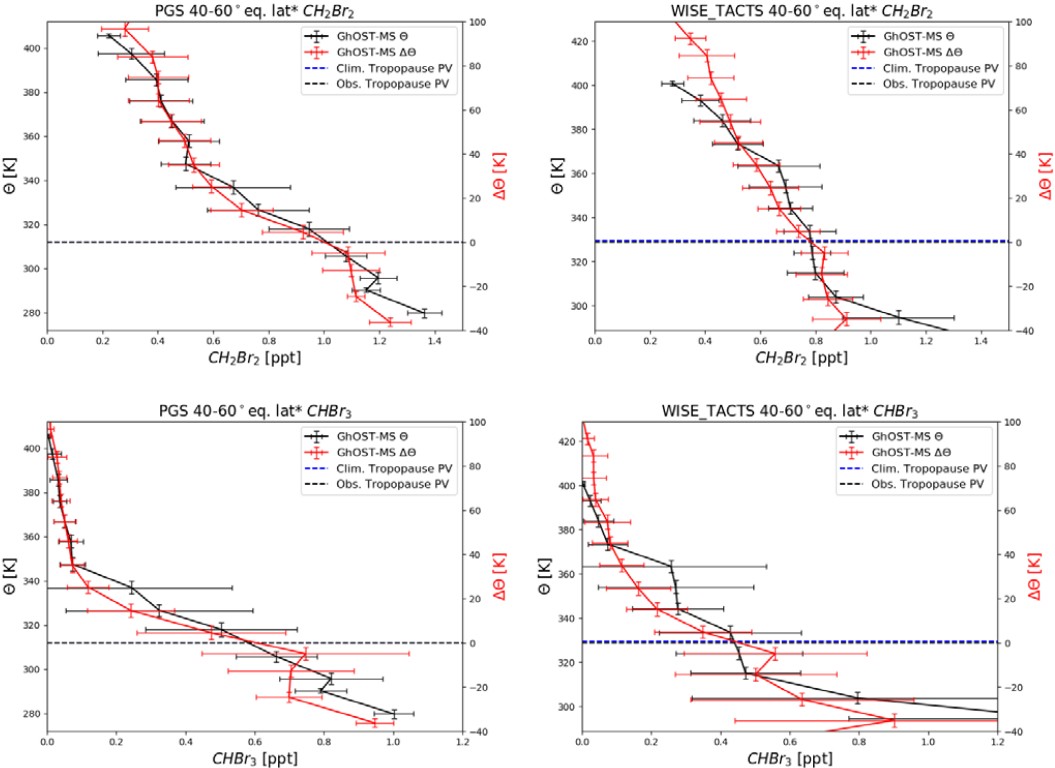

**Figure 4:** Vertical profiles of $CH_2Br_2$ and $CHBr_3$ (parts per trillion, ppt; $10^{-12}$) averaged over 40-60° of equivalent latitude* and all flights during the PGS (Polar Stratosphere in a Changing Climate, Investigation of the Life cycle of gravity waves and Seasonality of Air mass transport and origin in the Lowermost Stratosphere) campaign (left, late December 2015 to March 2016) and from the merged data set from the Transport and Composition in the Upper Troposphere/Lowermost Stratosphere and Wave driven isentropic exchange campaigns (right, representative of late summer to fall). The data are displayed as function of potential temperature and potential temperature above the tropopause. The dotted blue line shows the zonal mean dynamical tropopause derived from ERA Interim during September and October of the respective years in the Northern Hemisphere between 40 and 60° latitude, while the black line is the average dynamical tropopause derived for the times and locations of our observations.



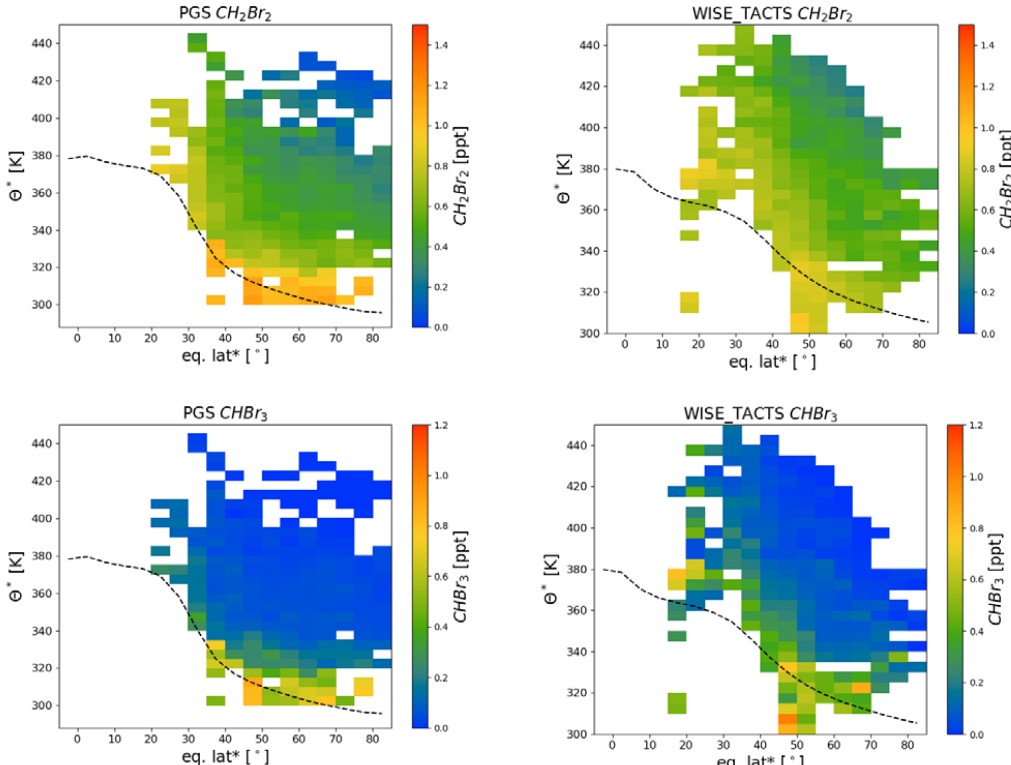

**Figure 5:** Altitude latitude cross sections of $CH_2Br_2$ and $CHBr_3$ (parts per trillion, ppt; $10^{-12}$) compiled from all flights during the during the
5  PGS (Polar Stratosphere in a Changing Climate, Investigation of the Life cycle of gravity waves and Seasonality of Air mass transport and origin in the Lowermost Stratosphere) campaign from late December 2015 to March 2016 (left) and the Transport and Composition in the Upper Troposphere/Lowermost Stratosphere (TACTS) and Wave driven isentropic exchange (WISE) campaigns representative of late summer/early fall conditions (right). The data are displayed as function of $\theta^*$ (see description in Section 2) and equivalent latitude*. The dynamical tropopause (dashed line) has been derived from ERA-Interim reanalysis, providing a climatological mean zonal mean value of
10  the tropopause.



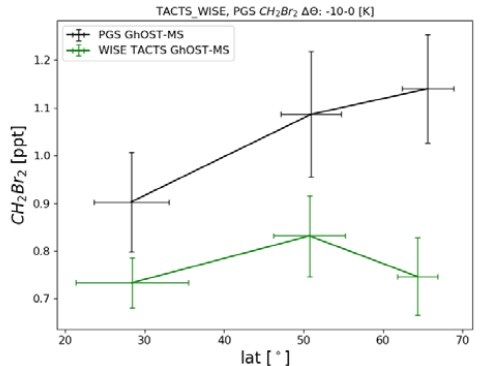

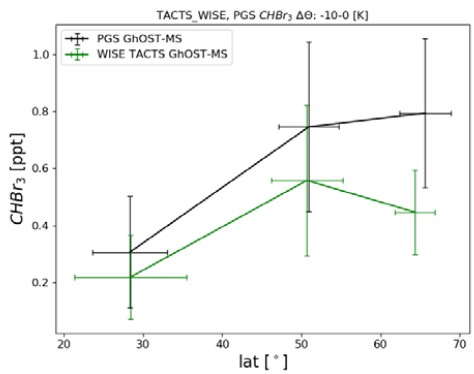

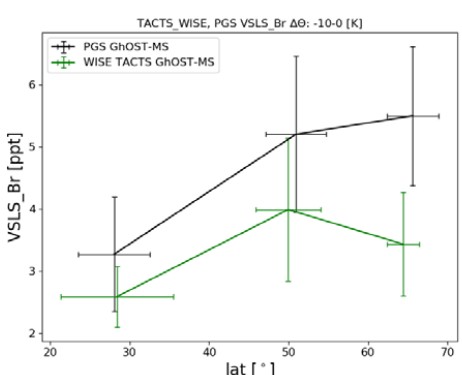

**Figure 6:** Latitudinal cross section of $CH_2Br_2$, $CHBr_3$ and total organic VSLS bromine (parts per trillion, ppt; $10^{-12}$) for all three campaigns, binned by latitude and averaged within 10 K below the local dynamical tropopause.





**Figure 7:** Vertical profiles of $CH_2Br_2$ and $CHBr_3$ and total organic VSLS bromine (parts per trillion, ppt; $10^{-12}$) averaged over 40-60° of
equivalent latitude* and all flights during the PGS (Polar Stratosphere in a Changing Climate, Investigation of the Life cycle of gravity
waves and Seasonality of Air mass transport and origin in the Lowermost Stratosphere) campaign from late December 2015 to March 2016
(left hand side) and from the combined WISE_TACTS (Wave driven isentropic exchange, WISE; and Transport and Composition in the
Upper Troposphere/Lowermost Stratosphere, TACTS) data set, representative of late summer to fall conditions. Also shown are model
results from the Toulouse Off-line Model of Chemistry And Transport (TOMCAT) and ECHAM/MESSy Atmospheric Chemistry (EMAC)
model using different emission scenarios (see text for details). Data from some flight of the TACTS campaign have bene omitted due to
some extremely high values, which are suspected to be a contamination. The data are displayed as function of potential temperature above
the dynamical tropopause. In case no model information on the tropopause altitude was available (TOMCAT), climatological tropopause
values have been used (see text for details).



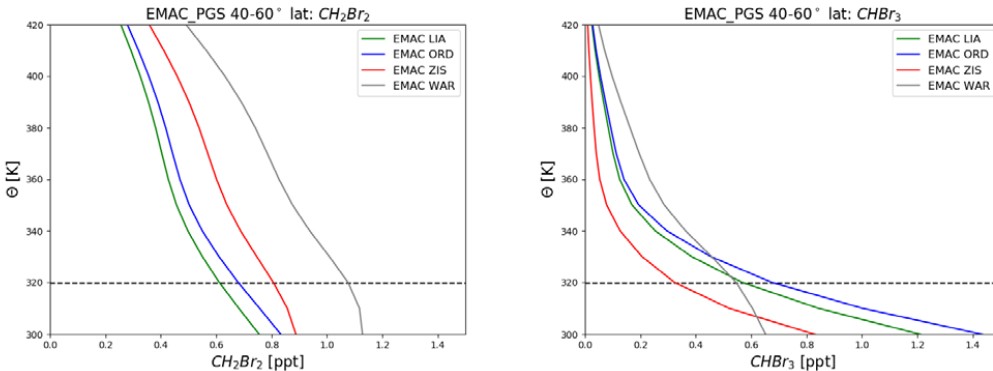

**Figure 8:** Vertical profiles of $CH_2Br_2$ and $CHBr_3$ (parts per trillion, ppt; $10^{-12}$) averaged over 40-60° latitude from four model simulations with the EMAC model using the emission scenarios by (Liang et al., 2010;Warwick et al., 2006;Ordoñez et al., 2012;Ziska et al., 2013). The data have been averaged for January/February and March, i.e. representative of the time period covered by the PGS (Polar Stratosphere in a Changing Climate, Investigation of the Life cycle of gravity waves and Seasonality of Air mass transport and origin in the Lowermost Stratosphere) campaign. The dashed line represents the model tropopause.

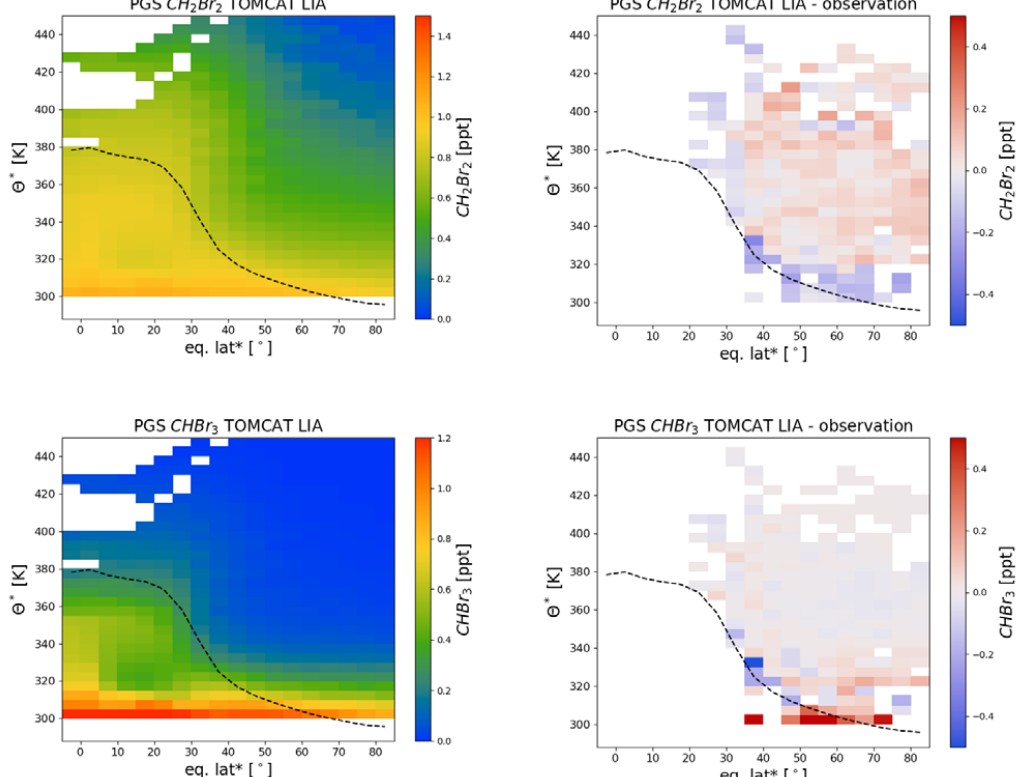

**Figure 9:** Latitude altitude cross section of CH$_2$Br$_2$ and CHBr$_3$ (parts per trillion, ppt; $10^{-12}$) for the Toulouse Off-line Model of Chemistry And Transport (TOMCAT) model using the Liang et al. (2010) emission scenario (left) and differences to the observations (right) for all flights during the PGS (Polar Stratosphere in a Changing Climate, Investigation of the Life cycle of gravity waves and Seasonality of Air mass transport and origin in the Lowermost Stratosphere) campaign from late December 2015 to March 2016. The data are binned using equivalent latitude* and $\theta^*$ as coordinates (see text for details). Also shown in the climatological mean tropopause (see text for details; dashed line).



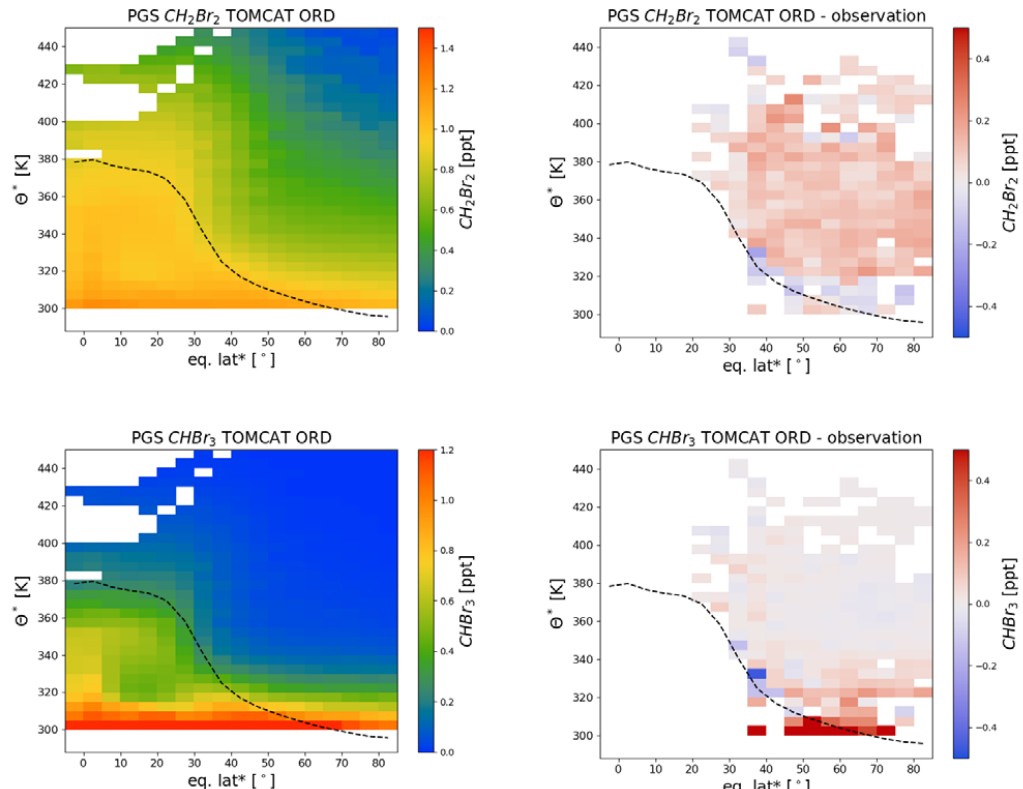

**Figure 10:** Latitude altitude cross section of $CH_2Br_2$ and $CHBr_3$ (parts per trillion, ppt; $10^{-12}$) for the Toulouse Off-line Model of Chemistry And Transport (TOMCAT) model using the Ordonez et al. (2012) emission scenario (left) and differences to the observations (right) for all flights during the PGS (Polar Stratosphere in a Changing Climate, Investigation of the Life cycle of gravity waves and Seasonality of Air mass transport and origin in the Lowermost Stratosphere) campaign from late December 2015 to March 2016. The data are binned using equivalent latitude* and $\theta^*$ as coordinates (see text for details). Also shown in the climatological mean tropopause (see text for details; dashed line).



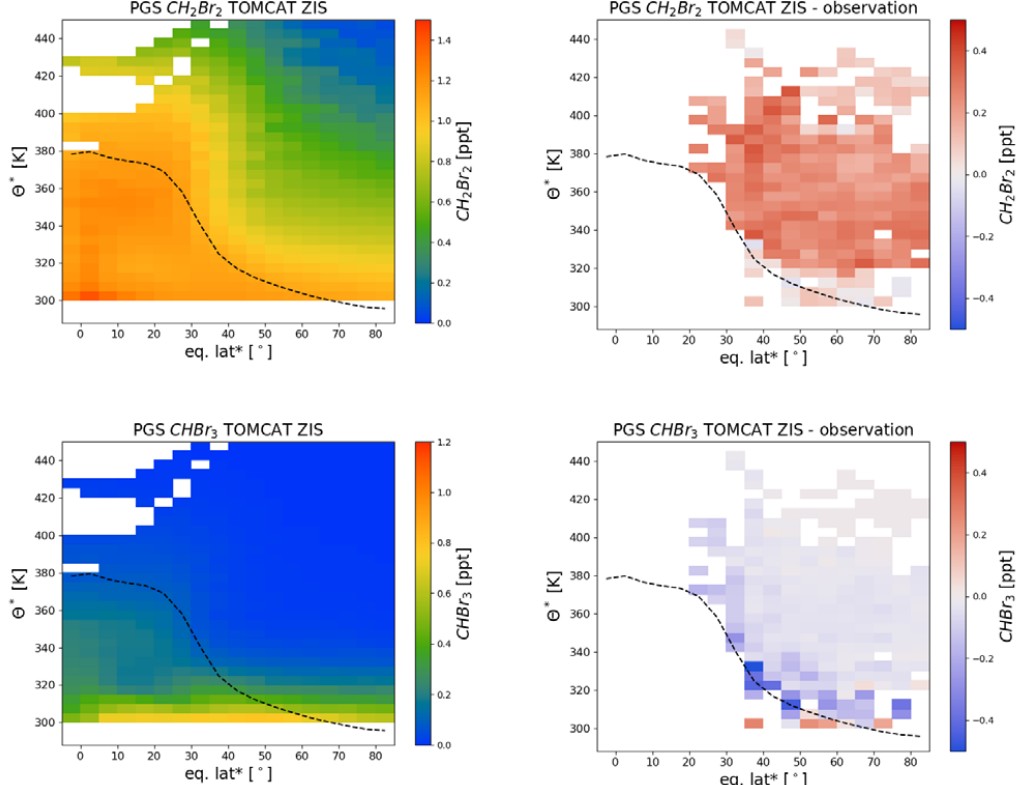

**Figure 11:** Latitude altitude cross section of $CH_2Br_2$ and $CHBr_3$ (parts per trillion, ppt; $10^{-12}$) for the Toulouse Off-line Model of Chemistry
And Transport (TOMCAT) model using the Ziska et al. (2013) emission scenario (left) and differences to the observations (right) for all
flights during the PGS (Polar Stratosphere in a Changing Climate, Investigation of the Life cycle of gravity waves and Seasonality of Air
mass transport and origin in the Lowermost Stratosphere) campaign from late December 2015 to March 2016. The data are binned using
equivalent latitude* and $\theta^*$ as coordinates (see text for details). Also shown in the climatological mean tropopause (see text for details; dashed
line).



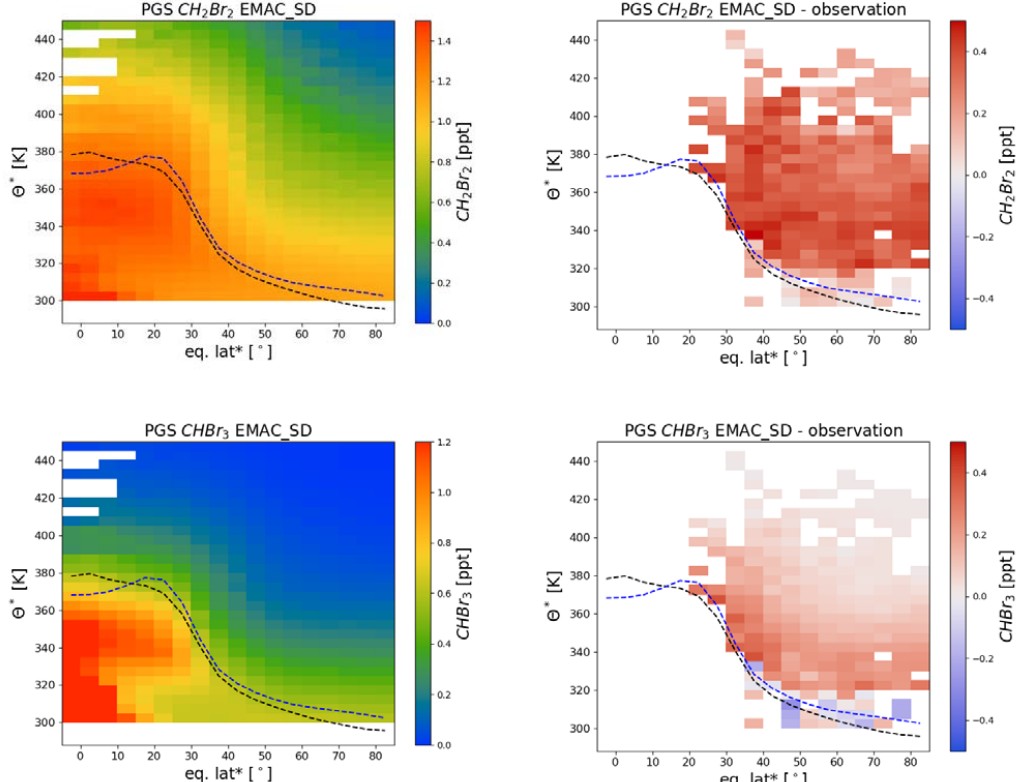

**Figure 12:** Latitude altitude cross section of CH₂Br₂ and CHBr₃ (parts per trillion, ppt; $10^{-12}$) for the ECHAM/MESSy Atmospheric
5   Chemistry (EMAC) model using the Warwick et al. (2006) emission scenario (left) and differences to the observations (right) for all flights
during the PGS (Polar Stratosphere in a Changing Climate, Investigation of the Life cycle of gravity waves and Seasonality of Air mass
transport and origin in the Lowermost Stratosphere) campaign from late December 2015 to March 2016. The data are binned using equivalent
latitude* and $\theta^*$ as coordinates (see text for details). Also shown in the climatological mean tropopause (black dashed line, see text for details)
and the model tropopause (dashed blue line, see text for details).





**Figure 13:** Latitude cross section of tropopause representative values of $CH_2Br_2$, $CHBr_3$ and total organic VSLS bromine (parts per trillion, ppt; $10^{-12}$) for all the measurements from the PGS (Polar Stratosphere in a Changing Climate, Investigation of the Life cycle of gravity waves and Seasonality of Air mass transport and origin in the Lowermost Stratosphere) campaign (left) and WISE_TACTS (Wave driven isentropic exchange, WISE; and Transport and Composition in the Upper Troposphere/Lowermost Stratosphere, TACTS) dataset (right) from observations in comparison to all model emissions scenario combinations. Data are binned by latitude and averaged over 10 K below the tropopause.



**Figure 14** Vertical profiles of $Br_y$ (solid lines; parts per trillion, ppt, $10^{-12}$) and total Bromine (dotted lines; ppt) from $CH_2Br_2$, from $CHBr_3$

5   and from total organic VSLS bromine averaged over 40-60° of equivalent latitude* for the winter PGS (Polar Stratosphere in a Changing Climate, Investigation of the Life cycle of gravity waves and Seasonality of Air mass transport and origin in the Lowermost Stratosphere) campaign (left, late December 2015 to March 2016) and for late summer to early fall period (right, Wave driven isentropic exchange (WISE) and Transport and Composition in the Upper Troposphere/Lowermost Stratosphere (TACTS) campaigns) in comparison to model results from the Toulouse Off-line Model of Chemistry And Transport (TOMCAT) and the ECHAM/MESSy Atmospheric Chemistry (EMAC)

10  model using different emission scenarios (see text for details on calculation of $Br_y$). Total Bromine is calculated from data at the tropical and extratropical tropopause and using assumptions about fractional input from these two source regions (see text for details). The data are displayed as function of potential temperature above the tropopause. In case no model information on the tropopause altitude was available (TOMCAT), climatological tropopause values have been used (see text for details).





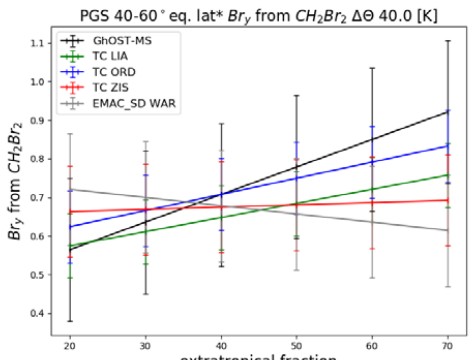
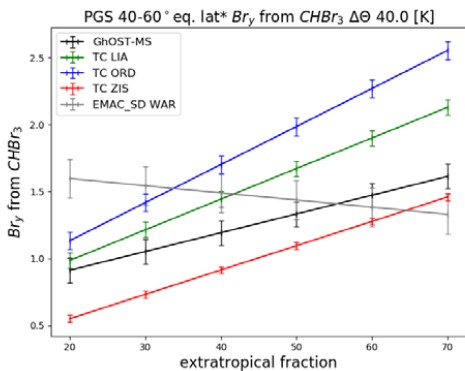

**Figure 15:** Sensitivity of $Br_y$ from $CH_2Br_2$ and $CHBr_3$ (parts per trillion, ppt; $10^{-12}$) at $\Delta\theta$ of 40 K as a function of the fraction of extratropical air for the PGS (Polar Stratosphere in a Changing Climate, Investigation of the Life cycle of gravity waves and Seasonality of Air mass transport and origin in the Lowermost Stratosphere) campaign from January to April 2016 for observations in comparison to the different model calculation.