# Peer review of "Bromine from short–lived source gases in the extratropical Northern Hemisphere UTLS"

_Atmospheric Chemistry and Physics, 2019_

## Referee Comment (RC1) · Anonymous Referee #1 · 21 Oct 2019

**Review ACP-2019-796**

Bromine from short-lived source gases in the Northern Hemisphere UTLS

**1   general comments**

```
evaluating the overall quality of the discussion paper
```

The paper presents novel observational data of VSLS from several aircraft campaigns and compares them with modeling results. It extends on and confirms previous findings. Hence, the title does reflect the contents of the paper.

Some scientific methods and assumptions need more thorough outline and proper discussion.

The results are sufficient to support the interpretations and conclusions, but not all results can be repeated based on the information given, e.g. the derivation of the tropopause for the campaigns labeled as "private conversation".

The authors give proper credit to related work and indicate their own contribution, but they could make it more clear from the start, what the current consensus regarding the emission scenarios is.

The abstract provides a complete summary, but it does not deliver regarding the WHY unit, e.g. the context and importance of the measurement. It gives the impression of a paper focused on observational data, whereas 2/3 of the figures and text are related to model evaluation. If the point of the paper is to evaluate existing emission scenarios through a new set of data, this should be made more clear. The overall presentation is well structured and clear.

The language is fluent. The authors should be more cautious with the usage of the term "significant". Some parts of the paper need editing. In particular, the manuscript does not follow the ACP guidelines in several points:

- Subsections should be consecutively numbered.

- Figure captions, figures, and tables:

    - Usage of full campaign names renders the captions imprecise and utterly unpleasant to read.
    - Poor choice of colors (red, green, blue, black, grey) within the line plots (vector graphics!) and the tick labels' font size make the figures hard to read (Fig. 7, Figs. 13-15).
    - The usage of "[ ]" around units in plots is depreciated (→ https://www.bipm. org/en/publications/si-brochure/section5-3.html)

- Some white spaces seem odd.

- Equations are not properly set, e.g. usage of "*" as indicator for multiplication.

The number and quality of references are appropriate.

**2 specific comments**

`individual scientific questions/issues`

- P4L33: *"It is clearly visible that the halocarbons correlate [...]"* Can you quantify this?

- P5L40-P6L1: *"Only bins which contain at least five data points [...]"* What criterion led to this choice? (See comment to Fig. 4 below.)

- P6L21-22: *"[...] the variability averaged over the four lowest stratospheric bins was always lower when using $\Delta\theta$ as a coordinate."* Can you elaborate on this? Why does the variability in the four lowermost stratospheric bins change in response to the transformation of coordinates $\Delta\theta$ (relative to the tropopause)? Am I right to assume that this is due to bins with mixed tropospheric/stratospheric data? If that is the case, "four lowest stratospheric bins" is misleading. Please elaborate on this.

- P7L5-7: *"The data have been binned in $5°$ latitude and 5 K intervals of potential temperature. As expected, the distributions closely follow the tropopause (indicated by the dashed line), with values decreasing with distance to the tropopause and also with increasing equivalent latitude."* In the previous section the authors

used *"[o]nly bins which contain at least five data points [...]"* in a much coarser binning. From the sampling frequency, one ought to assume that the shown averaged VSLS concentrations per bin are based on similarly sized numbers of entries, but this neither emerges from the text nor the figures (Fig. 5, Figs. 9-12). Furthermore, earlier in the manuscript the authors find that $\Delta\theta$ is the coordinate of choice for this study, but they don't use it in these figures. If it was possible to show the data in relative coordinates, this would strengthen their point (*"closely follow the tropopause"*). The authors should elaborate on this.

- P7L7-9: *"The distributions observed during the WISE and the TACTS campaigns show significant amounts of $CH_2Br_2$ [...]"* What purpose does "significant" serve in this context? Elaborate on the actual significance or drop the word.

- P7L10-16: *"[. . .] the most stratospheric air [...]"* and *"[. . .] very high mean age of air [...]"* If the authors' point is to state that the amount of VSLS in the stratosphere is a function of its residence time there, they should make this more clear.

- P7L28-29: *"[. . .] are significantly larger [...]"* Can you quantify this? Else drop the term "significantly".

- P7L36-38: *"[. . .] significantly slower [...]"* Same as above.

- P8L10-12: *"[. . .] has been extended from the ESCiMo simulations to cover our campaign time period (see Section 2)."* The authors give little detail about the extent of this extension. In Section 2, this "extension" of the specified dynamics simulation from the original ESCiMo simulations is not even mentioned at all. The authors refer to Jöckel et al. (2016) for the description of the set-up used. However, Jöckel et al. (2016) give at least 4 simulation set-ups (e.g. RC1SD-base-07/10/10a, RC2-base-04) on which this extension might be based. Presuming this extension includes the time span 2014-2017, it is not clear which prescribed tracer emissions were used. The authors need to provide more details.

- P8L25-28: *"As no direct tropopause information was available for the TOMCAT output [...]"* Refering to Section 2 the authors state: *"Local tropopause information for the flights with HALO have been derived from ERA-interim data [. . .]"* Could the tropopause information from ERA interim, with which the CTM was driven, be derived?

- P8L29: *"[. . .] EMAC tropopause and the climatological tropopause differed by less than 3 K[...]"* How is the EMAC tropopause defined? How does it differ from the WMO/PV definition? In principle, they are comparing a "modeled" tropopause in EMAC, which has been nudged to ERA interim, with a tropopause climatology directly derived from ERA interim. They do not discuss that fact. What is the point here? Temporal and spatial stability of the tropopause in the time frame of interest?

- P9L10-18: The authors are referring a sensitivity study conducted within the EMAC framework's quasi chemistry-transport model (QCTM) mode (Graf 2017). Four different emission scenarios for VSLS are compared therein. They conclude their qualitative comparison of Figs. 7, 8 with the conclusion: *"It is therefore clear that the observed differences are not primarily caused by the model but rather by the emission scenarios."* With respect to the shape of the vertical VSLS concentration profiles, this may actually be true. But this statement does not hold if one takes a closer look at, e.g. the VSLS concentrations at the tropopause between EMAC and TOMCAT. On the back of an envelop – normalize the EMAC QCTM sensitivity studies (Fig. 8) with respect to the Warwick scenario and compare, e.g. Ziska scenarios, with TOMCAT in Figure 7. One finds that TOMCAT mixing ratios of $CH_2Br_2$ at the tropopause seem to be about 0.3 ppt higher than in EMAC. The spread between the scenarios is roughly 0.5 ppt ( 0.2 ppt if the Warwick scenario is excluded). Hence, there seems to be a substantial difference between the models, too.

- Section 5: The authors describe a "simplified approach" to estimate total and inorganic bromine from VSLS in the extratropical stratosphere. They assume a linear mixing of two air masses, wherein the fraction of bromine with tropical origin increases linearly with increasing $\Delta\theta$. The bromine mixing ratios are only evaluated once, at the tropical (TTP) and extratropical tropopause (ExTTP). This ansatz completely neglects the actual transport (horizontal as well as vertical) and therefore the further photochemical transformation (depletion) which VSLS undergo. Based on kinetics, one would expect the concentration of any of the VSLS (let's call it [A] for 'any') at the tropopause ($[A](0,0)$) to follow a powerlaw:

$$[A](t,\theta) = [A](0,0) \cdot e^{-k \cdot (t(\theta) - t_0)}, \qquad (1)$$

with e.g., $k \propto J_a + k_A(T)[OH]$. If we look at [A] at $\Delta\theta$ and a time $t'$ and assume the same linear superposition of air masses as the authors:

$$[A](t', \delta\theta) = (f^{\text{ex-trop}} - 1)[A]_{\text{TTP}} \cdot e^{-k \cdot \Delta t_a(\Delta\theta)} + (f^{\text{ex-trop}})[A]_{\text{ExTTP}} \cdot e^{-k \cdot \Delta t_b(\Delta\theta)} \quad (2)$$

Although the authors state *"[...] for models it is necessary to have a realistic representation not only of chemistry but also of transport in the lowermost stratosphere."*, this is not discussed, e.g. in a critical evaluation of the approach itself. Consulting Figure 4, the linear approach seems to be justified for $[CH_2Br_2]$, whereas $[CHBr_3]$ clearly follows a powerlaw. This is not discussed properly in the manuscript. The authors should reevaluate their approach and change it (if possible) or at least discuss it more thoroughly.

- Figure 4: It is not clear whether the shown error bars refer to the standard deviation ($\sigma$) in each bin or to its standard error ($s = \sigma/\sqrt{(N)}$). Latter is a more reasonable choice. In any case, displaying data in such way does only make sense as long as the distribution in each bin is Gaussian. Otherwise, a violin plot may be more valid. Have the authors checked their distributions?

[Figure]

- Figure 7: Caption L11: *"[...] extremely high values [...]"* What is "extreme" in this context? Which exclusion criterion has been used? The manuscript doesn't provide any further information on this. Please elaborate on this matter in the appropriate section.

**3   technical corrections**

purely technical corrections

**3.1   General**

Typesetting of the name "Ordóñez" in citations is not coherent and incorrect at times.

**3.2   Specific**

- P1L22: *"Distributions [. . .] below the tropopause shows [...]"* → "show"

- P1L26: *"A scenario which has emissions most strongly concentrated to low latitudes [...]"*. This sentence needs rephrasing – maybe: "A scenario with emissions mainly confined to low latitudes [...]"

- P3L8-10: *"Two pathways for input of halogens from short-lived gases are discussed: Source Gas Injection (SGI),where the halogen is transported to the stratosphere in the form of the source gases; and Product Gas Injection (PGI), where photochemical breakdown products of source gases are transported into the stratosphere, usually in inorganic form (i.e. Bry )."* This sentence is not concise. Please rephrase.

- P3L11-12: *"While halogens transported into the stratosphere due to PGI are usually directly in a form available for catalytic ozone depletion reactions, halogens from source gases must first be released in the stratosphere photochemically."* You may rephrase this along the lines: "Halogenes from product gases are readily available for catalytic ozone depletion reaction. Source gases have to undergo a photochemical transformation first."

- P3L23: *"The main source of brominated VSLS is believed to be[...]"* Although orally excepted, one should refrain from the usage of "believe" in a scientific context. Please rephrase. You may use, e.g., "most likely" or "observations indicate".

- P3L41-43 and P4L1-3: *"These observations are compared [...]"* and *"[. . .] these observations are [. . .] presented [. . .] and compared [...]"*. These sentences are almost identical. You may merge them into the latter.

- P4L7-11: *"An isothermal channel uses [...]"* and *"The second channel [. . .] uses [...]"* You may rephrase these in passive voice.

- P4L31,33: *"mean age"*. Supposedly "mean age of air" as used later on in the manuscript.

  Section 2: The authors only mention the spatial resolution of their models. It might be worth to mention the temporal resolution of the model output, too.

- P6L28: *"As in previous work [...]"* → "works"

- P6L28-33: First of all, most of this is a repetition of the text written in the beginning of section 3 and can be dropped. *"As in previous work [...]"* and *"However, we propose a somewhat different approach [...]"* These two sentences are slightly contradicting. They can and should be merged along the lines: "We slightly diverge from approaches in previous works [...]"
- P6L36-38: *"In order to ensure that this tropopause value is representative [. . .]"* This sentence is too long and might be grammatically incorrect (*"[...] when we have observations [...]"*). Please rephrase.

- P7L7-9: *"[...] which has a rather long lifetime in the cold upper troposphere and lower stratosphere (Hossaini et al., 2010) even quite deep into the stratosphere."* This sentence is quite unclear and needs rephrasing.

- P7L9-10: *"[. . .] is strongly depleted [...]"* Can you quantify this statement?

- P7L10-12: *"[. . .] flight levels [...]"* You may refer to "flight altitudes" instead.

- P7L36-38: *"This is most likely [. . .]"* The authors may rephrase this sentence into two. The natural breakpoint would be *"[. . .] with latitude. As [...]"*.

- P8L6-7: *"[. . .] are based on the emission scenario by Warwick et al (2006) [. . .] was run with different emission scenarios (Ordoñez et al., 2012;Ziska et al., 2013;Liang et al., 2010)."* The authors should mention which or the 8 scenarios by Warwick et al (2006) and which of the three by Liang et al. (2010) they have used.

- P8L36: *"overestaimtion"* → overestimation

- P9L4-6: *"Because of the different chemical lifetimes [. . .] above 20 K above the tropopause [...]"* This sentence might need some rephrasing. The authors may use $\Delta\theta > 20$ K.

- P9L10-12: *"In Order to [...]"* This sentence is too long and not concise. Please rephrase.

- P9L12-13: *"Note that these simulations [...]"* You may drop "Note that"

- P9L23-24: *"Again, we use equivalent latitude*[...]"* This has been stated several times by now. As the authors indicated by using *"[a]gain [...]"*. They should consider dropping the whole sentence or the "again" therein.

- P9L39: *"The direct comparison of the distributions between the different model data sets is also interesting."* This statement is redundant. Please rephrase, e.g. "We will have a look at [...]"

- P11L8: *"In this Section [...]"* Add a comma after "section".

- P11L33: The indent of the equation number is incorrect. The authors may reduce the equation to $\mathrm{Br_{tot}}(\Delta\theta) = \mathrm{Br}_y(\Delta\theta) + \mathrm{Br_{org}}(\Delta\theta)$

- P12L20: *"Figure 14 compares [...]"* There is a grammatically issue here. A figure cannot compare anything. Please rephrase using passive voice.

- P13L18: *"have e a"* → Remove the "e" in between.

- P13L30: *"[. . .] a large dataset [...]"* Can you quantify this?

- P13L36: *"[. . .] which is line [...]"* → "which is in line"

- P13L40: *"in high latitudes"* → "at high latitudes"

- P14L3: *"[. . .] with large differences produced by the different emissions."* It may be better to use "caused" in this sentence.

- P14L3-5: *"Overall, for $CH_2Br_2$ [...]"* Please consider rephrasing this sentence.

- Figure 1: The figure is too wide. Would it be possible to color code the different flights temporally?

- Figure 2: Is is possible to highlight the flight path displayed in Figure 3?

- Figure 4: Caption L8: *"dotted"* → "dashed"

- Figure 7: Referred to choice of colors under "general". Caption L10: *"bene"* → "been"

- Figure 14: Please include the information given in L1 of the caption also in the legend of the plots.

- Figure 15: The x-axis label of the lowermost panel seems to be incorrect (compared to the others in the figure) → *"Br total and Bry from VSLS"*. Caption L1: *"[...] at $\Delta\theta$ of 40 K [...]"* → "at $\Delta\theta = 40K$"

---

## Referee Comment (RC2) · Rafael Pedro Fernandez (Referee) · 23 Oct 2019

The paper presents a complete set of carbon-bonded VSLS source gases (SGs) measurements performed in the Northern Hemisphere mid-latitudes using a GC-MS instrument on board an aircraft, as well as a comprehensive comparison of the observations with a complete set of model simulations oriented to evaluate the contribution of tropical and extra-tropical injection of VSLS to the lowermost stratosphere. The main results of the work are: i) the troposphere-to-lowermost-stratosphere transport of VSLS SGs through the extra-tropical tropopause is larger than that occurring within the tropical tropopause; ii) the contribution of both tropical and extra-tropical VSLS injection must be considered in order to reproduce the VSLS abundance within the mid-latitudes lower stratosphere below 400 K; and iii) the models and inventories used in this work show

certain limitations in reproducing the VSLS reactive transport and estimate the release of inorganic bromine (Bry) in the NH lower stratosphere. The paper also includes a seasonal, latitudinal and vertical analysis of VSLS abundance in the UTLS.

I found the paper very interesting and very well organized, presenting results in a clear and comprehensive format, and including interesting and constructive discussions. It is worth noting that even when the altitude/latitude-dependent observations itself would be worthwhile to be published, the authors have decided to go forward and present a comprehensive model-observation inter-comparison, which contributes to improves the general understanding of the reactive-transport efficiency of VSLS species within the UTLS. In particular, I found very descriptive and intuitive the vertical coordinate system ($\Delta\theta$ and $\theta$*) they used to represent all results relative to the altitude of the tropopause, which allows a consistent description of the vertical and latitudinal decay of VSLS once they are injected to the stratosphere. At the very end of the paper, a simplified approach (eq. 4) is used to estimate the total amount of inorganic bromine (Bry) released by VSLS within the mid-latitude lower stratosphere, highlighting the major importance of properly reproducing these Bry levels in model simulations oriented to determine the ozone impact of VSLS.

Having said this, I believe that the manuscript posses a handful set of specific issues and many technical details (including figures and tables captions) that must be corrected before final publication.

Major Concerns:

1. I would like to start mentioning that most of the "important questions" that came to my mind while I was reading the manuscript had already been responded (i.e., as I moved forward with the lecture and reached Section 5). Even when this should be taken as a mainly positive comment, it also implies that some of the analysis/discussions given at the end could be (at least partially) shifted to earlier sections, to help the reading and support the analysis. For example: Sections 3 is concentrated on carbon-bonded

VSLS mixing ratios close and above the tropopause in the lower stratosphere, but there is only a brief mention and sideway comparison with the SGI values compiled in the last WMO report (P7,L27). The reader needs to wait until Table 5 is presented (Section 5, P12) to reach a complete discussion and comparison with WMO values, and it is only at this point that the importance of the model-observation inter-comparison (presented in Section 4) becomes evident. In doing so, note that Table 5, which provides values for VSLS SGs within the tropics and extra-tropics, is introduced in the only section of the paper focused on PGs (Section 5). 2. Observations from TACTS and WISE campaigns have been merged into a unique dataset (WISE_TACTS) because they were performed during the same seasons. Even though I found this procedure correct, I wonder if the authors have analyzed this data separately to evaluate if there is at least any glimpse of VSLS SGs trend within the NH-UTLS (there are ∼5 years between both campaigns). The authors declare they combined both dataset based on "general" observational evidence (P3,L29), but I think they should justify this procedure by evaluating specifically their unique and novel dataset (P4,L29).

3. Authors should be really careful and consistent when using the wording "total bromine". Until Section 5 is reached, only carbon-bonded (i.e., organic) bromine is considered, and total bromine is referred as the sum of CH2Br2 + CHBr3 + minor_VSLS (P6,L2; P8,L15). But later on Sections 5 and 6, "total bromine" points out to the sum of Br_org + Br_inorg (P11, Eq. 1). Please, be consistent and refer to "total organic bromine" and/or "total bromine" whenever appropriate.

4. Although the vertical profiles for CH2Br2 and CHBr3 are analyzed in detail, there are no mentions regarding the "error bars" presented in the figures (e.g., P6,L20; Fig. 4 caption). Do the error bars correspond to 1-sigma or 2-sigma? Besides this, the authors should explain why the vertical error bars for $\Delta\theta$ are not the same for the different bins, as well as why the error bars for CHBr3 are larger than for CH2Br2? Is this only due to the shorter lifetime of CHBr3, which shows quite different vertical profiles depending on the exact latitude within the 40-60° bin (as observed in the latitude-altitude

cross-sections), or could this also be attributed to differences in their regional source strengths? Note that the variability is only considered in (P7,L34; Fig. 6) when addressing how the VSLS tropopause abundance changes with latitude, but I think that a more complete comparison of this latitudinal and vertical variation within the 20-40°N bin should be provided (at least the two major VSLS).

5. P8,L18: "The contribution from these mixed bromochlorocarbons to total VSLS bromine are typically on the order of 20%, while about 80% of total VSLS bromine in the upper troposphere and lower stratosphere is due to CH2Br2 and CHBr3". → Are this percentages computed using WISE_TACTS and PSG data? Have you compared this findings with other studies (i.e., Fernandez et al., 2014)? Have you thought about presenting combined results of your observations of the sum of these minor_VSLS into a figure or table? (currently only results for the sum of CH2Br2 + CHBr3 + minor_VSLS are given)? As minor_VSLS posses in some cases lifetimes larger than CH2Br2, this information could be useful for future studies. The importance of the minor VSL contribution becomes also evident in Section 5 (P13,L11) when the overall contribution from longer-lived VSLS to Bry is discussed.

Specific Comments:

1. Title and Abstract: Shouldn't the title be more specific on the extra-tropical (or the "tropical vs. extra-tropical") contribution of VSLS bromine to the UTLS?? In addition, the abstract has an excessive focus on the results obtained with the different models and emissions scenarios (including specific statements for some of the scenarios giving the best and worst agreement). I would expect the abstract to focus on the contribution of tropical vs. extra-tropical contribution to SGI, and in any case to provide a rough estimate of the relative contribution of each of these two pathways to the overall organic and inorganic lower stratospheric bromine (and providing only a general mention to the similarities and discrepancies between models and observations and its dependence on latitude and season).

2. Section 3 (P5): The first paragraph of section 3 describes the spatial and vertical coordinates used for representing measured and modeled data. In my opinion, the selection of $\theta$, $\Delta\theta$ and $\theta^*$ variables really improves the analysis and interpretation of the results. However, I believe the initial description of how these variables are computed is not clear enough, and the reader needs to go back and forth between figures and text to completely catch up the difference among them. For example: a) $\Delta\theta$ is mostly used for vertical profiles figures, whereas $\theta^*$ is used for latitude-altitude cross-sections (which is not clearly mentioned in the text); b) there are at least 2 or 3 places where $\Delta\theta$ and $\theta^*$ vertical coordinates are defined, and in some cases slight differences on the definitions are observed (P5,L31 and P6,L32). In particular specify if the vertical coordinate is computed above the "local" tropopause "for each latitude" and how is it added to the "climatological" tropopause.

3. P11,L2-5: "Most importantly, the overall levels, especially in the low latitudes, are much higher than our observations and also much higher than the tropical observations compiled in the WMO report (Engel and Rigby, 2018). This will result in too much VSLS bromine being simulated in the stratosphere, and therefore also in a misrepresentation of the input to the lowermost stratosphere via the different pathways". → In addition to the general description focused on the 340-400 K range, I found interesting that poleward of 40° and below 320 K (see Fig. 9-10) there is a negative model bias for CH2Br2 mixing ratio exactly at the extra-tropical tropopause, while at the same time there is a positive model bias for CHBr3. This is not mentioned nor explained in the text. 4. P11,L28: The simplified approach considering fext-trop and ftrop" is very intuitive and helps to visualize the contribution from tropical vs extra-tropical bromine from VSLS, but it would be useful to provide at the end a conclusive sentence of which are the most probable fractional contributions from tropics and extra-tropics to the overall bromine in the UTLS. Certainly, concurrent Bry measurements would be required to close the whole bromine budget, but at least from Figure 15 ($\Delta\theta$ = 40 K) it seems that fext-trop values close to 20-40% produce the closest agreement between model and observations. However, on the analysis presented for Figure 14, fext-trop is 60% at

this height, isn't it? I suggest to expand the analysis and discussion of fext-trop and ftrop" (you only dedicate a few lines to this subject at the end of Section 5). Finally, why are not equivalent results for WISE_TACTS provided in Fig. 15? 5. P12,L28-29: "The larger Bry derived in the model calculations above 60 K is caused by the higher total bromine values from CH2Br2, which are caused by the higher CH2Br2 levels at the tropical tropopause in comparison to the observations." P12,L32-33: "In the lower part the discrepancy is more due to higher simulated CH2Br2 in the lowermost stratosphere than found in the observations" → This could "partially" be the reason, but it could also be due to using an improper fext-trop value at this specific vertical level and/or due to the simplified linear approach used. Please elaborate on this. 6. P14,L29: "we have shown that there will be significant differences in stratospheric Bry depending on the emission scenario, which can be as high as 2 ppt, corresponding to a difference of a factor 2 relative to observation-derived values". → Being this sentence included in the conclusions (and also mentioned in the original abstract), I suggest informing not only the largest (i.e., worst) difference, but also the minimum model-observation differences, as well as the range of model bias results for the models which show a better performance.

Technical Corrections:

P1,L17: "The instrument is extremely sensitive due to the use of chemical ionisation, allowing detection limits in the lower parts per quadrillion (10 -15 ) range". → Is this information of major importance to be included on the abstract? Consider also including the GS-MS acronym in the preceding sentence.

P1,L35: "Depending on the underlying emission scenario, differences of a factor 2 in reactive bromine derived from observations and model outputs are found for the lowermost stratosphere, based on source gas injection." → Consider rephrasing, and also mentioning the range of agreement of models (see comment above).

P3,L30: "A further future increase has been projected" → A future increase of VSLS

emissions has been suggested"

P3,L38: "In order to investigate the regional variability of bromine input into the lowermost stratosphere and the inorganic bromine loading of the extratropical lowermost stratosphere, we have performed a range of airborne measurement campaigns ..." → You explicitly mention "inorganic bromine" but not "organic bromine" in a sentence focused on the novel measurements dataset. Consider revising, as you've only measured carbon-bonded species, and only inferred inorganic bromine.

P3,L42;P4,L3: Please specify which are "the implications" you are pointing at.

P4,L5: Consider changing the subtitle to "Instrumentation and Observations"

P4,L22: Check for consistency between the year of the TACTS campaign between the text (2011) and table 2 (2012).

P4,L29: "covered a similar time period and latitude range" → you mean same seasons, consider rephrasing.

P5, L2: "ESCiMo (Earth System Chemistry ntegrated Modelling)" → Integrated

P5,L12: TOMCAT acronym already defined above (P5,L2).

P5,L7;P5,L13: If both EMAC and TOMCAT are driven by exactly the same ECMWF ERA-Interim reanalysis data, this should be mentioned explicitly. This will help to override additional uncertainties regarding differences between models. Also, although EMAC can be run as a CCM model, it should also be clear that for the current SD simulations, the model behaves like a CTM.

P5,L17: "Emitted VSLS (CHBr3, CH2Br2, CH2BrCl, CHBr2Cl and CHBrCl2) are destroyed by reaction with OH and photolysis in the model" → I assume this is also the case for the EMAC model described in the preceding paragraph. You should explicitly mention this to avoid confusion.

P5,L24: Although the paper reads perfectly well using the $\theta$ vertical coordinate, it

should be mentioned at least during the model description which are the equivalent altitude/pressure values for the tropical/extratropical tropopause $\theta$ levels used in this work.

P5,L36: "we have also binned the data in potential temperature in 10 K potential temperature intervals" → repetitive

P5,L38 and elsewhere: "relative to the mean tropopause observed during the campaigns" → in many places the authors make reference to "the campaigns" in a general meaning, when I understand they are pointing out to "each dataset" obtained during the campaign, and not the campaign itself.

P5, L38: "The results are presented ..." → consider rephrasing the whole sentence, as it is very difficult to understand. It should also be mentioned at least once that whenever you mention winter, spring or fall, you are always pointing out to "boreal" seasons. It is not necessary to repeat it all over the text, but I only found it mentioned properly once in the conclusions (P13, L39).

P6,L15: "again in line with their atmospheric lifetimes, which generally decrease with an increase in the bromine atomicity of the molecule". Atomicity should also be used in P7,L25.

P7,L10: "The shorter-lived CHBr3 is strongly depleted already about 20 K above the tropopause" → Based on Figure 5, this is only the case during PGS, but not for WISE_TACTS during the summer. Could this be related to stronger convective transport during the summer? Please explain.

P7,L20-22: "In order to ..." → there are many sentences that begins with this wording. Although I found it correct, please avoid it using more than once within a paragraph (i.e., here it is used in two consecutive sentences).

P7,L23: "Again, for the tropospheric data, standard latitude has been chosen, while equivalent latitude was used for all data with $\Delta\theta$ above zero" → This got me confused:

Figure 6 focus on extra-tropical tropopause values (averaged considering bins 10 K below the local tropopause). Wouldn't this correspond to negative $\Delta\theta$?

P7,L29: explicitly point at Tables 3 and 4.

P7,L30 and Figure 6: Could the "negative latitudinal gradient" observed for WISE_TACTS be somehow unrealistic/largely-biased because of using a small amount of measurements below the tropopause at larger latitudes? Or because of any type of cos(lat) averaging factor?

P7,L34: "derived around the tropopause" → wouldn't it be "below" (-10 K) the tropopause.

P8,L1-5: Sentence is too long. Please rephrase.

P8,L13: "Here we compare vertical profiles, geographical distributions and latitudinal gradients between our observations and the model results". → What do you mean by "geographical distributions"? $20°$-$40°$ bin?? If not appropriate, please remove.

P8,L23: "about 40 K" → "∼40 K"

P9,L4: "Using the Ziska et al. (2013) emission scenario, the overestimation of CH2Br2 and the underestimation of CHBr3 tend to cancel out, resulting in a reasonable agreement in total VSLS bromine. Because of the different chemical lifetimes of the two species, this results in a wrong vertical distribution of Bry with too high mixing ratios above 20 K above the tropopause in winter and a much steeper vertical gradient in late summer." → First, in the initial sentence it should also been explicitly mentioned that, in addition to the CH2Br2 overestimation and the CHBr3 underestimation, the contribution form "minor VSLS" which is also considered for the total "organic" bromine results is based on the Ordoñez inventory ... adding an additional uncertainty to the different contributions that "cancel out" each other. Second, What do you mean by "vertical distribution of Bry" in this context?. Please be careful when pointing out to the inorganic or organic bromine in this section.

P9,L23: Fig. 9 to 12 instead of 9 and 10?

P10,L6: "much lower" → please be more specific

P10,L15: "Therefore, we compare the observed mole fractions of the brominated VSLS in the upper troposphere with those determined from the different model setups, in order to investigate if the models are able to represent the latitudinal gradient in upper tropospheric mole fractions." → please rephrase.

P10:L17: "or respectively modelled (EMAC)" → please rephrase (see comment on respectively below)

P10,L32-37: The text goes back and forth a couple of times between wintertime results and summertime results. It would be simpler to describe all results for one season before moving to the other.

P10,L36: Here and elsewhere . . . "extremely high" → what do you mean by extremely? Wouldn't just "larger" be enough?

P11,L12: "is expected to add more bromine on top of SGI"

P11,L14: "imply for the total bromine and inorganic bromine" → you mean total organic bromine or total (organic + norganic) bromine? Or both? Please see the general comment above.

P11,L18: Too many "and" within a single sentence. Please rephrase

P11,L23: "No studies on mass fractions are available for the campaigns discussed here, so we will rely on previous studies for these fractions." → I do not understand the rationale for including this sentence. Please make it clear or remove.

P11,L24: "order of magnitude difference" is normally used to point at scaling factors like 10, 100, 1000. Please rephrase.

P11,L41: Here and elsewhere. "at the tropical, respectively extratropical (40-60°N)

tropopause," → It is very unfamiliar for me the way the "respectively" sentences have been written throughout the text. I suggest replacing them by "at the tropical and extratropical (40-60°N) tropopause, respectively". See also P12,L19-L23.

P12,L6: "by averaging the model respective observations" → This sentence is senseless, in any case "the model results"

P12,L14: fext-trop decreases with altitude, not increases.

P14,L6: "with a downward revision" → do you mean shifted to the lower edge of the range of emissions?

P14,L26: "The bromine budget in the lower stratosphere will depend on the relative fraction of air from the tropical and extratropical tropopause. The relative contribution of extratropical air will decrease with altitude and should reach zero at about 400 K potential temperature.." → is it the future (will) tense appropriate here??. In any case, I believe the authors are pointing at a decrease with altitude and not latitude here (if not, please explain the idea and make it clear).

Tables and Figures:

Most of Figures and Table Captions include the "long-name" of each of the Campaigns instead of just providing their "short-name"/acronym (PSG=POLSTRACC+GW-CYCLE+SALSA, TACTS_WISE, HALO, ECMWF, etc.), which is not only simpler, but also more familiar to everyone. Using the short-name version will certainly improve the captions readability. Also, there is no need to define ppt each time you use it (only once at the beginning within the text is enough).

All Tables: Consider using a "one line title" at the top of each table, and then provide all specific information regarding the season, specific campaign, altitude/latitude range, etc. as footnotes on the table.

All multiple-panel Figures should indicate, in addition to the (letf, PSG) and (right,WISE_TACTS) information, that results are provided for (top, CH2Br2), (middle,

CHBr3) and (bottom, total organic bromine). (This example was based on Fig. 13, and should be adapted to the specific figure).

Table 3 and 4: What does "TP" stands for (tropopause)?. Why Table 3 provides values for TP + 30-40 K, while Table 4 only for TP + 40 K? Define what does the stdev. stands for and how is computed? Wouldn't it better to provide the stdev. value with a +/- sign (0,55 +/- 0,09) ppt within the mole fraction column?

Table 5: Have you used the annual mean or the seasonal mean for computing the tropical tropopause values?? (P12,L6). In case a seasonal mean has been used, please specify which months have been used for the model output. Replace bromide by bromine. Rephrase respectively. Explicitly indicate that the Br_ext-trop and Br_trop are for ($\Delta\theta = 0$). Indicate that ML stands for Mid-latitudes, which you called extra-tropics throughout the text. Define explicitly in a table footnote the $\Delta\theta$, latitude range and any other relevant information that has been used for computing the values presented in Table 5.

Figure 4: What are the horizontal and vertical bars? 1-2 sigma? Also, in the text it would be good to explain within the main text why for WISE_TACTS there are some points for which the mixing ratio as a function of $\Delta\theta$ can be computed ($\Delta\theta = 100$), but not for the $\theta$ coordinate ($\theta = 420$ K).

Figure 5: Consider introducing a dashed/dotted vertical line indicating the 40-60° boundaries (and any other important latitude) used for the extra-tropics vertical profile computation.

Figure 7: Consider reducing the length of the caption, and moving some of information at the end of the caption to the main text.

Figure 8: Specify output for this simulations is not in SD mode as for the other simulations (here and in the main text).

Figure 9-12: Consider including vertical dashed/dotted lines as suggested for Fig. 5.

Also . . . for the model (right) panels: Why there are some "empty/blank" boxes within tropical lower stratosphere above 400K? Is that because of the vertical model resolution and/or upper limit of the models?

Figure 13: Have you considered the idea of expanding the lower latitudinal edge of the figure to 0° Lat, and include the model/WMO results for the Tropical mean (as shown in Table 5)? Also, for the most poleward latitudinal bin, it looks like the "modeled" values are ploted at a higher latitude than the "observations". This must be due to the total number of data-points used to compute the VSLS value. This should be explained either in the caption or in the text.

Figure 15: Why not including panel for WISE_TACTS in this figure?

---

## Referee Comment (RC3) · Anonymous Referee #3 · 6 Nov 2019

The manuscript by Keber et al. describes measurements of short-lived organic bromine gases from several airborne campaigns in the UTLS mostly north of $40°$ latitude in different seasons. The authors then evaluate how different combinations of models and emission scenarios compare to observations. The study is relevant to better quantifying the role of organic bromine in catalytic cycles that deplete stratospheric ozone. The main points of the paper are the demonstration of a latitudinal gradient in UTLS organic bromine, a winter maximum in VSL Br gases in the UTLS, recognition that VSL Br concentration near the extratropical tropopause is greater than that at the tropical tropopause, and that current emission scenarios show a range of imperfections that limit their accuracy. I might argue that all of these conclusions are not new, but the measurements, importantly, quantify the levels of UTLS organic Br that provide a

solid basis for more detailed and quantitative analysis. Thus, the manuscript will be a valuable contribution to the further understanding of halogen chemistry in the lower stratosphere.

I think the strength of the manuscript is in the presentation of the data and identification of the seasonal and latitudinal variability in the measured Br gases. However, much of the paper focused on the model and emission scenario comparisons to the data. The authors state in their summary, "While it is not the main purpose of this paper to evaluate emission scenarios. . ." they nonetheless spend a large amount of effort to do just that, and they do so in more detail than is needed to demonstrate their point that models have problems. If model comparison to data is a significant part of the message, then I would like to see (perhaps in the Supplement) how the models compare to measurements for CFCs, halons, CH4 or other gases with better understood tropospheric distributions. Does model SF6 compare well to the binned measurements from these missions? This would help provide some context for looking at the Br distributions and their deviations from model predictions. My suggestion, though, is not to add more model discussion in this paper, but to reduce the level of different comparisons to models that is done in the text and just highlight major issues associated with emission scenarios.

I also had some trouble sorting through the different manipulations that the authors used to help with the comparisons between measurements and models. The authors go into great detail in how they adjusted and binned the data to presumably reduce variability from variations in dynamics and transport, and to provide a more robust comparisons to model outputs. While I would agree that these might be reasonable adjustments, I found the multiple presentations confusing and left me wondering about what I was viewing in the plots and how the data might look with no adjustments. Perhaps the authors can think about possibly simplifying (or better explaining) the discussion of the adjustments. Even though the goal of the adjustments was to reduce variability due to dynamical factors, the resulting distributions retained significant spread in the data,

particularly within -+40K of the tropopause. The authors chose to focus on the average profiles, but it would seem that a more detailed evaluation of the variation (and its comparison to variance in a model) could produce interesting results on sources, transport and variability of Br in the critical region near the tropopause. Maybe a subsequent paper can do this.

In section 5, I did not understand the need to go through the linear mixing model to show how the emission scenarios that placed high (or low) CH2Br2 (or CHBr3) in the tropical (or extratropical) tropopause were different from measured values. Data summarized in Table 5 is sufficient to demonstrate the impact of "incorrect" modeled organic Br in the tropics or extratropics. Why go through assumptions that aren't necessarily realistic? Or you might be able to improve the mixing model with data from other gases, such as SF6, which can be used to perhaps better estimate tropical vs extratropical fractions of air.

Given the goal to provide a more robust assessment of the role of Br in the lowermost stratosphere at N latitudes, and given the availability of multiple models, I would have found valuable some discussion of the actual impact of different Br levels on the ozone budget of the impacted region. How does an uncertainty of -+ 1 ppt VSL Br propagate to modify ozone destruction rates?

Other comments:

Minor VSL Br. A) I would suggest adding profiles of the minor VSL Br in a Supplement. B) The authors state that "the observed decrease with altitude in the stratosphere is consistent with the relative lifetimes of the different compounds". While generally true, this does not seem to be the case for CHBrCl2 compared to CH2Br2 (page 7 line 15). Lifetimes of these compounds differ by a factor of about 3.5, but the gradients appear close to the same. Perhaps something is wrong with the lifetime estimate? C) Tables 1 and 3 show the mixing ratio of CH2BrCl as 0.1 – 0.2 ppt (100 – 200 ppq), but the GhOST-MS is reported in Table 1 to have a detection limit of 130 ppq (0.13 ppt). Not

sure how this is possible. I note also that the GhOST-MS characteristics are quite different here compared to that reported in Sala et al. (2014). Was there a significant modification to the instrument?

Uncertainties. Page 6, top. Please report uncertainties associated with the total organic bromine levels reported here and elsewhere.

Organic/Total Br. . .I second the recommendation of one of the other reviewers to be more careful in terminology of total bromine/total organic bromine, etc.

Data availability. I did not see that these data are available in any public archive. Please list how the data from these flights can be accessed.

Typo: Table 1 "Reproducibility"

---

## Author Comment (AC1) · 20 Dec 2019

**Reviewer #1**

*1 General comments*

5 The paper presents novel observational data of VSLS from several aircraft campaigns and compares them with modeling results. It extends on and confirms previous findings. Hence, the title does reflect the contents of the paper.

Some scientific methods and assumptions need more thorough outline and proper discussion.

10 *This is a rather general remark, which makes it hard to give a specific answer. We have added some further explanation where specific remarks have been given, especially in section 5 with respect to the derivation of $Br_y$.*

The results are sufficient to support the interpretations and conclusions, but not all results can be repeated based on the information given, e.g. the derivation of the tropopause for the campaigns labeled as "private conversation".

15 *The tropopause information was indeed labelled as "private communication". However, the details and references are given and Michael Sprenger from ETH is acknowledged, therefore the private communication has been deleted.*

The authors give proper credit to related work and indicate their own contribution, but they could make it more clear from the start, what the current consensus regarding the emission scenarios is.

20 *This is a good suggestion. We have added the following text in the Introduction describing the conclusion from the most thorough review papers on this by Hossaini et al., after the introduction of the 4 scenarios:*

*"Hossaini et al. (2013) concluded that the lowest suggested emissions of $CHBr_3$ (Ziska et al., 2013)) and the lowest suggested emissions of $CH_2Br_2$ (Liang et al., 2014) yielded the overall best agreement in the tropics and thus the most realistic input of stratospheric bromine from VSLS. They also concluded that "Averaged globally, the best*
25 *agreement between modelled $CHBr_3$ and $CH_2Br_2$ with long-term surface observations made by NOAA/ESRL is obtained using the top-down emissions proposed by Liang et al. (2010)".*

The abstract provides a complete summary, but it does not deliver regarding the WHY unit, e.g. the context and importance of the measurement. It gives the impression of a paper focused on observational data, whereas 2/3 of
30 the figures and text are related to model evaluation. If the point of the paper is to evaluate existing emission scenarios through a new set of data, this should be made more clear.

*The main point of the paper is not to evaluate emission scenarios, but rather to show the increase in observed tropopause values with latitude especialld during winter and to discuss how large the differences between our*
35 *observations and different model setups are. We have added a sentence at the beginning of the abstract explaining the WHY, as suggested by the reviewer:*

*"These rather short lived gases are an important source of bromine to the stratosphere, where they can lead to depletion of ozone. The measurements have been obtained using an in-situ gas …."*

40 The overall presentation is well structured and clear. The language is fluent. The authors should be more cautious with the usage of the term "significant". Some parts of the paper need editing. In particular, the manuscript does not follow the ACP guidelines in several points:

*The term significant has been replaced at several places:*

45 *The Asian monsoon has also been named as a possible pathway for transport of bromine from VSLS to the stratosphere: significant changed to possible*

*The distributions observed during the WISE and the TACTS campaigns show rather high levels of $CH_2Br_2$: significantly changed to rather high levels.*

*The average values derived here for the 10 K interval below the extratropical tropopause are significantly*
50 *larger. Significantly deleted*

*This is most probably related to the increase in lifetime with latitude, as especially during the wintertime PGS campaign the photolytical breakdown in higher latitudes is significantly slower than in lower latitudes. Significantly deleted*

*The EMAC model with the Warwick et al. (2006) emissions significantly overestimates both CH2Br2 and*
55 *CHBr3 in the lowermost stratosphere of the mid latitudes. Significantly changed to substantially*

*As has been shown in the comparison of the vertical profiles, significant differences between model results and observations are found, especially in the case of the Ziska et al. (2013) emissions in … significant deleted*

*The deviations between the TOMCAT model using the Ziska et al. (2010) emissions and the EMAC model*
60 *using the Warwick et al. (2006) emissions are significantly larger. Significantly replaced by substantially*

*In this case, both CH2Br2 and CHBr3 are overestimated significantly in the lower stratosphere. Significantly deleted*

*who showed that TOMCAT using the Warwick et al. (2006) emission scenario significantly overestimated HIAPER Pole-to-Pole Observations (HIPPO) in Northern .. significantly deleted*

5 *As shown in the previous Section, significant discrepancies exist between the various combinations of models and emission scenarios with respect to our observations, both around the tropopause and in the lower stratosphere. Significant deleted*

*In contrast, EMAC results using the Warwick et al. (2006) emissions still show significant amounts of CHBr3: significant changed to substantial*

10 *Therefore, despite the fact that EMAC still shows significant remaining CHBr3 rather deep into the lowermost stratosphere, this model setup significantly .. first significant replaced by substantial, second deleted.*

*Interestingly, while the Ziska et al. (2013) emissions in TOMCAT showed some significant differences, in particular of CHBr3 at the tropopause, the differences in total Bry are not as large. Significant removed*

15 *.. of the Ziska et al. (2013) scenario with seasonally varying emissions, yielded significantly higher tropopause values. Significantly removed.*

*However, it is clear from the comparison with the scenario by Warwick et al. (2006), which restricts emissions to latitudes below 50°, that the sources of these short-lived brominated compounds are not only in the tropics, but that significant emissions must also occur in higher latitudes. Significant replaced by*
20 *substantial*

*Using simplified assumptions about the fractional distributions, we have shown that there will be significant differences in stratospheric Bry depending on the emission scenario, significant replaced by substantial*

*As shown in our sensitivity study (Section 5), the assumptions on the relative contribution of the different source regions has a significant impact especially on the Bry produced from CHBr3 in the lowermost*
25 *stratosphere. Significant replaced by substantial*

*Southern hemispheric distributions are expected to differ significantly from northern hemispheric distributions.. significantly deleted*

• Subsections should be consecutively numbered.
30 *Subsection numbers have been added.*

• Figure captions, figures, and tables:
–Usage of full campaign names renders the captions imprecise and utterly unpleasant to read.
*Full campaign names have been deleted*

–Poor choice of colors (red, green, blue, black, grey) within the line plots (vector graphics!) and the tick labels' font size make the figures hard to read (Fig. 7, Figs. 13-15)
*None of the other reviewers seemed to have problems with the colors and also we feel that they are a rather common choice. Colors are unchanged. However, we have increased the tick label fonts and made these*
40 *consistent throughout all plots. We have also increased the resolution of the plots, which hopefully also increases the readability.*

.–The usage of "[ ]" around units in plots is depreciated (→https://www.bipm.org/en/publications/si-brochure/section5-3.html)
45 *This is not a general ACP guideline. Many papers in ACP use exactly the [ ] notation for units. We could change this if the typesetting requires it. Unchanged.*

• Some white spaces seem odd.
*We have deleted white spaces that seemed odd; if there are more than this could be solved during the typesetting*
50 *process.*

• Equations are not properly set, e.g. usage of "*" as indicator for multiplication.
*\* has been changed to ·*

55 The number and quality of references are appropriate.

*2 specific comments*

• P4L33:"It is clearly visible that the halocarbons correlate [...]"Can you quantify this?

*We are not sure if a quantification of this would really provide additional information, as it would be unclear to which degree a spread around a correlation function is due to instrumental issues and to atmospheric conditions. Also some model (linear/polynomial) would be needed. We have chosen not to explore this in further depth.*

• P5L40-P6L1:"Only bins which contain at least five data points [...]"What criterion led to this choice? (See comment to Fig. 4 below.)

*This number has been chosen arbitrarily, but with the intention of assuring that one single measurement should not influence the mean too much, yet still having a reasonable latitude/altitude coverage. We feel that this needs no further explanation in the paper.*

• P6L21-22:"[...] the variability averaged over the four lowest stratospheric bins was always lower when using Δθ as a coordinate. "Can you elaborate on this? Why does the variability in the four lowermost stratospheric bins change in response to the transformation of coordinates Δθ (relative to the tropopause)? Am I right to assume that this is due to bins with mixed tropospheric/stratospheric data? If that is the case, "four lowest stratospheric bins" is misleading. Please elaborate on this.

*Ideally (if the tropopause attribution were perfect) such mixed bins would exist only in θ and not in Δθ coordinates. The variability in the stratosphere decreases when using tropopause-centred Δθ co-ordinates instead of θ coordinates, due to the elimination of mixing of tropospheric and stratospheric data to a large degree. We have made this clearer by explicitly stating:*
*"For all campaigns, the variability averaged over the four lowest stratospheric bins when using Δθ, was always lower than in the 4 lowest bins above the climatological tropopause using θ as a coordinate(see Tables 3 and 4). This shows that using the tropopause centered coordinate system Δθ reduces the variability and that this coordinate system is thus best suited to derive typical distributions."*

• P7L5-7:"The data have been binned in5◦latitude and 5 K intervals of potential temperature. As expected, the distributions closely follow the tropopause (indicated by the dashed line), with values decreasing with distance to the tropopause and also with increasing equivalent latitude. "In the previous section the authors used"[o]nly bins which contain at least five data points [...]"in a much coarser binning. From the sampling frequency, one ought to assume that the shown averaged VSLS concentrations per bin are based on similarly sized numbers of entries, but this neither emerges from the text nor the figures (Fig. 5, Figs. 9-12). Furthermore, earlier in the manuscript the authors find that Δθ is the coordinate of choice for this study, but they don't use it in these figures. If it was possible to show the data in relative coordinates, this would strengthen their point ("closely follow the tropopause"). The authors should elaborate on this.

*Thank you for pointing this out. Indeed we used our modified potential temperature coordinate (as stated in the beginning of section 3.2. and in the Figure caption and on the axis of Figure 5). We have modified the text here to explain this more clearly:*
*"The data have been binned in 5° latitude and 5 K intervals of the modified potential temperature coordinate $\theta^*$."*

*As with respect to the narrower sampling in this representation: the reason for this is, that this is only used for a graphical comparison, while the sampling of the vertical profiles is used for quantitative comparisons. Therefore the idea was to have a more robust sampling for the quantitative comparison and to use larger ranges of Δθ.*

• P7L7-9:"The distributions observed during the WISE and the TACTS campaigns show significant amounts ofCH2Br2[...]"What purpose does "significant" serve in this context? Elaborate on the actual significance or drop the word.

*See general comment above: changed and quantified to*
*rather high levels of CH2Br2, in the lower stratosphere, with a depletion of only about 35% at 40-50 K above the tropopause*

• P7L10-16:"[...] the most stratospheric air [...]"and"[...] very high mean age of air [...]"If the authors' point is to state that the amount of VSLS in the stratosphere is a function of its residence time there, they should make this more clear.

*We changed the sentence introducing the mean age to:*
*"This stratospheric character is in agreement .."*

• P7L28-29:"[...] are significantly larger [...]"Can you quantify this? Else drop the term "significantly".

*Deleted significantly.*

• P7L36-38:"[...] significantly slower [...]"Same as above.

*Deleted significantly.*

• P8L10-12:"[...] has been extended from the ESCiMo simulations to cover our campaign time period (see Section 2)."The authors give little detail about the extent of this extension. In Section 2, this "extension" of the specified dynamics simulation from the original ESCiMO simulations is not even mentioned at all. The authors refer to Jöckel et al. (2016) for the description of the set-up used. However, Jöckel et al. (2016) give at least 4 simulation set-ups (e.g. RC1SD-base-07/10/10a, RC2-base-04) on which this extension might be based. Presuming this extension includes the time span 2014-2017, it is not clear which prescribed tracer emissions were used. The authors need to provide more details.

*Specified:*

*"The SC1SD-base-01 run which has been used here has been branched off from RC1SD-base-10 (see Jöckel et al., 2016) at January 1, 2000 using the RCP8.5 emissions and greenhouse gas scenario."*

• P8L25-28:"As no direct tropopause information was available for the TOMCAT output [...] "Refering to Section 2 the authors state: "Local tropopause information for the flights with HALO have been derived from ERA-interim data [...] "Could the tropopause information from ERA interim, with which the CTM was driven, be derived?• P8L29:"[...] EMAC tropopause and the climatological tropopause differed by less than 3 K [...]"How is the EMAC tropopause defined? How does it differ from the WMO/PV definition? In principle, they are comparing a "modeled" tropopause in EMAC, which has been nudged to ERA interim, with a tropopause climatology directly derived from ERA interim. They do not discuss that fact. What is the point here? Temporal and spatial stability of the tropopause in the time frame of interest?

*The point is that we are comparing our measurements to the model in tropopause-relative coordinates. It is therefore important to show that the average tropopauses used for this show good agreement, otherwise discrepancies could also be due to inconsistent vertical attribution. A sentence has been added to explain this:*

*"As we are comparing our observations to the models in tropopause relative coordinates, we have also compared this climatological tropopause with the tropopause derived from the EMAC model results for the time of our campaigns."*

• P9L10-18: The authors are referring a sensitivity study conducted within the EMAC framework's quasi chemistry-transport model (QCTM) mode (Graf 2017). Four different emission scenarios for VSLS are compared therein. They conclude their qualitative comparison of Figs. 7, 8 with the conclusion: "It is therefore clear that the observed differences are not primarily caused by the model but rather by the emission scenarios. "With respect to the shape of the vertical VSLS concentration profiles, this may actually be true. But this statement does not hold if one takes a closer look at, e.g. the VSLS concentrations at the tropopause between EMAC and TOMCAT. On the back of an envelop – normalize the EMAC QCTM sensitivity studies (Fig. 8) with respect to the Warwick scenario and compare, e.g. Ziska scenarios, with TOMCAT in Figure 7. One finds that TOMCAT mixing ratios of $CH_2Br_2$ at the tropopause seem to be about 0.3 ppt higher than in EMAC. The spread between the scenarios is roughly 0.5 ppt ( 0.2 ppt if the Warwick scenario is excluded). Hence, there seems to be a substantial difference between the models, too.

*We thank the reviewer for pointing towards this. Indeed differences between the models cannot be excluded as an additional point. However, no direct comparisons are possible, as Figure 8 is based simulation with EMAC,for a different time period. We only state that the differences are not primarily caused by the models, which is still valid. However, we have rephrased the statement to emphasize that a similar picture arises when comparing the 4 scenarios in one model, in particular with respect to the order of the scenarios (highest CH2Br2 in the Warwick scenario and lowest vertical gradient in the stratosphere for CHBr3 in the Warwick scenario):*

*"Differences between the different models are certainly a factor in the explanation of model-observation differences. However, it is clear that the pattern when comparing all scenarios in the EMAC model is similar to that described above and that differences in the emission scenarios are the main driver of model-observation differences."*

• Section 5: The authors describe a "simplified approach" to estimate total and inorganic bromine from VSLS in the extratropical stratosphere. They assume a linear mixing of two air masses, wherein the fraction of bromine with tropical origin increases linearly with increasing $\Delta\theta$. The bromine mixing ratios are only evaluated once, at the tropical (TTP) and extratropical tropopause (ExTTP). This ansatz completely neglects the actual transport (horizontal as well as vertical) and therefore the further photochemical transformation (depletion) which VSLS undergo. Based on kinetics, one would expect the concentration of any of the VSLS (let's call it [A] for 'any') at the tropopause ($[A](0,0)$) to follow a power law:$[A](t,\theta) = [A](0,0)\cdot e^{-k\cdot(t(\theta)-t_0)}$,(1)with e.g.,$k \propto J_a + k_A(T)[OH]$. If we look at [A] at $\Delta\theta$ and a time t′ and assume the same linear superposition of air masses as the authors: $[A](t′,\delta\theta) =$

$$(f_{ex-trop}-1)[A]_{TTP} \cdot e^{-k \cdot \Delta t_a(\Delta \theta)} + (f_{ex-trop})[A]_{ExTTP} \cdot e^{-k \cdot \Delta t_b(\Delta \theta)} \quad (2)$$ Although the authors state"[...] for models it is necessary to have a realistic representation not only of chemistry but also of transport in the lowermost stratosphere.", this is not discussed, e.g. in a critical evaluation of the approach itself. Consulting Figure 4, the linear approach seems to be justified for [CH$_2$Br$_2$], whereas [CHBr$_3$] clearly follows a powerlaw. This is not discussed properly in the manuscript. The authors should reevaluate their approach and change it (if possible) or at least discuss it more thoroughly.

*We are not convinced that we have completely understood the point that the reviewer is making, why our ansatz should yield a linear relation between e.g. [CH2Br2] fraction and Δθ. The linear approach is only made for the* **total Bromine** *(sum of inorganic and organic bromine). For the organic part (which we measure, see also Fig. 4) no such linear relation is expected or made: First, there is no linear relationship between Δθ and transit time t', second, we assume two different input mixing ratios for tropical and extratropical fractions and third, the chemical lifetime is extremely dependant on the surroundings (photolysis rates, OH radical concentrations and temperature). As we believe that the reviewer (and also reviewer #3) may not have completely understood that our linear ansatz is only restricted to the total bromine, we have reworked the explanation:*

*To make the approach clearer we have added in the introduction to section 5:*

*"The inorganic bromine from SGI can be derived as the difference between the organic bromine in the source region (tropopause) and the organic bromine still observed or modelled at a certain stratospheric location."*

*And a little further down the same paragraph to motivate that we need some sort of assumptions on the relative importance of both source regions:*

*"As both regions show different levels or organic bromine source gases, the relative contribution of these source regions needs to be known to derive total bromine which entered the stratosphere and thus also inorganic bromine from SGI."*

• Figure 4: It is not clear whether the shown error bars refer to the standard deviation (σ) in each bin or to its standard error (s=σ/√(N)). Latter is a more reasonable choice. In any case, displaying data in such way does only make sense as long as the distribution in each bin is Gaussian. Otherwise, a violin plot may be more valid. Have the authors checked their distributions?

*The error bars given should be understood as a variability range, not as an uncertainty range. Therefore, we have not used standard errors, as we want to show the variability range of the observations. We also note that a violin plot would make this Figure (and others) unreadable, as a separate violin would be needed at each altitude interval. While, as most observational data, our data are not strictly Gaussian, we thus prefer to stay with the simple uncertainty ranges. We have removed some extreme outliers before the calculation, which are believed to be from contamination, therefore the data are not influenced by some extreme outliers. This has been checked also by comparting means and medians, which we found to differ by less than 5%, showing that the distributions are rather symmetrical. Therefore we have not changed the Figures, but we have added an explanation in section 3.1.*

*"We have checked the validity of using means to represent the data, by comparing means and medians. Differences were always below 5% of the mean tropopause values. We have thus chosen to use means throughout this paper. The uncertainties given in all Figures are 1 sigma standard deviations of these means."*

• Figure 7: Caption L11:"[...] extremely high values [...]"What is "extreme" in this context? Which exclusion criterion has been used? The manuscript doesn't provide any further information on this. Please elaborate on this matter in the appropriate section.

*These same data have also been omitted in other Figures and tables. We have therefore deleted this explanation in the caption of Figure 7 and added a statement in section 2.1 where the observational data are introduced:*

*"For this combined data set, some observations from the TACTS campaign have been omitted, where some extremely high values of VSLS (up to a factor of 10 above typical tropospheric values) were observed in the UTLS which are suspected of being contaminated. The source of the contamination is, however, unknown"*

*3 technical corrections*
*3.1 General*
Typesetting of the name "Ordóñez" in citations is not coherent and incorrect at times.

*We searched the entire document and made the spelling consistent.*

3.2 Specific

• P1L22:"Distributions [...] below the tropopause shows [...]"→"show"

*changed*

• P1L26:"A scenario which has emissions most strongly concentrated to low latitudes [...]". This sentence needs rephrasing – maybe: "A scenario with emissions mainly confined to low latitudes [...]"

*Changed as suggested*

• P3L8-10:"Two pathways for input of halogens from short-lived gases are dis-cussed: Source Gas Injection (SGI),where the halogen is transported to the stratosphere in the form of the source gases; and Product Gas Injection,where photochemical breakdown products of source gases are transported into the stratosphere, usually in inorganic form (i.e. Bry )."This sentence is not concise. Please rephrase.

*We have rephrased and broken up into two sentences:*

*"Halogen atoms can be transported to the stratosphere in the form of the organic source gas (Source Gas Injection (SGI)) or in the inorganic form as photochemical breakdown products of source gases (Product Gas Injection (PGI))"*

• P3L11-12:"While halogens transported into the stratosphere due to PGI are usually directly in a form available for catalytic ozone depletion reactions, halogens from source gases must first be released in the stratosphere photochemically."You may rephrase this along the lines: "Halogenes from product gases are readily available for catalytic ozone depletion reaction. Source gases have to undergo a photochemical transformation first."

*Nice formulation; we have adapted it as is.*

• P3L23:"The main source of brominated VSLS is believed to be[...]"Although orally excepted, one should refrain from the usage of "believe" in a scientific context. Please rephrase. You may use, e.g., "most likely" or "observations indicate".

*Rephrased to:*

*"Observations indicate that the main source of brominated VSLS is from oceans and in particular from coastal regions."*

• P3L41-43 and P4L1-3:"These observations are compared [...]"and"[...] these observations are [...] presented [...] and compared [...]". These sentences are almost identical. You may merge them into the latter.

*First sentence has been deleted and merged into second sentence, which now includes a mentioning of the different emission scenarios:*

*"Typical distributions of brominated VSLS derived from these observations are then presented in Section 3 and compared to model output from two different atmospheric models run with the different emission scenarios mentioned above in Section 4."*

• P4L7-11:"An isothermal channel uses [...]"and "The second channel [...] uses [...]"You may rephrase these in passive voice.

*Changed to passive voice: "An Electron Capture Detector (ECD) is used in an isothermal channel, in a similar set-up as used during the SPURT campaign (Boenisch et al., 2009;Boenisch et al., 2008;Engel et al., 2006) to measure $SF_6$ and CFC-12 with a time resolution of one minute."*

• P4L31,33:"mean age". Supposedly "mean age of air" as used later on in the manuscript.

*Yes, "of air" has been added*

Section 2: The authors only mention the spatial resolution of their models. It might be worth to mention the temporal resolution of the model output, too.•

*Added in the model description:*

*"The EMAC SD-simulations with 90 vertical levels, as described in detail by Jöckel et al. (2016), were integrated with an internal model time step length of 12 minutes and the data has been output every 10 hours from which the monthly averages on pressure levels have been derived."*

*And:*

*"The TOMCAT (Toulouse Off-line Model of Chemistry And Transport) model (Chipperfield, 2006;Monks et al., 2017) is driven by analyzed wind and temperature fields taken 6-hourly from the ECMWF ERA-Interim product. Here, the model was run with T42 horizontal resolution (2.8° by 2.8°) and with 60 vertical levels, extending from the surface to ~60 km. The internal model timestep was 30 minutes and tracers were output as monthly means."*

P6L28:"As in previous work [...]"→"works"•

*We think that previous work is correct here, as it used in a general way. However, we will be happy to modify this based on suggestions during the typesetting process. Unchanged*

P6L28-33: First of all, most of this is a repetition of the text written in the beginning of section 3 and can be dropped. "As in previous work [...]"and "However, we propose a somewhat different approach [...]"These two sentences are slightly contradicting. They can and should be merged along the lines: "We slightly diverge from approaches in previous works [...]"

*We have modified according to the suggestion:*

*"We slightly diverge from the coordinate system used to present zonal mean latitude-altitude distributions used in in previous work (e.g. Boenisch et al., 2011;Engel et al., 2006), where equivalent latitude and potential temperature where used as horizontal and vertical coordinates. We use equivalent latitude\* as a horizontal coordinate …. "*

• P6L36-38:"In order to ensure that this tropopause value is representative [...]"This sentence is too long and might be grammatically incorrect ("[...] when we have observations [...]"). Please rephrase.

*Split up into two sentences:*

*"In order to ensure that this tropopause value is representative also for the period of our observations, we compare the potential temperature of the campaign-based tropopause with the climatological tropopause. The campaign based tropopause has been calculated by averaging the tropopause at all locations for which observations are available during the campaign."*

• P7L7-9:"[...] which has a rather long lifetime in the cold upper troposphere and lower stratosphere (Hossaini et al., 2010) even quite deep into the stratosphere."This sentence is quite unclear and needs rephrasing.

*Rephrased and quantified:*

*"The distributions observed during the WISE and the TACTS campaigns show rather high levels of $CH_2Br_2$ in the lower stratosphere, with a depletion of only about 35% at 40-50 K above the tropopause. This is consistent with the rather long lifetime of $CH_2Br_2$ in the cold upper troposphere and lower stratosphere (Hossaini et al., 2010)."*

• P7L9-10:"[...] is strongly depleted [...]"Can you quantify this statement?

*Quantified to "The shorter-lived $CHBr_3$ is depleted by about 85% already at 20-30 K above the tropopause during the winter campaign PGS."*

• P7L10-12:"[...] flight levels [...]"You may refer to "flight altitudes" instead.

*Changed as suggested*

• P7L36-38:"This is most likely [...]"The authors may rephrase this sentence into two. The natural breakpoint would be"[...] with latitude. As [...]".

*We looked at this sentence and considered splitting it up. However, there is a causality in here, which would make it awkward to split this into two sentences. Unchanged.*

• P8L6-7:"[...] are based on the emission scenario by Warwick et al (2006) [...]was run with different emission scenarios (Ordoñez et al., 2012;Ziska et al.,2013;Liang et al., 2010)."The authors should mention which or the 8 scenarios by Warwick et al (2006) and which of the three by Liang et al. (2010) they have used.

*We have specified that scenario # 5 from Warwick and scenario A from Liang have been used in the model description.*

• P8L36:"overestaimtion"→overestimation: *changed*

• P9L4-6:"Because of the different chemical lifetimes [...] above 20 K above the tropopause [...]"This sentence might need some rephrasing. The authors may use∆θ>20 K.

*Good suggestion: we have rephrased as suggested:*

*"Because of the different chemical lifetimes of the two species, this results in a wrong vertical distribution of Bry with too high mixing ratios at ∆θ>20 K in winter and a much steeper vertical gradient in late summer."*

• P9L10-12:"In Order to [...] "This sentence is too long and not concise. Please rephrase.

*Rephrased to: "We additionally compare model data from EMAC simulations using all four emission scenarios (Graf, 2017) in order to investigate if the large deviation of the Warwick et al. (2006) emission scenario is due to the EMAC model or due to the specific emission scenario."*

• P9L12-13:"Note that these simulations [...]"You may drop "Note that"

*dropped*

• P9L23-24:"Again, we use equivalent latitude\*[...]"This has been stated several times by now. As the authors indicated by using"[a]gain [...]". They should consider dropping the whole sentence or the "again" therein.

*In order to make it clear that slightly different vertical coordinates are used for observations and models we rephrased as follows:*

*"While we use equivalent latitude\* as the latitudinal coordinate for the observations and θ\* as vertical coordinate, the zonal mean data are displayed as function of latitude and potential temperature θ for the model results."*

• P9L39:"The direct comparison of the distributions between the different model data sets is also interesting."This statement is redundant. Please rephrase, e.g."We will have a look at [...]"

*We have deleted this sentence.*

• P11L8:"In this Section [...]"Add a comma after "section". *done*

• P11L33: The indent of the equation number is incorrect. The authors may reduce the equation to$Br_{tot}(\Delta\theta) = Br_Y(\Delta\theta) + Br_{org}(\Delta\theta)$

*changed*

• P12L20:"Figure 14 compares [...]"There is a grammatically issue here. A figure cannot compare anything. Please rephrase using passive voice.

*Changed to passive voice.*

• P13L18:"have e a"→Remove the "e" in between. *removed*

• P13L30:"[...] a large dataset [...]"Can you quantify this?

*Changed to "We present a large dataset of around 4000 in-situ measurements of five brominated VSLS"*

• P13L36:"[...] which is line [...]"→"which is in line" *changed*

• P13L40:"in high latitudes"→"at high latitudes" *changed*

• P14L3:"[...] with large differences produced by the different emissions. "It maybe better to use "caused" in this sentence.

*Changed to caused*

• P14L3-5:"Overall, for CH2Br2[...]"Please consider rephrasing this sentence.

• Figure 1: The figure is too wide. Would it be possible to color code the different flights temporally?

*We suppose the Figure size could be reduced during the final typesetting. With respect to colour coding, this would mean also including a legend, which would make the Figure too busy. Not changed.*

• Figure 2: Is it possible to highlight the flight path displayed in Figure 3?

*As the flight shown in Figure 3 is rather arbitrary, we believe that colour coding it would give no additional information. Not changed.*

• Figure 4: Caption L8:"dotted"→"dashed"

*done*

• Figure 7: Referred to choice of colors under "general". Caption L10:"bene"→"been"•

*Bene has been changed to been; colours unchanged (see reply to general comment above)*

Figure 14: Please include the information given in L1 of the caption also in the legend of the plots.

*Including this information in the legend would actually make the legend very busy. We have instead changed the heading of the plots and included "dashed" in parentheses behind the total bromine to make this clear.*

• Figure 15: The x-axis label of the lowermost panel seems to be incorrect (compared to the others in the figure)→"Br total and Bry from VSLS".

*We suppose this comment refers to Figure 14; corrected.*

Caption L1:"[...] at $\Delta\theta$ of 40 K [...]"→"at $\Delta\theta$ = 40K"

*done*

**Reviewer #2, Rafael Fernandez**

The paper presents a complete set of carbon-bonded VSLS source gases (SGs) measurements performed in the Northern Hemisphere mid-latitudes using a GC-MS instrument on board an aircraft, as well as a comprehensive comparison of the observations with a complete set of model simulations oriented to evaluate the contribution of tropical and extra-tropical injection of VSLS to the lowermost stratosphere. The main results of the work are: i) the troposphere-to-lowermost-stratosphere transport of VSLS SGs through the extra-tropical tropopause is larger than that occurring within the tropical tropopause; ii) the contribution of both tropical and extra-tropical VSLS injection must be considered in order to reproduce the VSLS abundance within the mid-latitudes lower stratosphere below 400 K; and iii) the models and inventories used in this work show certain limitations in reproducing the VSLS reactive transport and estimate the release of inorganic bromine (Bry) in the NH lower stratosphere. The paper also includes a seasonal, latitudinal and vertical analysis of VSLS abundance in the UTLS.I found the paper very interesting and very well organized, presenting results in a clear and comprehensive format, and including interesting and constructive discussions. It is worth noting that even when the altitude/latitude-dependent observations itself would be worthwhile to be published, the authors have decided to go forward and present a comprehensive model-observation inter-comparison, which contributes to improves the general understanding of the reactive-transport efficiency of VSLS species within the UTLS. In particular, I found very descriptive and intuitive

the vertical coordinate system ($\triangle\theta$ and $\theta^*$) they used to represent all results relative to the altitude of the tropopause, which allows a consistent description of the vertical and latitudinal decay of VSLS once they are injected to the stratosphere. At the very end of the paper, a simplified approach(eq. 4) is used to estimate the total amount of inorganic bromine (Bry) released byVSLS within the mid-latitude lower stratosphere, highlighting the major importance ofproperly reproducing these Bry levels in model simulations oriented to determine the ozone impact of VSLS. Having said this, I believe that the manuscript posses a handful set of specific issues and many technical details (including figures and tables captions) that must be corrected before final publication.

*Major Concerns:*

1. I would like to start mentioning that most of the "important questions" that came to my mind while I was reading the manuscript had already been responded (i.e., as I moved forward with the lecture and reached Section 5). Even when this should be taken as a mainly positive comment, it also implies that some of the analysis/discussions given at the end could be (at least partially) shifted to earlier sections, to help the reading and support the analysis. For example: Sections 3 is concentrated on carbon-bonded VSLS mixing ratios close and above the tropopause in the lower stratosphere, but there is only a brief mention and sideway comparison with the SGI values compiled in the last WMO report (P7,L27). The reader needs to wait until Table 5 is presented (Section 5, P12) to reach a complete discussion and comparison with WMO values, and it is only at this point that the importance of the model-observation inter-comparison (presented in Section 4) becomes evident. In doing so, note that Table 5, which pro-vides values for VSLS SGs within the tropics and extra-tropics, is introduced in the only section of the paper focused on PGs (Section 5).

*We have referenced Table 5 already in section 3.3. and 4.3. where we point to the tropical values from the WMO 2018 report. We have also, as mentioned by Rafael Fernandez in a specific comment included the tropical values from WMO 2018 in Figure 6 and also in Figure 13. We hope that this brings forward the point that extratropical values, in particular during winter are higher than tropical values.*

*Added the following in the text in section 3.3. (latitudinal gradients) after introducing the tropical WMO values.:*

*"These upper TTL values have also been included as reference Figure 6 (see also Table 5)."*

*Added/modified the following in section 4.3.:*

*"In order to investigate if the models are able to represent the latitudinal gradient in upper tropospheric mole fractions, we compare the observed extratropical mole fractions of the brominated VSLS in the upper troposphere (section 3.3) and compiled tropical observations (Engel and Rigby, 2018) with those determined from the different model setups."*

*And referenced Table 5 in section 4.3.*

2. Observations from TACTS and WISE campaigns have been merged into a unique dataset (WISE_TACTS) because they were performed during the same seasons. Even though I found this procedure correct, I wonder if the authors have analyzed this data separately to evaluate if there is at least any glimpse of VSLS SGs trend within the NH-UTLS (there are~5 years between both campaigns). The authors declare they combined both dataset based on „general" observational evidence (P3,L29), but I think they should justify this procedure by evaluating specifically their unique and novel dataset (P4,L29).

*We have not analysed the data set with respect to a possible trend, as we believe that there are too many uncertainties associated with this and that the variability is far too high to detect trends, which are expected to be very small. However, we have decided to add an appendix to the paper, which contains plots of the observations (eq. to Figures 4, 5 and 6) for the WISE and TACTS campaign separately.*

3. Authors should be really careful and consistent when using the wording "total bromine". Until Section 5 is reached, only carbon-bonded (i.e., organic) bromine is considered, and total bromine is referred as the sum of CH2Br2 + CHBr3 + minor VSLS(P6,L2; P8,L15). But later on Sections 5 and 6, "total bromine" points out to the sum of Br_org + Br_inorg (P11, Eq. 1). Please, be consistent and refer to "total organic bromine" and/or "total bromine" whenever appropriate.

*The reviewer is absolutely correct in this statement. We have gone through the manuscript and have made sure that when total refers to the sum of bromine from all 5 species, we always include organic (total organic bromine) or inorganic (total inorganic bromine) with it, in order to separate it from total bromine as the sum of organic and inorganic bromine.*

4. Although the vertical profiles for CH2Br2 and CHBr3 are analyzed in detail, there are no mentions regarding the "error bars" presented in the figures (e.g., P6,L20; Fig.4 caption). Do the error bars correspond to 1-sigma or 2-sigma? Besides this, the authors should explain why the vertical error bars for △θ are not the same for the different bins, as well as why the error bars for CHBr3 are larger than for CH2Br2? Is this only due to the shorter lifetime of CHBr3, which shows quite different vertical profiles de-ending on the exact latitude within the 40-60◦ bin (as observed in the latitude-altitude cross-sections), or could this also be attributed to differences in their regional source strengths? Note that the variability is only considered in (P7,L34; Fig. 6) when addressing how the VSLS tropopause abundance changes with latitude, but I think that a more complete comparison of this latitudinal and vertical variation within the 20-40◦N bin should be provided (at least the two major VSLS).

*The error bars denote 1 sigma variability. The reason for the much larger uncertainty range in CHBr3 is indeed the shorter lifetime, which implies stronger gradients and overall larger variability.*

*The meaning of the error bars (vertical and horizontal) is now explained in the manuscript in section 3.1, also in relation to a remark by reviewer #1 with respect to using sigma (Gaussian distribution):*

*We have checked the validity of using means to represent the data, by comparing means and medians. Differences were always below 5% of the mean tropopause values. We have thus chosen to use means throughout this paper. The uncertainties given in all Figures are 1 sigma standard deviations of these means, both for the vertical and horizontal error bars.*

*With respect to the larger relative uncertainties for CHBr3 and also the different vertical error ranges we have explained towards the end of section 3.1.:*

*The short lifetime and strong vertical gradient of CHBr3 is also reflected I the largest relative variability (see Tables 3 and 4).*

5. P8,L18: "The contribution from these mixed bromochlorocarbons to total VSLS bromine are typically on the order of 20%, while about 80% of total VSLS bromine in the upper troposphere and lower stratosphere is due to CH2Br2 and CHBr3".→Are this percentages computed using WISE_TACTS and PSG data? Have you compared this findings with other studies (i.e., Fernandez et al., 2014)? Have you thought about presenting combined results of your observations of the sum of these minor_VSLS into a figure or table? (currently only results for the sum of CH2Br2 + CHBr3 + minor_VSLSare given)? As minor_VSLS posses in some cases lifetimes larger than CH2Br2, this information could be useful for future studies. The importance of the minor VSL contribution becomes also evident in Section 5 (P13,L11) when the overall contribution from longer-lived VSLS to Bry is discussed.

*The mean values just below the tropopause and also at 30-40 K above the tropopause are given for all 5 VSLS species in Tables 3 and 4. We have added the plots as for Figures 4 and 7 for the minor VSLS in the appendix. The 20% contribution of minor VSLS is consistently derived from our observations and from the WMO 2018 compilation (Table 5). This is now stated in the text:*
*"This relative contribution of 20% from minor VSLS is found in our observations (Tables 3 and 4) as well as in the values compiled in Engel and Rigby (2018) (see Table 5) and is slightly larger than that derived e.g. in Fernandez et al. (2014)."*

*Specific Comments:*
1. Title and Abstract: Shouldn't the title be more specific on the extra-tropical (or the"tropical vs. extra-tropical") contribution of VSLS bromine to the UTLS?? In addition, the abstract has an excessive focus on the results obtained with the different models and emissions scenarios (including specific statements for some of the scenarios giving the best and worst agreement). I would expect the abstract to focus on the contribution of tropical vs. extra-tropical contribution to SGI, and in any case to provide a rough estimate of the relative contribution of each of these two pathways to the over-all organic and inorganic lower stratospheric bromine (and providing only a general mention to the similarities and discrepancies between models and observations and its dependence on latitude and season).

*We added extratropical in the title and slightly modified the abstract to include and explanation of the importance of the measurements ("Why aspect mentioned by reviewer #1). We believe that the aspect of tropical vs. extratropical SGI is already mentioned in the abstract: "Distributions of the five source gases and total organic bromine just below the tropopause show an increase in mixing ratio with latitude, in particular during polar winter. This increase in mixing ratio is explained by increasing lifetimes at higher latitudes during winter. As the mixing ratio at the extratropical tropopause are generally higher than those derived for the tropical tropopause, extratropical troposphere-to-stratosphere transport will result in elevated levels of organic bromine in comparison to air transported over the tropical tropopause." Therefore we did not add anything here, but hope that the explanation of Why added more context. The issue of fractional contribution of tropical and extratropical air masses is only treated as a sensitivity study. We have pronounced these issues more specifically in the abstract now:*

*"In a sensitivity study we find maximum differences of a factor 2 in inorganic bromine in the lowermost stratosphere from source gas injection derived from observations and model outputs. The discrepancies depend on the emission scenarios and the assumed contributions from different source regions. Using better emission scenarios and reasonable assumptions on fractional contribution from the different source regions, the differences in inorganic bromine from source gas injection between model and observations is usually on the order of 1 ppt or less."*

2. Section 3 (P5): The first paragraph of section 3 describes the spatial and vertical coordinates used for representing measured and modeled data. In my opinion, the selection of θ, △θ and θ* variables really improves the analysis and interpretation of the results. However, I believe the initial description of how these variables are computed is not clear enough, and the reader needs to go back and forth between figures and text to completely catch up the difference among them. For example: a) △θ is mostly used for vertical profiles figures, whereas θ* is used for latitude-altitude cross-sections (which is not clearly mentioned in the text); b) there are at least 2 or 3 places where △θand θ* vertical coordinates are defined, and in some cases slight differences on the definitions are observed (P5,L31 and P6,L32). In particular specify if the vertical coordinate is computed above the "local" tropopause "for each latitude" and how is it added to the "climatological" tropopause.

*The use of θ* in the latitude altitude cross sections is now clearly mentioned in the text and also we have deleted the additional explanation of θ* given in section 3.2. and referenced the definition of θ* at the beginning of section 3. We also specified that the local tropopause is the reference for △θ at the beginning of section 3. In addition we have explained at the beginning of the last paragraph of section 3.2. again that θ* is used here (it just stated potential temperature before).*

*Explanation of the different vertical coordinates at the beginning of section 3:*

*"These are potential temperature θ, potential temperature above the local tropopause Δθ, and finally a coordinate we refer to as θ*, which is calculated by adding the potential temperature of the mean tropopause to Δθ."*

End of section 3.2., explanation of the coordinates used:

*"The data have been binned in 5° latitude and 5 K intervals of the modified potential temperature coordinate θ*."*

3. P11,L2-5: "Most importantly, the overall levels, especially in the low latitudes, are much higher than our observations and also much higher than the tropical observations compiled in the WMO report (Engel and Rigby, 2018). This will result in too much VSLS bromine being simulated in the stratosphere, and therefore also in a misrepresentation of the input to the lowermost stratosphere via the different pathways".→In addition to the general description focused on the 340-400 K range, I found interesting that poleward of 40∘ and below 320 K (see Fig. 9-10) there is a negative model bias for CH2Br2 mixing ratio exactly at the extra-tropical tropopause, while at the same time there is a positive model bias for CHBr3. This is not mentioned nor explained in the text.

*We thank Rafael Fernandez for pointing towards this feature. We have no good explanation for this, with the exception that for the models, the tropopause sometimes seems to be more "penetrable" than for the observations. It can also not be excluded that this is somehow related to the way that the models have been averaged. We decided to add this feature, without trying to explain it. This may indeed be an interesting aspect for further studies.*
*Added text at the end of section 4:*
*"We also note that poleward of 40°N and below 320 K (see Figures 9 and 10) there is a small negative model bias for CH2Br2 at the extratropical tropopause when using the scenarios by (Liang et al., 2010) and (Ordóñez et al., 2012). At the same time the model simulations using these two scenarios yield a substantial positive bias for CHBr3 in the same region. This will result in a misrepresentation of the input of brominated VSLS source gases to the lowermost stratosphere via the different pathways."*

4. P11,L28: The simplified approach considering fext-trop and ftrop" is very intuitive and helps to visualize the contribution from tropical vs extra-tropical bromine from VSLS, but it would be useful to provide at the end a conclusive sentence of which are the most probable fractional contributions from tropics and extra-tropics to the overall bromine in the UTLS. Certainly, concurrent Bry measurements would be required to close the whole bromine budget, but at least from Figure 15 (△θ= 40 K) it seems that fext-trop values close to 20-40% produce the closest agreement between model and observations. However, on the analysis presented for Figure 14, fext-trop is 60% at this height, isn't it? I suggest to expand the analysis and discussion of fext-trop and ftrop" (you only dedicate a few lines to this subject at the end of Section 5). Finally, why are not equivalent results for WISE_TACTS provided in Fig. 15?

*Firstly, Rafael Fernandez is correct in pointing out that concurrent Bry measurements might be a valuable tool in investigating this. This could potentially form the basis for further studies. We would like to emphasize again, that this is a sensitivity study, and that a good agreement with the model may not be interpreted as the right fractional contribution, as it is more likely that several errors may balance each other. We would thus not use this to discuss the right fractional input.*

*With respect to the WISE TACTS campaign: Similar Figures to Figure 14 and 15 are now available in an appendix.*

5. P12,L28-29: "The larger Bry derived in the model calculations above 60 K is caused by the higher total bromine values from CH2Br2, which are caused by the higher CH2Br2 levels at the tropical tropopause in comparison to the observations." P12,L32-33: "In the lower part the discrepancy is more due to higher simulated CH2Br2 in the lowermost stratosphere than found in the observations"→This could "partially" be the reason, but it could also be due to using an improper fext-trop value at this specific vertical level and/or due to the simplified linear approach used. Please elaborate on this.

*The text is a discussion of the differences under these assumptions. Different assumptions would yield different results, as discussed in the sensitivity study in Figure 15. We have made this clearer by rephrasing:*
*"Under the given assumptions about fractional input, the larger $Br_y$ derived in the model calculations above 60 K is caused by the higher total bromine values from $CH_2Br_2$, which are caused by the higher $CH_2Br_2$ levels at the tropical tropopause in comparison to the observations."*
*"Under the given assumptions about fractional input, the discrepancy in the lower part is more due to higher simulated $CH_2Br_2$ in the lowermost stratosphere than found in the observations."*

6. P14,L29: "we have shown that there will be significant differences in stratospheric Bry depending on the emission scenario, which can be as high as 2 ppt, corresponding to a difference of a factor 2 relative to observation-derived values".→Being this sentence included in the conclusions (and also mentioned in the original abstract), I suggest informing not only the largest (i.e., worst) difference, but also the minimum model-observation differences, as well as the range of model bias results for the models which show a better performance.

*It is rather difficult to explicitly state the best agreement, as by coincidence, this may be close to 0 at some places. The reason that this value was given is simply to mention the order of magnitude of possible biases. We have restricted the statement of the 2 ppt to Warwick et al. (2006) and added:*
*"Typical differences in $Br_y$ when using the other scenarios are on the order of 1 ppt."*

*Technical Corrections:*
P1,L17: "The instrument is extremely sensitive due to the use of chemical ionisation,allowing detection limits in the lower parts per quadrillion (10 -15 ) range".→Is this information of major importance to be included on the abstract? Consider also including the GS-MS acronym in the preceding sentence.

*We would like to keep the sentence on the high sensitivity, as the use of chemical ionisation in GC-MS is rather special. We have included the acronym GC-MS in the first sentence.*

P1,L35: "Depending on the underlying emission scenario, differences of a factor 2 in reactive bromine derived from observations and model outputs are found for the lowermost stratosphere, based on source gas injection."→Consider rephrasing, and also mentioning the range of agreement of models (see comment above).
*Rephrased to:*
*"Depending on the underlying emission scenario, maximum differences of a factor 2 in inorganic bromine from source gas injection derived from observations and model outputs are found for the lowermost stratosphere."*
*And added the following to the abstract:*
*"Using better emission scenarios, the differences in inorganic bromine from source gas injection between model and observations is usually on the order of 1 ppt or less".*

P3,L30: "A further future increase has been projected"→A future increase of VSLS emissions has been suggested"

*changed to suggested*

P3,L38: "In order to investigate the regional variability of bromine input into the lowermost stratosphere and the inorganic bromine loading of the extratropical lowermost stratosphere, we have performed a range of airborne measurement campaigns ..."→You explicitly mention "inorganic bromine" but not "organic bromine" in a sentence focused on the novel measurements dataset. Consider revising, as you've only measured carbon-bonded species, and only inferred inorganic bromine.

*The part of the sentence mentioning inorganic bromine has been deleted.*

P3,L42;P4,L3: Please specify which are "the implications" you are pointing at.

*Changed from implications to differences*

P4,L5: Consider changing the subtitle to "Instrumentation and Observations"

*Changed as suggested*

P4,L22: Check for consistency between the year of the TACTS campaign between the text (2011) and table 2 (2012).

*It's 2012; corrected*

P4,L29: "covered a similar time period and latitude range"→you mean same seasons, consider rephrasing.

*Changed from time period to time of the year.*

P5, L2: "ESCiMo (Earth System Chemistry ntegrated Modelling)"→Integrated
*corrected*

P5,L12: TOMCAT acronym already defined above (P5,L2). *Deleted*

P5,L7;P5,L13: If both EMAC and TOMCAT are driven by exactly the same ECMWF ERA-Interim reanalysis data, this should be mentioned explicitly. This will help to override additional uncertainties regarding differences between models. Also, although EMAC can be run as a CCM model, it should also be clear that for the current SD simulations, the model behaves like a CTM.

*We added the following:*
*"For the results presented in this paper, EMAC was operated in the so-called specific dynamics mode, in which the synoptic scale meteorology is relaxed towards meteorological reanalysis data."*

P5,L17: "Emitted VSLS (CHBr3, CH2Br2, CH2BrCl, CHBr2Cl and CHBrCl2) are destroyed by reaction with OH and photolysis in the model"→I assume this is also the case for the EMAC model described in the preceding paragraph. You should explicitly mention this to avoid confusion.

*This is of course true; the main reason for this sentence was to point towards the source of the kinetical data used in TOMCAT. We have rephrased as follows:*
*"Chemical breakdown by reaction with OH and photolysis in the model for all VSLS ($CHBr_3$, $CH_2Br_2$, $CH_2BrCl$, $CHBr_2Cl$ and $CHBrCl_2$) are calculated using the relevant kinetic data from Burkholder et al. (2015)."*

P5,L24: Although the paper reads perfectly well using the θ vertical coordinate, it
should be mentioned at least during the model description which are the equivalent altitude/pressure values for the tropical/extratropical tropopause θ levels used in this work.

*As we are using potential temperature in the entire manuscript, we think that adding an altitude information in km would actually link to a vertical coordinate which is not used in the paper at all. Therefore we have not derived this information and not included it.*

P5,L36: "we have also binned the data in potential temperature in 10 K potential tem-perature intervals"→repetitive

*No, this is not repetitive, as the second phrase explains the potential temperature binning. We have rephrased for more clarity:*
*"For potential temperature binning, the 10 K bins have been chosen ranging from 40 K below the mean tropopause to 100 K above the mean tropopause."*

P5,L38 and elsewhere: "relative to the mean tropopause observed during the cam-paigns"→in many places the authors make reference to "the campaigns" in a general meaning, when I understand they are pointing out to "each dataset" obtained during the campaign, and not the campaign itself.

*We have replaced campaign by measurements, even though we feel that this is rather the same thing.*

P5, L38: "The results are presented ..."→consider rephrasing the whole sentence, as it is very difficult to understand. It should also be mentioned at least once that whenever you mention winter, spring or fall, you are always pointing out to "boreal" seasons. It is not necessary to repeat it all over the text, but I only found it mentioned properly once in the conclusions (P13, L39).

*We put the seasons for which the measurements are representative in parentheses to improve the readability of the sentence. We also added the reference to the Northern Hemisphere in the parentheses. It now reads:*

*"The results are presented for the two main VSLS bromine source gases $CH_2Br_2$ and $CHBr_3$, averaged over equivalent latitude\* of 40-60°N in Figure 4 for PGS (Northern Hemispheric winter) and the WISE_TACTS combined data set (Late summer to fall, Northern Hemisphere)."*

P6,L15: "again in line with their atmospheric lifetimes, which generally decrease with an increase in the bromine atomicity of the molecule". Atomicity should also be used in
P7,L25.

*The term atomicity is not very widely used and may not be familiar to many readers of ACP. We therefore prefer to stay with the term "number of bromine atoms in the molecule".*

P7,L10: "The shorter-lived CHBr3 is strongly depleted already about 20 K above the tropopause"→Based on Figure 5, this is only the case during PGS, but not for WISE_TACTS during the summer. Could this be related to stronger convective transport during the summer? Please explain.

*This is more likely an effect due to the strong diabatic descent in the vortex during winter. We quantified the depletion (see comment from reviewer #1) and added a statement about the effect mainly occurring during winter. The explanation that this is probably related to the descent in the vortex during winter was already presented in the following. We have not added more explanation, as the restriction of the statement to the winter campaign relates the low values to the following explanation.*

P7,L20-22: "In order to ..."→there are many sentences that begins with this wording. Although I found it correct, please avoid it using more than once within a paragraph(i.e., here it is used in two consecutive sentences).

*Second sentence has been changed to:*
*"All data in a range of 10 K below the local dynamical tropopause have been averaged to characterize the upper tropospheric input region."*

P7,L23: "Again, for the tropospheric data, standard latitude has been chosen, while equivalent latitude was used for all data with△θabove zero"→This got me confused:
Figure 6 focus on extra-tropical tropopause values (averaged considering bins 10 K below the local tropopause).

Wouldn't this correspond to negative△θ?

*That is correct and indeed for the upper tropospheric data we have used latitude and not equivalent latitude. The axis description of Figure 6 is therefore lat. and not eq. lat. We have rephrased to make this clearer:*

*"For these upper tropospheric data, standard latitude has been chosen and not equivalent latitude as for the stratospheric data."*

P7,L29: explicitly point at Tables 3 and 4.

*done*

P7,L30 and Figure 6: Could the "negative latitudinal gradient" observed for WISE_TACTS be somehow unrealistic/largely-biased because of using a small amount of measurements below the tropopause at larger latitudes? Or because of any type of cos(lat) averaging factor?

*The reviewer is correct that the amount of data north of 60° lat is much smaller than south of 60° N for the TACTS and WISE campaigns. However, it is also well understandable that the mixing ratios to not show a further decrease to the north, as due to longer insolation during NH summer the lifetimes are expected to decrease again. This discussion is taken up a few lines further down, where we note that: "Nevertheless, there is a clear tendency for an increase in tropopause values with latitude, particularly during Northern Hemisphere winter. This is most probably related to the increase in lifetime with latitude, as especially during the wintertime PGS campaign the photolytical breakdown in higher latitudes is slower than in lower latitudes."*

P7,L34: "derived around the tropopause"→wouldn't it be "below" (-10 K) the tropopause.

*Changed to* *"in the upper troposphere".*

P8,L1-5: Sentence is too long. Please rephrase.

*The sentence has been broken down into two sentences:*

*"As bromocarbons are an important source of stratospheric bromine, it is worthwhile to investigate if current models can reproduce the observed distributions shown in Section 3. This is a prerequisite to realistically simulate the input of bromine from VSLS source gases to the stratosphere, but also the further chemical breakdown and the transport processes related to the propagation of these gases in the stratosphere."*

5  P8,L13: "Here we compare vertical profiles, geographical distributions and latitudinal gradients between our observations and the model results".→What do you mean by "geographical distributions"? 20◦-40◦ bin?? If not appropriate, please remove.

*Geographical distributions has been replaced by latitude-altitude cross section, which is clearer.*

P8,L23: "about 40 K"→"~40 K"

10  *In this case, we would prefer to stay with the written "about", unchanged.*

P9,L4: "Using the Ziska et al. (2013) emission scenario, the overestimation of CH2Br2 and the underestimation of CHBr3 tend to cancel out, resulting in a reasonable agreement in total VSLS bromine. Because of the different chemical lifetimes of the two species, this results in a wrong vertical distribution of Bry with too high mixing ratios above 20 K above the tropopause in winter and a much steeper vertical gradient in late summer."→First, in the

15  initial sentence it should also been explicitly mentioned that, in addition to the CH2Br2 overestimation and the CHBr3 underestimation, the contribution form "minor VSLS" which is also considered for the total "organic" bromine results is based on the Ordoñez inventory...adding an additional uncertainty to the different contributions that "cancel out" each other. Second, What do you mean by "vertical distribution of Bry" in this context?. Please be careful when pointing out to the inorganic or organic bromine in this section.

20  *First: changed to: "When adding the contribution from minor VSLS based on the scenario by Ordóñez et al. (2012) this results in a reasonable agreement in total VSLS organic bromine.*

*Second: this sentence has been deleted.*

P9,L23: Fig. 9 to 12 instead of 9 and 10? *Yes, changed*

P10,L6: "much lower"→please be more specific

25  *Added difference: "Our low latitude observations from HALO during late summer and fall (WISE and TACTS) and the values compiled in the WMO 2018 report for the tropics (Engel and Rigby, 2018) are lower by about 0.3-0.5 ppt than the values of CH2Br2 in the tropics using the Warwick et al. (2006) and Ziska et al. (2013) emissions."*

P10,L15: "Therefore, we compare the observed mole fractions of the brominated VSLS in the upper troposphere with those determined from the different model setups, in order to investigate if the models are able to represent

30  the latitudinal gradient in uppertropospheric mole fractions."→please rephrase.

*Rephrased: "In order to investigate if the models are able to represent the latitudinal gradient in upper tropospheric mole fractions, we compare the observed mole fractions of the brominated VSLS in the upper troposphere with those determined from the different model setups."*

P10:L17: "or respectively modelled (EMAC)"→please rephrase (see comment on respectively below)

35  *We have had this checked by a native speaker, who had no objections to the way that respectively is used here. In any case, we expect that this will also be checked once more during the typesetting process. Unchanged*

P10,L32-37: The text goes back and forth a couple of times between wintertime results and summertime results. It would be simpler to describe all results for one season before moving to the other.

*We considered this. However, the logic is to start with the observations and then to compare with the models.*
40  *Unchanged.*

P10,L36: Here and elsewhere..."extremely high"→what do you mean by extremely? Wouldn't just "larger" be enough?

*Changed to larger*

P11,L12: "is expected to add more bromine on top of SGI"

45  The reference to SGI has been added and we have made two sentences out of this one sentence: "The input of bromine into the stratosphere in the inorganic form (product gas injection) is expected to add more bromine in

addition to the SGI discussed here. However, PGI cannot be investigated with the source gas measurements presented here."

P11,L14: "imply for the total bromine and inorganic bromine"→you mean total organic bromine or total (organic + inorganic) bromine? Or both? Please see the general comment above.

5 *We have checked the usage and made it more consistent: throughout the manuscript we have made sure that the term total is always used together with organic if it only refers to total organic bromine from VSLS. Especially here, we have specified in parentheses behind total bromine that it refers to the organic and inorganic bromine*

P11,L18: Too many "and" within a single sentence. Please rephrase

*Rephrased to two sentences:* We have shown in Sections 3 and 4 that the organic bromine around the tropopause
10 shows significant variability and also latitudinal gradients. We have also shown that large differences between the different model setups and observations are found.

P11,L23: "No studies on mass fractions are available for the campaigns discussedhere, so we will rely on previous studies for these fractions."→I do not understand the rationale for including this sentence. Please make it clear or remove.

15 *In order to make the meaning clearer we have extended the sentence: "No studies on mass fractions are available for the campaigns discussed here,* so we will rely on previous studies for these fractions as discussed in the introduction to estimate the fractions of tropical and extratropical air in the lowermost stratosphere."

P11,L24: "order of magnitude difference" is normally used to point at scaling factors like 10, 100, 1000. Please rephrase.

20 *Changed to "how much"*

P11,L41: Here and elsewhere. "at the tropical, respectively extratropical (40-60◦N) tropopause,"→It is very unfamiliar for me the way the "respectively" sentences have been written throughout the text. I suggest replacing them by "at the tropical and extratropical (40-60◦N) tropopause, respectively". See also P12,L19-L23.

*We have had this checked by a native speaker, who had no objections to the way that respectively is used here.*
25 *In any case, we expect that this will also be checked once more during the typesetting process. Unchanged*

P12,L6: "by averaging the model respective observations"→This sentence is sense-less, in any case "the model results"

*We have changed to model results and specified that extratropical input values are meant.*

P12,L14: fext-trop decreases with altitude, not increases.

30 *Thank you, you are right. Changed.*

P14,L6: "with a downward revision"→do you mean shifted to the lower edge of the range of emissions?

*We refer to the range being reduced and some older very high emission estimates being judged unrealistic in the last WMO assessment.* The sentence has been changed to :

*a recently proposed revision of the best estimate of global CH2Br2 emissions towards the lower edge of previous*
35 *estimates*

P14,L26: "The bromine budget in the lower stratosphere will depend on the relative fraction of air from the tropical and extratropical tropopause. The relative contribution of extratropical air will decrease with altitude and should reach zero at about 400 K potential temperature.."→is it the future (will) tense appropriate here??. In any case, I believe the authors are pointing at a decrease with altitude and not latitude here (if not, please explain the idea and
40 make it clear).

*Changed to present tense to emphasize that this is a general feature and, yes, we refer to altitude.*

**Tables and Figures:**

45 Most of Figures and Table Captions include the "long-name" of each of the Cam-paigns instead of just providing their "short-name"/acronym (PSG=POLSTRACC+GW-CYCLE+SALSA, TACTS_WISE, HALO, ECMWF, etc.), which is not only simpler, but also more familiar to everyone. Using the short-name version will certainly improve the captions readability. Also, there is no need to define ppt each time you use it (only once at the beginning within the text is enough).

50 *Done in all tables and Figures.*

All Tables: Consider using a "one line title" at the top of each table, and then provide all specific information regarding the season, specific campaign, altitude/latitude range,etc. as footnotes on the table.

*Due to using acronyms in the table headers these have become much shorter. We would thus prefer to stay with the original presentation.*

5 All multiple-panel Figures should indicate, in addition to the (letf, PSG) and(right,WISE_TACTS) information, that results are provided for (top, CH2Br2), (middle, CHBr3) and (bottom, total organic bromine). (This example was based on Fig. 13, and should be adapted to the specific figure).

*Done by including top/middle/bottom in parentheses behind the substance name.*

Table 3 and 4: What does "TP" stands for (tropopause)?. Why Table 3 provides values for TP + 30-40 K, while 10 Table 4 only for TP + 40 K? Define what does the stdev. Stands for and how is computed? Wouldn't it better to provide the stdev. value with a +/- sign (0,55 +/- 0,09) ppt within the mole fraction column?

*The heading in Table 4 has been updated to read TP + 30-40 K (which it is, similar to the column in Table 3). The stdev. column has not been included in the mixing ratio column, as this column shows the average standard deviation in the 4 lowest stratospheric bins, with the purpose of showing that this is significantly reduced when* 15 *using △θ. In order to explain better what this column means we have included the following explanation in the table headers for Table 3 and 4:*

*The 10K bin standard deviations in the table represent the variability averaged over the four lowest stratospheric bins.*

*The abbreviation TP has been introduced in the heading of both tables. Also we have changed the column heads* 20 *to θ and △θ.*

Table 5: Have you used the annual mean or the seasonal mean for computing the tropical tropopause values?? (P12,L6). In case a seasonal mean has been used, please specify which months have been used for the model output. Replace bromide by bromine. Rephrase respectively. Explicitly indicate that the Br_ext-trop and Br_trop are for (△θ= 0). Indicate that ML stands for Mid-latitudes, which you called extra-tropics throughout the text. Define 25 explicitly in a table footnote the△θ, latitude range and any other relevant information that has been used for computing the values presented in Table 5.

*Bromide has been changed to bromine.*

*The explanation of the tropical values is detailed in the table heading: For the Tropics, annual average for the years 2012 to 2016 have been calculated between 10°N and 10°S in a potential temperature range from 365 to 375 K.* 30 *The tropical values for the observations are from the observations compiled in the 2018 WMO report (Engel and Rigby, 2018) in the tropics between 365 and 375 K potential temperature.*

*The latitude ranges for the extratropics are also stated as 40-60°N.*

*ML has been replaced by extratropics.*

Figure 4: What are the horizontal and vertical bars? 1-2 sigma? Also, in the text it would be good to explain within 35 the main text why for WISE_TACTS there are some points for which the mixing ratio as a function of△θ can be computed (△θ= 100), but not for the θ coordinate (θ= 420 K).

*The following has been added to explain the error bars in the caption:*

*Both vertical and horizontal error bars denote 1 sigma variability.*

*The reason why sometimes sampling in △θ reaches higher values is that somtimes the tropopause is lower than* 40 *the climatological tropopause. A sentence explaining this has been added in the main text:*

*Due to this different sampling a higher range in Δθ is achieved than in θ, as the actual tropopause altitude varies.*

Figure 5: Consider introducing a dashed/dotted vertical line indicating the 40-60∘boundaries (and any other important latitude) used for the extra-tropics vertical pro-file computation.

*We have considered this, but have decided against it, as we believe that this would highlight an area which is* 45 *actually arbitrary and is only motivated by the comparison plots for the vertical profiles. No changes.*

Figure 7: Consider reducing the length of the caption, and moving some of information at the end of the caption to the main text.

*The caption length has been reduced by only using the acronyms of campaigns and models.*

Figure 8: Specify output for this simulations is not in SD mode as for the other simulations (here and in the main 50 text).

*This was actually a mistake in the original manuscript. These EMAC simulations are in SD mode, but not for the correct time period. We modified the Figure caption:*

*"Model results are for nudged simulations of EMAC but do not cover the time period of our observations."*

*And in the text we have extended the following sentence:*

*Note that these simulations are only available for the time period up to 2011.*

Figure 9-12: Consider including vertical dashed/dotted lines as suggested for Fig. 5.

*We have considered this, but have decided against it, as we believe that this would highlight an area which is actually arbitrary and is only motivated by the comparison plots for the vertical profiles. No changes.*

Also...for the model (right) panels: Why there are some "empty/blank" boxes within tropical lower stratosphere above 400K? Is that because of the vertical model resolution and/or upper limit of the models?

*The reason for the blanc boxes is the vertical resolution of the model, as values are not interpolated, but rather shown as blanc where no values are available. A sentence has been added to explain this in the Figure captions:*

*Boxes in which due to the vertical resolution of the model no values are available are left blanc.*

Figure 13: Have you considered the idea of expanding the lower latitudinal edge of the figure to 0◦Lat, and include the model/WMO results for the Tropical mean (as shown in Table 5)? Also, for the most poleward latitudinal bin, it looks like the "modelled" values are plotted at a higher latitude than the "observations". This must be due to the total number of data-points used to compute the VSLS value. This should be explained either in the caption or in the text.

*We thank the reviewer for this valuable suggestion. The Figures 13 and also 6 have been extended in this way to include the tropical WMO values and also model values (Fig. 13) including a 0-20°N bin. We have also added a sentence explaining the different latitude centres:*

*Due to the different sampling of the observations and the models the centers of the different latitude bins are not the same for observations and models.*

Figure 15: Why not including panel for WISE_TACTS in this figure?

*This would add too many plots to the paper and we would also like to make clear that this is basically a sensitivity study, why it can be shown with one campaign only. The respective Figure has however been produced and is now shown in the appendix.*

**Reviewer # 3**

The manuscript by Keber et al. describes measurements of short-lived organic bromine gases from several airborne campaigns in the UTLS mostly north of 40◦latitude in different seasons. The authors then evaluate how different combinations of models and emission scenarios compare to observations. The study is relevant to better quantifying the role of organic bromine in catalytic cycles that deplete stratospheric ozone. The main points of the paper are the demonstration of a latitudinal gradient in UTLS organic bromine, a winter maximum in VSL Br gases in the UTLS, recognition that VSL Br concentration near the extratropical tropopause is greater than that at the tropical tropopause, and that current emission scenarios show a range of imperfections that limit their accuracy. I might argue that all of these conclusions are not new, but the measurements, importantly, quantify the levels of UTLS organic Br that provide a solid basis for more detailed and quantitative analysis. Thus, the manuscript will be a valuable contribution to the further understanding of halogen chemistry in the lower stratosphere.

I think the strength of the manuscript is in the presentation of the data and identification of the seasonal and latitudinal variability in the measured Br gases. However, much of the paper focused on the model and emission scenario comparisons to the data. The authors state in their summary, "While it is not the main purpose of this paper to evaluate emission scenarios . . ." they nonetheless spend a large amount of effort to do just that, and they do so in more detail than is needed to demonstrate their point that models have problems. If model comparison to data is a significant part of the message, then I would like to see (perhaps in the Supplement) how the models compare to measurements for CFCs, halons, CH4 or other gases with better understood tropospheric distributions. Does model SF6 compare well to the binned measurements from these missions? This would help provide some

context for looking at the Br distributions and their deviations from model predictions. My suggestion, though, is not to add more model discussion in this paper, but to reduce the level of different comparisons to models that is done in the text and just highlight major issues associated with emission scenarios.

*This suggestion (less model comparisons) is contradictory especially to reviewer #2, who actually suggested more model comparisons. We have chosen to include more model comparison in the appendix, but not in the main part of the paper. We also considered reducing the number of comparisons. This number is rather large, as 4 different combinations of emission scenarios and model are used. The extension to better constrained species (e.g. halons, CFCs etc) would indeed be an interesting aspect, but we feel that it is beyond the scope of this paper, as this would go into a model evaluation direction. Therefore, we have left the level of model comparison in the main text as it is. However, we agree with the reviewer that as a large part of the paper is devoted to comparing the observations with models and different scenarios, the statement "While it is not the main purpose of this paper to evaluate emission scenarios…" is odd. We have removed this statement.*

I also had some trouble sorting through the different manipulations that the authors used to help with the comparisons between measurements and models. The authors go into great detail in how they adjusted and binned the data to presumably reduce variability from variations in dynamics and transport, and to provide a more robust comparisons to model outputs. While I would agree that these might be reasonable adjustments, I found the multiple presentations confusing and left me wondering about what I was viewing in the plots and how the data might look with no adjustments. Perhaps the authors can think about possibly simplifying (or better explaining) the discussion of the adjustments. Even though the goal of the adjustments was to reduce variability due to dynamical factors, the resulting distributions retained significant spread in the data, particularly within -+40K of the tropopause. The authors chose to focus on the average profiles, but it would seem that a more detailed evaluation of the variation (and its comparison to variance in a model) could produce interesting results on sources, transport and variability of Br in the critical region near the tropopause. Maybe a subsequent paper can do this.

*The use of tropopause centred coordinates is something rather common in UTLS studies. It has also been shown by previous studies that variability is considerably reduced using these coordinates. Also, we have shown that the variability in our data is reduced when using tropopause centred coordinates, while the general shape and overall vertical profiles remain quite similar (see Figure 4 and tables 3 and 4, which uses tropopause centred coordinates in comparison to potential temperature). In order to make this point clearer, we have added some explanation of this in the section 3.1. on mean vertical profiles. The description has been updated for more clarity (see also comment by reviewer #1):*

*"For all campaigns, the variability averaged over the four lowest stratospheric bins when using $\Delta\theta$, was always lower than in the 4 lowest bins above the climatological tropopause using $\theta$ as a coordinate(see Tables 3 and 4). This shows that using the tropopause centered coordinate system $\Delta\theta$ reduces the variability in the stratosphere and that this coordinate system is thus best suited to derive typical distributions. In the troposphere, the variability is larger when using $\Delta\theta$ coordinates than for $\theta$, indicating that the variability in the free troposphere is not influenced by the potential temperature of the tropopause."*

*We agree with the reviewer that a point-by-point discussion, which mal also include comparison of other species with better constrained lower boundary values may be the focus of a subsequent paper.*

In section 5, I did not understand the need to go through the linear mixing model to show how the emission scenarios that placed high (or low) CH2Br2 (or CHBr3) in the tropical (or extratropical) tropopause were different from measured values. Data summarized in Table 5 is sufficient to demonstrate the impact of "incorrect" modeled organic Br in the tropics or extra tropics. Why go through assumptions that aren't necessarily realistic? Or you might be able to improve the mixing model with data from other gases, such as SF6, which can be used to perhaps better estimate tropical vs extratropical fractions of air.

*The reason why we want to extend the study to the inorganic bromine is that inorganic bromine is the form which can participate in ozone chemistry and is thus what really needs to be known. Unfortunately, $SF_6$ from input of air from the tropics, respectively the extra-tropics cannot be distinguished and there is no straight forward way of determining these fractions. What would be needed would be tracers with linearly independent time series in the tropics and extra-tropics. This in parts possible with the combination of $CO_2$ and $SF_6$. However, this is a study on its own and high quality $CO_2$ data are needed which are not available for all missions of HALO. This is why we use estimates from previous studies and explicitly state that this is a sensitivity study and not meant to provide a robust value of how large model over- or underestimation of bromine from VSLS source gas injection is.*

*In order to explain better the interest in discussing possible effects on inorganic bromine, we have added a sentence in the introduction and also in the beginning of section 5:*

*Introduction:*

*"Source gases have to undergo a photochemical transformation first before the bromine is in the inorganic form which it can interact with ozone."*

*In section 5 the following sentence has been moved further upward to explain the importance of inorganic bromine earlier:*

*"Inorganic bromine is of key importance, as this is the form of bromine which can influence ozone through e.g. catalytic ozone depletion cycles."*

*To make the need for a mixing model clearer we have added two sentences in the introduction to section 5:*

*"The inorganic bromine from SGI can be derived as the difference between the organic bromine in the source region (tropopause) and the organic bromine still observed or modelled at a certain stratospheric location."*

*And a little further down the same paragraph:*

*"As both regions show different levels or organic bromine source gases, the relative contribution of these source regions needs to be known to derive total bromine which entered the stratosphere and thus also inorganic bromine from SGI."*

Given the goal to provide a more robust assessment of the role of Br in the lowermost stratosphere at N latitudes, and given the availability of multiple models, I would have found valuable some discussion of the actual impact of different Br levels on the ozone budget of the impacted region. How does an uncertainty of -+ 1 ppt VSL Br propagate to modify ozone destruction rates?

*While an uncertainty of 1 ppt of Bry will lead to different responses iin different regions. We have added a statement to give a rough context:*

*"For example, it has been shown that inclusion of about 5 ppt of Bry from VSLS leads to an expansion of the ozone hole area of $\sim$ 5 million km$^2$ and an increase in maximum Antarctic ozone hole depletion by up to 14 % (Fernandez et al., 2017). The impact of bromine on ozone is most pronounced in the lowest part of the stratosphere (Hossaini et al., 2015)."*

*Other comments:*

Minor VSL Br.

A) I would suggest adding profiles of the minor VSL Br in a Supplement.

*Yes, they have been added to an appendix.*

B) The authors state that "the observed decrease with altitude in the stratosphere is consistent with the relative lifetimes of the different compounds". While generally true, this does not seem to be the case for CHBrCl2 compared to CH2Br2 (page 7 line 15). Lifetimes of these compounds differ by a factor of about 3.5, but the gradients appear close to the same. Perhaps something is wrong with the lifetime estimate?

*We have checked the lifetime estimates from Table 1, and they are in line with the WMO 2014 report. However, we observed that the values in Table 3 for the TACTS_WISE dataset where still from an older version. We therefore updated the values in Table 3. The values in Table 4 (PGS were correct).*

*$CH_2BrCl_2$ and $CH_2Br_2$ are not separated chromatographically during normal measurements of GhOST-MS, as this would require too much time (their physicochemical properties are very similar). Therefore, as shown in Sala et al. (2014), a procedure using an observed correlation between the two and of-line measurements of the relative sensitivities and the standard mixing ratios are used to determine CH2Br2 and CHBrCl2 from one single peak. During PGS and WISE the chromatography was tuned in order to separate the two peaks during individual flights (which results in a much poorer temporal resolution) to determine this correlation. We have explained this in brief and referenced to Sala et al. (2014) for a detailed explanation. It could be that the vertical gradient is more determined by lifetime for shorter lived compounds, whereas it is more determined by large scale transport for longer lived compounds. This would explain a decreasing dependency of the vertical gradient with increasing lifetime. We have i) briefly explained the procedure to derive CH2Br2 and CHBrCl2 mixing ratios and ii) added a short discussion on the lifetime dependency.*

*Explanation of separation of the two species in section 2.1.:*
*"As explained in Sala et al. (2014), $CH_2BrCl_2$ and $CH_2Br_2$ are not separated chromatographically during normal*

*measurements of GhOST-MS, as this would require too much time. Instead, a correlation between the two species*

*from either independent measurements or measurements of the two species from dedicated flights are used. Such*

*dedicated flights have been performed during the WISE and the PGS campaign. The procedure how $CH_2BrCl_2$ and*

*CH$_2$Br$_2$ are derived from the single chromatographic peak with this additional information is explained in (Sala et al., 2014)."*

*Discussion of vertical gradients in section 3.1. has been rephrased and focussed on the shorter lived gases, where the relationship is clearer:*

5 *"Of all species discussed here, CHBr$_3$ showed the largest vertical gradients, followed by CHBr$_2$Cl. This is well in line with their atmospheric lifetimes (see Table 1), which will generally decreases with an increase in bromine atoms in the molecule and is shortest for CHBr$_3$, followed by CHBr$_2$Cl.The relationship between lifetime and vertical gradient is less clear for the longer lived species, where vertical profiles are expected to be more influenced by transport."*

C) Tables1 and 3 show the mixing ratio of CH2BrCl as 0.1 – 0.2 ppt (100 – 200 ppq), but the GhOST-MS is reported in Table 1 to have a detection limit of 130 ppq (0.13 ppt). Not sure how this is possible. I note also that the GhOST-MS characteristics are quite different here compared to that reported in Sala et al. (2014). Was there a significant modification to the instrument? Uncertainties.

15 *Yes, there was indeed a significant modification to the instrument: this is a change in the chemical ionisation (CI) gas. While we used pure Methane during SHIVA (Sala et al., 2014) and during TACTS, we had to change this to pure Argon (PGS) for safety reasons and then later to an Argon/Methane mixture (5% Methane, non burnable). Also, we checked the detection limits again. They were indeed much poorer during WISE than during other campaigns, mainly due to the change in CI gas. We have now also added the detection limits and LOD during*
20 *TACTS and PGS to Table 1 and added a comment to that in the text:*

*"While CH$_4$ has been used as chemical ionization gas for the TACTS campaign and for the tropical measurements discussed in Sala et al. (2014), a change in chemical ionization gas was necessary for later measurements due to safety reasons. During the PGS campaign, we used pure Argon, which resulted in very good sensitivities but an*
25 *interference with water vapour. In order to avoid this interference for the mid-latitude (more humid measurements) during WISE, a mixture of Argon and methane (non –burnable, below 5% Methane) was used as ionization gas. These (and some other) changes resulted in different performances of the instrument during different campaigns. Typical performance of the instrument is given for the WISE and PGS campaign in Table 1 for the brominated hydrocarbons."*

30 *The sentence referring to Table 1 two lines further down has been deleted. Table 1 has been modified to include both PGS and WISE characteristics. Also the Table header to Table 1 has been modified accordingly.*

Page 6, top. Please report uncertainties associated with the total organic bromine levels reported here and elsewhere. Organic/Total Br . . . I second the recommendation of one of the other reviewers to be more careful in terminology of total bromine/total organic bromine, etc.

35 *As explained above, we have revisited the manuscript to ensure that the use is now consistent. If total does not refer to the sum of organic and inorganic, it has been specified as total organic or total inorganic bromine.*

*With respect to the uncertainties we have added values for uncertainties in Tables 3 and 4. Note that these uncertainties are only observed variabilities (which are a combination of atmospheric variability and measurements uncertainties), but do not include uncertainties in calibration. As noted above, Table 3 has been updated to new*
40 *values, as those given where based on a previous version of the evaluation software.*

Data availability. I did not see that these data are available in any public archive. Please list how the data from these flights can be accessed.

*Data availability information has been added.*

[revised manuscript text omitted]

---

## Referee Report (RR1)

**Review ACP-2019-796**

The authors addressed the remarks and suggestions made by the referees in a appropriate manner. Some minor issues with formulations remain of which some are probably subject to the final typesetting and language check process.

**1 technical corrections**

- P5 L22-24: *"For EMAC data, we use [...]"* and P5 L28-30: *"For the results presented, [...]"* The "so-called specified dynamics mode" is now explained twice in the same paragraph. The authors should merge those two sentences.

- P6L35: *"[...] was lower by up to 5 ppt lower [...]"* Repetition of "lower".

- P10 L5: *"hese simulations [...]"* A "T" is missing here.

- P12 L1: *"this will result [...]"* t→T

---

## Author Response (AR2)

Comments to the Author:
Dear authors,

please find enclosed two referee reports on the revised version of your manuscript. Both referees are satisfied with the revision, but one or the referees has still one minor issue that should be considered before publication in ACP. Additionally, I would like to ask you to consider the technical corrections given below before submitting the next revision of your manuscript:

Best regards, Farahnaz Khosrawi

Technical corrections:
P2, L1-17: Spaces between the references are missing. *Done everywhere*
P3, L29: One closing parentheses obsolete (Ziska et al reference). *removed*
P5, L41: "are" at the end of sentence obsolete.*removed*
P6, L20: Late -> late *changed*
P6, L40: space between the end of the first and begin of the second sentence missing. *done*
P7, L9: space between "coordinates" and "(see Table 3 and 4) missing. *done*
P7, L20: close parentheses first (before the full stop). done
P8, Ö10: remove "e" after fullstop. *done*
P10, L5: hese -> These *done*
P10, L39: better to write "show similar distributions". *done*
P11, L13: table 5 -> Table 5 *done*
P11, L42: Citations should be embedded in the text (this parentheses around the year). *done*
P12, L9: skip parentheses around see e.g. Engel and Rigby. *Changed to "(so called source gas injection, SGI, see e.g. Engel and Rigby (2018))"*
P16, L17: add "DFG" in parentheses (after German Science Foundation). *done*
P16, L17: foundation -> Foundation *done*
P22, Table 1 caption: space between global and stratospheric obsolete. *removed*
P24, Table 5 caption: space between text in parentheses and "used" missing. *Space included*
P26, Figure 4: Add "N" to the coordinates given in the figure and caption.
P26, Figure 4 caption. use Greek letter for sigma. *done*
P27, Figure 5 caption: space between "CHBr3" and "(bottom)" missing. *done*
P29, Figure 7 and 8: add "N" to coordinates given. *done*
P30, Figure 8 caption: are you showing here a three month average or monthly averages of these three months. This is not clear. *We have made this clearer by stating: "The data have been averaged for the period from January to March, i.e. representative of the time period covered by the PGS campaign."*
P31, Figure 9, 10, 11, 12 caption: blanc -> blank. *Done in Fig 9, but obsolete for 10-12 due to the use of "As Fig …"*
P32, Figure 10 caption: Simply write as Fig 9, but using........ *done*
P33, Figure 11 caption: same here, write as in Fig 9 and 10, but using..... *done*
P34, Figure 12 caption: same here, write as in Fig 9-10, but using........ *done*
P36, Figure 14: add N to the coordinates given. *done*
P26, Figure 14 caption: Bromine -> bromine *done*
Supplement: Also here add " N" to the coordinates given in the figures. *done*
Use for sigma the Greek letter. Shorten Figure captions by writing as in Fig. XX, but .........(give here the differences to the previous figure).

*We have referenced the appropriate Figures in the main text by adding as Fig XX in the section titles of the supplement. In the captions, we have used as Fig SY to reference other Figures in the supplement. In this way the Figure captions in the supplement are "self-contained".*

Rev. 1:

The authors' have addressed most all of my comments, except for one. I am still concerned about the comment about relationship between lifetime and vertical gradient. While I accept the authors' comments that lower lifetime gases will have higher gradients, and that longer lived compounds will have gradients impacted more by transport, I would like the authors to more clearly state that CHBrCl2 is an outlier. They seem to just gloss over this obvious discrepancy in their comment. The lifetime of CHBrCl2 is much closer to CHBr2Cl, but it's gradient is much closer to CH2Br2 and CH2BrCl (and lifetime is a factor of >3 lower!). If transport is the major factor, then this would imply a very different proportion of this gas relative to others that is mixing in to change the observed gradient. While this is not a huge part of the paper, the information contained in the gradients is highly relevant and requires explanation. If there is something different about CHBrCl2, then it should be pointed out. Though not directly to this point, it also seems that the seasonal difference for bromoform is notably different from other gases. The lifetime ratio of all gases between fall and winter is about 2.1. The ratio of the observed gradient however is less: (1.5 - 1.8) for gases other than bromoform. The ratio of gradient between fall and winter for bromoform is only 1.1. If lifetime is the major control, I would think that bromoform would be more in line with the others. So something different appears to be influencing bromoform. Perhaps this could be compared to model predictions.

*We apologize for not taking this comment into account. It was lost under the large amount of reviewer comments.*

*With respect to the first point (CHBrCl2 being an outlier), we have now added the following:*

*"In particular, the vertical gradient of $CHBrCl_2$ is closer to the vertical gradient of $CH_2Br_2$ than to that of $CHBr_2Cl$, although the lifetime should be closer to $CHBr_2Cl$. This could be related to the way that $CHBrCl_2$ is derived, as it is not chromatographically separated from CH2Br2 (see section 2.1 and Sala et al. (2014)). "*

*WQith respect to the second issue mentioned by the reviewer: This is due to the fact that during PGS $CHBr_3$ was often already nearly completely depleted at 20-30 K above the tropopause. It was already stated in the manuscript at the end of the second paragraph in section 2.1.:*

*"The strongest vertical gradients with respect to both $\theta$ and $\Delta\theta$ were observed during the winter campaign PGS, with the exception of $CHBr_3$, which was nearly completely depleted for all campaigns at 40 K above the tropopause and thus shows very similar averaged gradients over this potential temperature region. When evaluated only for the first 20 K above the tropopause, the gradient of $CHBr_3$ was also highest during PGS."*

*For more clarity we have changed the last sentence as follows:*

*"... the gradient of $CHBr_3$ was also much larger during PGS then during WISE and TACTS."*

Rev. 2:
The authors addressed the remarks and suggestions made by the referees in a appropriate manner. Some minor issues with formulations remain of which some are probably subject

to the final typesetting and language check process.

1 technical corrections
_ P5 L22-24: "For EMAC data, we use [...]" and P5 L28-30: "For the results presented, [...]" The "so-called specified dynamics mode" is now explained twice in the same paragraph. The authors should merge those two sentences. *Second explanation of specified dynamics deleted*
_ P6L35: "[...] was lower by up to 5 ppt lower [...]" Repetition of "lower". *Second lower deleted*
_ P10 L5: "hese simulations [...]" A "T" is missing here. *done*
_ P12 L1: "this will result [...]" t!T *done*

**Additional changes implemented to improve the manuscript**

In addition to the changes suggested by the editor and the reviewers, we have implemented a few stylistic changes. In particular, we have reduced the use of the word "value" and replaced it by "mixing ratio" or XXXX to be more specific. We have also eliminated a few cases where formulas were in bold type and where subscripts had not been properly implemented (e.g. Bry to $Br_y$). In addition to these changes we have in some occasions split long sentences. All these changes can be seen in the track-changed manuscript and have in no case any implication for the meaning.

[revised manuscript text omitted]